# The Computational Advantage of Depth in Learning High-Dimensional Hierarchical Targets

Yatin Dandi[1,2], Luca Pesce[1], Lenka Zdeborová[2], and Florent Krzakala[1]

[1]Information, Learning and Physics Laboratory. Ecole Polytechnique Fédérale de Lausanne (EPFL), CH-1015 Lausanne, Switzerland.
[2]Statistical Physics of Computation Laboratory. Ecole Polytechnique Fédérale de Lausanne (EPFL), CH-1015 Lausanne, Switzerland.

## Abstract

Understanding the advantages of deep neural networks trained by gradient descent (GD) compared to shallow models remains an open theoretical challenge. In this paper, we introduce a class of target functions (single and multi-index Gaussian hierarchical targets) that incorporate a hierarchy of latent subspace dimensionalities. This framework enables us to analytically study the learning dynamics and generalization performance of deep networks compared to shallow ones in the high-dimensional limit. Specifically, our main theorem shows that feature learning with GD successively reduces the effective dimensionality, transforming a high-dimensional problem into a sequence of lower-dimensional ones. This enables learning the target function with drastically less samples than with shallow networks. While the results are proven in a controlled training setting, we also discuss more common training procedures and argue that they learn through the same mechanisms.

Understanding the computational benefits of deep neural networks over their shallow counterparts is a central question in modern machine learning theory [76, 87]. While shallow models can approximate any complex functions [27], deep networks almost universally exhibit remarkable advantages in practice [49, 4]. There has been much progress in approximation theory on the advantage of depth (see e.g. [62, 79, 61, 68] and reference therein), however, the dynamics of learning with gradient descent is a more complex question. A fundamental open problem is thus:

*Can one quantify the computational advantage of deep models trained with gradient-based methods with respect to shallow models in some analyzable setting?*

One line of work on GD-based methods in deep networks leading to interesting results is in the setting of deep *linear* network —see e.g. [73, 46, 12, 51, 40]. While deep linear networks offer valuable insights into nonlinear learning dynamics, their simplicity renders them insufficient to capture the complexity of hierarchical feature learning.

Another popular line of research is to study the dynamics of gradient-based methods learning multi-index functions with shallow models [19, 17, 39, 22, 1, 81]. Multi-index functions provide a rich class of targets, but their efficient learnability by shallow two-layer networks [11, 52] undermines their utility as benchmarks for understanding the computational advantages of depth. This motivates the following consideration:

*What is the natural model of targets to consider for understanding the emergent computational advantage of depth when training with gradient-based methods?*

The present paper addresses both these questions. To answer the latter, we introduce a class of target functions designed to probe the hierarchical structure and computational potential of deep networks. These *Multi-Index Gaussian-Hierachical Target* (MIGHT) functions encapsulate a hierarchy of latent subspaces with varying dimensionalities. We then proceed to answer the former interrogative

by analyzing the learning dynamics of multi-layer neural networks on such targets, providing a characterization of the computational advantages afforded by depth. We show how depth enables a hierarchical decomposition of tasks, reducing the effective dimensionality at each layer, and leading to a quantifiable improvement in sample complexity over shallow models.

# 1 Hierarchical Targets and Main Results

## 1.1 Single-Index Gaussian Hierarchical Targets

Our simpler setting, where the task —using Gaussian i.i.d. data $\{\mathbf{x}_\mu\}_{\mu=1}^n \in \mathbb{R}^{n\times d}$— to learn the following Single-Index Gaussian Hierarchical Target (SIGHT) function class that we write in three equivalent forms as:

$$f^\star(\mathbf{x}) = g^\star\left(\frac{\mathbf{a}^{\star\top} P_k\left(W^\star\mathbf{x}\right)}{\sqrt{d^{\varepsilon_1}}}\right), \ \mathbf{x} \in \mathbb{R}^d, \tag{1}$$

$$= g^\star\left(\frac{\mathbf{a}^{\star\top} P_k\left(\mathbf{z}^\star\right)}{\sqrt{d^{\varepsilon_1}}}\right), \ \mathbf{z}^\star = W^\star\mathbf{x} \in \mathbb{R}^{d^{\varepsilon_1}}, \tag{2}$$

$$= g^\star\left(h^\star\right), \ h^\star = \mathbf{a}^\star \cdot P_k\left(\mathbf{z}^\star\right)/\sqrt{d^{\varepsilon_1}} \in \mathbb{R}. \tag{3}$$

Here $P_k$ is a fixed polynomial applied component-wise, and $d^{\varepsilon_1}$ denotes the dimensionality of the *second-layer features* (non-linear features) in the intermediate layer, which we choose to be $\varepsilon_1 \in (0,1)$. The *first-layer* features (linear features) are $\mathbf{z}^\star = W^\star\mathbf{x}$, where $W^\star \in \mathbb{R}^{d^{\varepsilon_1}\times d}$ has orthonormal unit vectors as rows, and $\mathbf{a}^* \in \mathbb{R}^{d^{\varepsilon_1}}$ is choosen randomly from a fixed distribution. We refer to the variable $h^*$ as the *index* in the name of the class. This construction, a generalization of the hidden manifold model [43], is motivated by the compositional structure present in real-world functions and by the analysis carried over by [84, 66]. The strictly decreasing dimensionality of the features across depth allows us to avoid the pitfall of the original hidden manifold model [43] that turns out to be equivalent to a Gaussian linear target [41, 45, 64].

## 1.2 Multi-Index Gaussian Hierarchical Targets

A simple generalization of the above construction is to include many non-linear features, leading to Multi-Index Gaussian Hierarchical Targets (MIGHT) defined as:

$$f^\star(\mathbf{x}) = g^\star\left(h_1^\star(\mathbf{x}), \ldots, h_r^\star(\mathbf{x})\right), \tag{4}$$

where

$$h_m^\star(\mathbf{x}) = \frac{1}{\sqrt{d^{\varepsilon_1}}}\mathbf{a}_m^{\star\top} P_{k,m}\left(W_m^\star\mathbf{x}\right), \ m = 1, \ldots r, \tag{5}$$

with now $r$ directions, each with their own layer weights ($\mathbf{a}_m$ and $W_m^\star$), and polynomials ($P_{k,m}$).

## 1.3 Deep Multi-Index Hierarchical Targets

Finally, we define the *deep* version of MIGHTs as

$$f^\star(\mathbf{x}) = g^\star\left(h_{L,1}^\star(\mathbf{x}), \ldots, h_{L,r}^\star(\mathbf{x})\right), \tag{6}$$

with Gaussian data $\{\mathbf{x}_\mu\}_{\mu=1}^n \in \mathbb{R}^d$, and where each features $\mathbf{h}_\ell^\star(\mathbf{x}) \in \mathbb{R}^{d_\ell^\varepsilon}$ are recursively defined as:

$$h_{\ell,m}^\star(\mathbf{x}) = \frac{1}{\sqrt{d^{\varepsilon_{\ell-1}-\varepsilon_\ell}}}\mathbf{a}_{\ell,m}^{\star\top} P_{k,m,\ell}\left(\mathbf{h}_{\ell-1,\{1+(m-1)d^{\varepsilon_{\ell-1}-\varepsilon_\ell},\ldots,md^{\varepsilon_{\ell-1}-\varepsilon_\ell}\}}^\star(\mathbf{x})\right), \tag{7}$$

with $\ell = 1 \cdots L$, $m = 1 \cdots d^{\varepsilon_\ell}$ and where $\mathbf{a}_{\ell,m}^\star \in \mathbb{R}^{d^{\varepsilon_{\ell-1}-\varepsilon_\ell}}$ acts on the $m_{th}$ block of the previous layer feature $\mathbf{h}_{\ell-1}^\star(\mathbf{x})$ (each of them being of size $d^{\varepsilon_{\ell-1}-\varepsilon_\ell}$). Again $P_{k,m,\ell}$ are fixed polynomials for $\ell = 1, \cdots, L-1$; $d^{\varepsilon_\ell}$ denotes the dimensionality of the features at layer $\ell$, which we choose to be strictly decreasing across depth, i.e., $1 > \varepsilon_1 > \varepsilon_2 > \cdots > \varepsilon_{L-1} > 0$, with $\mathbf{h}_L^\star \in \mathbb{R}^r$ being finite-dimensional. This "tree-like" construction ensures that for any layer index $\ell \in 1, \cdots, L$, the hidden features $h_{\ell,m}^\star(\mathbf{x})$ remain independent for different index $m \in 1, \cdots, d^{\varepsilon_\ell}$. (Appendix D.1)

Finally, the $1^{\text{st}}$-layer features are defined as

$$\mathbf{h}_1^\star(\mathbf{x}) = \mathbf{z}^\star = W^\star\mathbf{x}, \tag{8}$$

where $W^\star \in \mathbb{R}^{d^{\varepsilon_1}\times d}$ has orthonormal unit vectors as rows. By explicitly incorporating multiple levels of non-linear feature transformations, each associated with a progressively reduced latent dimensionality, it models the deep hierarchical structure is a feature of complex real-world tasks, see e.g. [55, 49, 65, 24, 74]. We exemplify SIGHT (1) and MIGHT (4) functions in Fig. 3, and their deep version (6), where the tree structure of the deep version of these targets is apparent, in Fig. 4.

## 1.4 Learning Model

We now consider learning SIGHT and MIGHT functions $f^\star(\mathbf{x})$ through an $L$-layer neural network, that is a standard multi-layer perceptron:

$$\hat{f}_\theta(\mathbf{x}) = b_L + \mathbf{w}_L^\top \sigma(\mathbf{b}_{L-1} + W_{L-1} \cdots \sigma(\mathbf{b}_1 + W_1(\mathbf{x}))), \tag{9}$$

where $\theta$ denotes the ensemble of trainable parameters $\{\mathbf{b}_\ell, W_\ell, \ell = 1 \cdots L\}$. The hidden layer weights have dimension $W_\ell \in \mathbb{R}^{p_\ell \times p_{\ell-1}}$ for $\ell \in \{2, \cdots, L-1\}$ with readout layer $W_L \in \mathbb{R}^{p_L}$ and first layer $W_1 \in \mathbb{R}^{p_1 \times d}$, and the biases $\mathbf{b}_\ell$ are in $\mathbb{R}^{p_\ell}$. We shall consider Empirical Risk Minimization (ERM) of the square loss $\hat{\mathcal{R}}(\{\mathbf{x}_\mu\}) = \sum_{\mu=1}^n \left( f^*(\mathbf{x}_\mu) - \hat{f}_\theta(\mathbf{x}_\mu) \right)^2$ with gradient descent.

## 1.5 Main Results in a Nutshell

The backbone of our results is the analysis of the asymptotic performance of learning SIGHT and MIGHT functions using multi-layer networks

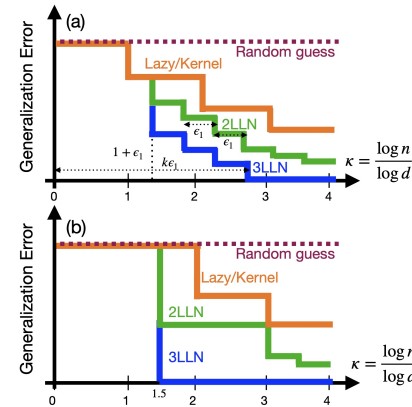

Figure 1: An illustration of the phase transitions in learning SIGHT according to the main Theorem 1 denoting the computational advantage of depth for two different target model: **(a)** generic shallow SIGHT function (eq. (1)) and **(b)** the example in eq. (11).

trained with Gradient Descent on Gaussian data, as both $n$ (the number of data) and $d$ (the dimension of the data) grow to infinity. We unveil a series of sharp thresholds in the sample complexity ratio $\kappa = \frac{\log n}{\log d}$ where neural networks learn the target with increasing accuracy. To summarize:

• Our targets offer a solvable playground to unveil the computational advantage of deep networks over shallow ones. The learning mechanism can be viewed as the reduction of the "effective dimension" in which networks trained on $f^\star(\mathbf{x})$ successively reduces the dimensionality of the search space: $d^{\varepsilon_1} \to d^{\varepsilon_2} \to d^{\varepsilon_3}, \cdots, \to r$. Depth acts as a progressive filter that *distills* data into lower-dimensional representations (a coarse-graining mechanism akin to renormalization in physics), enabling the learning of subsequent layers.

• We focus the rigorous analysis in the paper on the case of shallow SIGHT functions (eq. (1)) learned by $3-$layer networks, where each layer is trained sequentially and independently. We prove that a three-layer network trained in a layer-wise fashion can learn a SIGHT function $f^\star(\mathbf{x})$ efficiently. Specifically, the network first recovers $W^\star$ using $\tilde{O}(d^{\varepsilon_1+1})$ samples, then reconstructs $h^*$ with $\tilde{O}(d^{k\varepsilon_1})$ samples (with $k$ denoting the degree of $P_k$, in case $k\varepsilon_1 < 1 + \varepsilon_1$ both happen at $1 + \varepsilon_1$), and finally fits $f^\star$ as a function of $h^\star$ using only $\tilde{O}(1)$ samples. This sample complexity aligns with predictions from the dimension-reduction/coarse-graining perspective, where earlier layers successively reduce the effective dimensionality of the learning problem. We also present additional results for deeper targets and networks.

• We further explore the problem through numerical simulations using more realistic training procedures than those covered by the theorems. Our results suggest that the dimensionality reduction mechanism remains broadly applicable. Notably, we illustrate such a phenomenon using 3-layer networks training all the layers jointly with standard backpropagation. We provide the code of our simulations at https://github.com/IdePHICS/ComputationalDepth.

## 1.6 Related Works

**Random Feature Models** — A key attribute enabling the effectiveness of neural networks is their ability to adjust to low-dimensional features present in the training data. However, interestingly, much of the current theoretical understanding of neural networks comes from studying their lazy regime, where features are not learned during training. One of the most pre-eminent examples of such "fixed-features" regimes are Random Feature (RF) models, initially introduced as a computationally efficient approximation to kernel methods by [69], they have gained attention as models of two-layer neural networks in the lazy regime. One of the main motivations is their sharp generalization guarantees in the high-dimensional limit [38, 42, 58, 59, 86, 34]. As mentioned, however, the performance of such methods, and of any kernel method in general, is limited. A fundamental theorem in [58] states that only a polynomial approximation up to degree $\kappa_{\mathrm{RF}}$ of any target $f^*$, with $\kappa_{\mathrm{RF}} = \min(\kappa_1, \kappa_2)$ when learning with $n = d^{\kappa_1}$ data and $p = d^{\kappa_2}$ features.

While even shallow networks can surpass these limitations [40, 17, 31], this relation for $\kappa_{\mathrm{RF}}$ plays a fundamental role in our analysis.

**Multi-index Models —** Despite the theoretical successes in describing fixed feature methods, the holy grail of machine learning theory remains a rigorous description of network adaptation to low-dimensional features. A popular model to study such low-dimensional structure in the learning performance is the *multi-index model*. For this class of target (denoted as $f^\star_{MI}$), the input datum $\mathbf{x}$ is projected on a $r-$dimensional subspace $W^\star = \{\mathbf{w}^\star_j, j \in 1 \cdots r\}$ and the input-output relation depend solely on a non-linear map $g^\star$ of these $r$ (linear) features :

$$f^\star_{\mathrm{MI}}(\mathbf{x}) = g^\star(\mathbf{x}^\top \mathbf{w}^\star_1, \ldots, \mathbf{x}^\top \mathbf{w}^\star_r) \tag{10}$$

While the information theoretical performance is well understood [18, 14], there has been intense theoretical scrutiny to characterize the sample complexity needed to learn multi-index models with shallow models. On the one hand, kernel methods can only learn a polynomial approximation [58]; on the other hand, the situation in neural networks appears more complicated at first as the hardness of a given $f^\star_{\mathrm{MI}}$ has been characterized by the "information" and "leap" exponents [19, 2, 30, 28, 32, 11, 52, 21, 77, 13]. It was shown, however, that simple modification of vanilla Stochastic Gradient Descent (SGD), such as Extra-Gradient methods or Sharpness Aware Minimizers, are able to attain sample complexity corresponding to Statistical Query (SQ) lower bound [11, 52], and are essentially optimal up to polylog factors in the dimension [28, 81]. A motivation of the present work is to go beyond such limitations and analyze hierarchical feature learning.

**3-Layers Networks —** Substantial effort has been devoted to investigating the approximation advantages conferred by deeper neural network architectures [79, 36, 70]. However, it remains unclear how these approximation gaps translate into sample complexity ones for neural networks when trained through gradient descent. An important step towards the role of depth in neural networks has been carried over by [84, 66], who proved separation results between the test performance of 2 & 3 layer networks. More precisely, [84] proved that 3-layer architectures with a fixed first layer can learn a target function of the form $g^\star(\mathbf{x}^\top A \mathbf{x})$ in $n = \tilde{O}(d^4)$ samples through a single-gradient step on the second layer, where $\mathbf{x} \in \mathbb{R}^d$ and $A \in \mathbb{R}^{d \times d}$. In contrast, 2-layer networks require a super-polynomial number of samples in terms of the degree of $g^\star$. [66] subsequently improved the sample complexity to $\tilde{O}(d^2)$ and generalized the result to functions of $p_{th}$-order polynomials. [37] further extended these results to learning multiple-nonlinear features. We go beyond these results to prove stronger separation results by analyzing fully trained networks without a fixed first layer.

**Coarse-graining —** The dimensionality reduction we describe is closely related to the concept of learning features across different scales. This idea has been explored in the context of machine learning through connections with the renormalization group [85] in physics, where each scale corresponds to a distinct set of features. Such techniques have inspired studies of deep neural networks [57, 53, 56]. Here, we present a concrete example of such a coarse-graining mechanism, illustrating how hierarchical structures can be analyzed explicitly.

**Hierarchical data models —** A key insight in explaining the superiority of deep over shallow networks is that depth enables neural networks to progressively reduce the effective dimensionality of the learned data representation [80, 5, 3, 72, 8, 35, 23]. This aligns with the latent hierarchical structure observed in real-world data, which deep models exploit through layer-wise composition. Leveraging these observations, the construction of hierarchical data models has been central in theoretical analysis [65, 6, 2, 74, 75, 24, 23]. Tree-like structures analogous to our SIGHT (1) and MIGHT (4) are considered in [68], leading to provable approximation benefits across depth. More generally, [67] linked compositional sparsity to efficient computability. Since learnability subsumes computability, such computational sparsity is expected to be necessary for efficiently learnable functions, further supporting our construction. However, since efficiently learnable functions form a strict subset of computable functions, functions learnable by gradient descent must possess additional structure on top of being compositions of local/sparse functions. Our class of targets shows that one such additional structure is obtained by insuring sufficient regularity/stability w.r.t intermediate features at each step. In our setting, such regularity is ensured by the presence of low-degree dependence on lower level features. This mirrors the dependence structure in real data, where for instance, the target labels for images/language datapoints have direct correlations with low-level features such as edges or bi-gram, trigram counts. Results supporting the benefits of deepth for tree-like hierarchical models have been provided by [65] and [23] who consider tree-structured inputs, in contrast to our focus on structured targets (but non structured input).

**Universality** — A crucial role in our analysis is played by the asymptotic Gaussianity of $\mathbf{h}_\ell^\star(\mathbf{x})$ which leads to a simplified description of how dependencies on $\mathbf{h}_\ell^\star(\mathbf{x})$ propagate to lower-level features. Such a property is a crucial component of the analysis in [66, 84]. Specifically, [66, 84] showed that the projection of $g^\star(\langle \mathrm{He}_k(\mathbf{x}), A \rangle)$ on degree-$k$ Hermite polynomials lies along the non-linear feature $\langle \mathrm{He}_k(\mathbf{x}), A \rangle$ while $g^\star$ has vanishing projections on lower degree terms. We generalize these results to describe the projections on all degree components.

## 2 Heuristic argument underlying the main results

Before presenting the main technical results, we describe here a heuristic argument describing the narrative behind the results. For concreteness, we focus here on learning a shallow SIGHT function (1) as a first step toward a broader understanding. For concreteness, we will discuss the following example (later used in Fig. 2):

$$f^\star(\mathbf{x}) = \tanh\left( \frac{{\mathbf{a}^\star}^\top P_3\left(W^\star \mathbf{x}\right)}{\sqrt{d^{\varepsilon_1 = 1/2}}} \right) \tag{11}$$

with a polynomial $P_3(x) = \mathrm{He}_2(x) + \mathrm{He}_3(x)$ (the second and third Hermite polynomials), $\varepsilon_1 = 1/2$, and discuss the performance of different learning architectures, highlighting the dimensionality reduction due to feature learning. The learning dynamics for general SIGHT (eq. (1)) and the particular example above (eq. (11)) are illustrated in Fig. 1 respectively in the top and bottom panel.

a) **Kernel methods, or random feature models**, can only learn a polynomial approximation of degree $\kappa$ in the Hermite basis of $f^\star$ if $n = O(d^\kappa)$ [58]. This is a strong limitation that leads to poor performance as the learning method is not sensitive to the presence of relevant low-dimensional structure, but rather only to the degree of the target. In the example (11), the lowest (Hermite) polynomial order is quadratic in $\mathbf{x}$ (as can be seen by expanding the $\tanh$): learning it thus requires $n = O(d^2)$ samples of data for a kernel method to beat random performance. Learning the cubic approximation would requires $n = O(d^3)$ samples, etc. The corresponding thresholds are sketched in orange in Fig. 1.

b) We now turn to **two layer net** of the form (we do not write explicitly the additional biases for clarity) with a number of neurons $p$ at least of order $\Theta(d^{k\varepsilon_1 + \delta})$

$$\hat{f}_\theta(\mathbf{x}) = \mathbf{w}_2^\top \sigma(W_1(\mathbf{x})) \tag{12}$$

Thanks to feature learning, such architecture should perform better: Indeed, for $W_1$ to learn the $d \times d^{\varepsilon_1}$ first-layer feature matrix $W^\star$, we need *at least* $n = O(d \times d^{\varepsilon_1})$ data. If $n \gg d^{1+\varepsilon_1}$, we thus expect that $W_1$ correlates with $W^\star$. Intuitively, $W_1$ is then close to a noisy random rotation of $W^\star$ and behaves roughly as $W_1 \approx Z_1 W^\star + Z_2$ (with $Z_1$ and $Z_2$ are essentially random matrices). The two-layer neural net thus now behaves as:

$$\hat{f}_\theta(\mathbf{x}) \approx \mathbf{w}_2^\top \sigma\left(Z_1 \mathbf{z}^\star + Z_3\right). \tag{13}$$

Fitting now the outer weights $\mathbf{w}_2$ leads, once again, to a random feature model, but now applied to the target eq. (2) seen as a function of $\mathbf{z}$ instead of eq (1) seen as a function of $\mathbf{x}$. This leads to an effective Random Feature model with respect to *the lower dimensional vector* $\{\mathbf{z}^\star \in \mathbb{R}^{d^{\varepsilon_1}}\}$. Thanks to this dimensional reduction from dimension $d$ to the effective one $d^{\varepsilon_1}$, we just need $n = (d^{\varepsilon_1})^\kappa$ samples of data to now fit a $\kappa$−th degree polynomial approximation of $f^\star$. This is a drastic improvement. Coming back to the example: with $n = O(d^{1+\varepsilon_1 = 1.5})$, $\kappa = 1.5$, data samples a two-layer net learns the first layer representation $W^\star$, leading to a dimensionality reduction from $d$ to $\sqrt{d}$. From $n = O(d^{3\varepsilon_1 = 1.5})$, $\kappa = 1.5$, we are also able to fit a (Hermite) polynomial approximation of degree 3 of the target viewed as a function of $\mathbf{z}$. The next order in the expansion of (11) is power 6 in $\mathbf{z}$, and thus will be fitted at $\kappa = 3$. We discuss the extension of the above arguments to two-layer networks trained with a general gradient-based algorithm in App. B .

c) We now finally consider **a three-layer neural networks**, with width $p_2 = p_1 = \Theta(d^{k\varepsilon_1 + \delta})$:

$$\hat{f}_\theta(\mathbf{x}) = \mathbf{w}_3^\top \sigma(W_2 \sigma(W_1 \mathbf{x})). \tag{14}$$

We still expect that $W_1$ learns the first-layer features $W^\star$ when $n \gg d^{1+\varepsilon_1}$, at which point:

$$\hat{f}_\theta(\mathbf{x}) \approx \mathbf{w}_3^\top \sigma(W_2 \sigma\left(Z_1 \mathbf{z}^\star + Z_3\right)) \tag{15}$$

However, contrary to the previously depicted shallow case, three-layer networks can further approximate $h^\star$ by updating the second layer. With each power of $d^{\varepsilon_1}$ we expect to be fit an

additional power approximation of $h^\star$ and, in particular, with $n = O(d^{k\varepsilon_1})$, we expect the second layer preactivation $h_2(\mathbf{x}) = W_2\sigma(W_1\mathbf{x})$ to correlate completely with the $(k-\text{polynomial})$ features $h^\star$. Therefore, denoting again $Z_4, Z_5$ as random matrices, a 3-layer network now acts as:

$$\hat{f}_\theta(\mathbf{x}) \approx \mathbf{w}_3^\top \sigma\left(Z_4 h^\star + Z_5\right), \tag{16}$$

Fitting now $\mathbf{w}_3$ leads to a random feature model on the *scalar* $h$, which can be fitted perfectly with any growing number of samples $n$. In other words, through successive coarse-graining from $d^{\text{eff}} = d \to \sqrt{d} \to 1$, we have reduced the dimension from a diverging one ($d$) to a finite one.

Note that generalization error as plotted in Fig. 1 can jump for two reasons as $n$ increases: either because of a reduction of the dimension $d_{\text{eff}}$, or because of an increase of polynomial fitting power within this dimension. The phenomenology is a bit simpler in the particular example (11), where the advantage of a three-layer net is considerable: for $n = O(d^{1.5})$, the network learns to represent the non-linear features $h^\star$ directly, and thus can learn the entire function.

While such parameter counting sounds reasonable, this heuristic may fail for general data distributions, as high-degree polynomials may localize on low-dimensional structures and develop heavy tails. However, for Gaussian and spherical measures, isotropy and hypercontractivity ensures that such polynomials remain delocalized and well-concentrated [Lemma 5]. Our analysis relies on proving that such a property holds under feature learning, and even for deep non-linear hidden features.

This scenario, illustrated in Fig. 1, extends *mutatis mutandis* to generic deep multi-layer MIGHT functions, where a sequence of transitions emerges progressively across the layers. Consider for instance the following hierarchical target function from eq. (7) (see also Fig. 4):

$$f^\star(\mathbf{x}) = \tanh\left(\frac{\mathbf{a}^{\star\top} P_{k'}\left(\mathbf{h}_2^\star\right)}{\sqrt{d^{\varepsilon_2}}}\right), \quad h_{2,m}^\star = \left(\frac{\mathbf{a}_{2,m}^{\star\top} P_k\left(\mathbf{h}_1^\star = W^\star\mathbf{x}\right)_{\{1+(m-1)d^{\varepsilon_1-\varepsilon_2},\ldots,md^{\varepsilon_1-\varepsilon_2}\}}}{\sqrt{d^{\varepsilon_1-\varepsilon_2}}}\right).$$

In this case we expect a reduction from $d \to d^{\varepsilon_1} \to d^{\varepsilon_2} \to 1$. The first one arises at $n = O(d^{\varepsilon_1+1})$ when learning $W^\star$, then at $n = O(d^{k\varepsilon_1+\varepsilon_2})$ (to learn all the $d^{\varepsilon_2}$ polynomials, each of them requiring $d^{k\varepsilon_1}$ data) and finally at $n = O(d^{k'\varepsilon_2})$ to learn the activation in the $\tanh$ (a single $k'$ polynomial in dimension $d^{\varepsilon_2}$). Note that while these must proceed in this order, some of these jumps can happen at the same value of $\kappa$. For instance, if $k'\varepsilon_2 < k\varepsilon_1 + \varepsilon_2$, then the last two jumps arise simultaneously.

## 3    Main Theoretical Results

We now turn to the main part of our results that describe learning of the SIGHT and MIGHT function classes with deep neural networks trained by gradient descent. We present a rigorous analysis of gradient-based Empirical Risk Minimization (ERM). Since a complete rigorous analysis of gradient descent in deep networks is extremely challenging – and hitherto elusive– we first present a rigorous description for the SIGHT target of eq. (1) under a specific deep-learning schedule. This approach enables us to provide precise theorems that capture the hierarchical learning process. We analyze the following training procedure:

- **Initilization:** The parameters of the model $\hat{f}_\theta(\mathbf{x}) = \mathbf{w}_3^\top\sigma(W_2\sigma(W_1\mathbf{x}))$ are initialized as $W_{1,i} \sim U(\mathbb{S}^{d-1}(1))$ for $i \in [p_1]$, $W_2 = \mathbf{I}_{p_1}$, and $w_{3,i} = 1$ for $i \in [p]$, where $U(\mathbb{S}^{d-1}(1))$ denotes the uniform distribution on the unit sphere in $\mathbb{R}^d$.
- **Layer-wise training**: (i) We first perform a pre-determined number $T_1$ of gradient updates on the first layer $W_1$ on independent batches of data for each step [30]. (ii) Subsequently, we re-initialize the second layer $W_2$ do a single large gradient step. (iii) Finally, we update $\mathbf{w}_3$ through ridge regression. Layer-wise training procedures are a common simplifying assumption in the analysis of two-layer networks [29, 1, 30]. A complete analysis of the joint training remains open even for two-layer networks except for training of layers at differing time scales [20, 22]. An interesting direction for future work is to rigorously show separation results between deep and shallow networks by constructing targets where joint training of the layers provably surpasses layer-wise training. A first attempt in this direction was considered in [7], who illustrated the advantage of joint-training through the mechanism of "backward feature correction".
- **Neuron-wise spherical projections**: While updating the first layer parameters $W_1$, we utilize spherical-gradient and project each neuron onto the unit sphere. Such spherical projections are commonly utilized in the literature on two-layer networks [19, 1].

• **Pre-conditioning of gradient for the second layer**: We use a pre-conditioning of the gradient step — broadly used in various optimization schedules (e.g. Adam [47])— using the sample-covariance of the features as preconditioning matrix, i.e.,

$$\Delta W_2 = -\eta(\frac{1}{n}\sigma(XW_1^\top)^\top(\sigma(XW_1^\top))^{-1}\nabla_{W_2}\mathcal{L}$$

Through the feature map $\mathbf{x} \to \sigma(W_1\mathbf{x})$, the updates of $W_2$ in parameter space translate to updates to $h_2(\mathbf{x})$ in function space. Without such pre-conditioning, online SGD leads to a worse sample complexity of $\Theta(d^{2k\varepsilon_1})$ as in the single-step analysis of [66], as we explain further in Appendix C.16. Although pre-conditioning plays an important role in the proof scheme, we argue that the core of the results hold in more realistic routine in Sec. 5.

With this algorithm, we can now study gradient descent and demonstrate the learning of a class of SIGHT function. The theorem will assume the following conditions:

• **Uniform weighting**: We set $\{a_i^\star = 1 \text{ for all } i \in 1 \cdots d^{\varepsilon_1}\}$: This operation ensures isotropic dependence along all components, simplifying the analysis. While $a_i^\star = 1$ is a particular choice of target weights, the training algorithm of the model is agnostic to this choice and we, therefore, obtain sample-complexity expected for a general non-linear feature of the form $\mathbf{a}^{\star\top} P_k(W^\star\mathbf{x})/\sqrt{d^{\varepsilon_1}}$.

• **Information exponent**: We shall indeed require that the information exponent [19, 2, 1] of $g^\star(\cdot)$ is 1 and that of $P_k(\cdot)$ is 2:

**Assumption 1.** *Let $z \sim \mathcal{N}(0,1)$ denote a standard normal variable. We assume that $\mathbb{E}[g^\star(z)z] \neq 0$, $\mathbb{E}[P_k(z)He_2(z)] \neq 0$.*

The condition on $g^\star(\cdot)$ is necessary, as gradient descent (without repetition) has a drastic worst complexity for exponents larger than 2. We expect however, that the condition on $P_k(\cdot)$ can be relaxed to information-exponent $\geq 2$ instead of being exactly 2. The information exponent of $P_k(\cdot)$ being 1 results in linear components that do not require recovery of the full subspace spanned by $W^\star$. Thus setting the information exponent of $P_k(\cdot)$ to 2 simplifies our analysis by avoiding the need for a separate treatment of such linear "spikes".

We further require the activations of the neural net $\sigma(\cdot)$ to be sufficiently expressive and to satisfy certain alignment conditions:

**Assumption 2.** $\sigma : \mathbb{R} \to \mathbb{E}$ *is analytic, non-polynomial with $\sigma'(0) \neq 0$ and there exist constants $L_1, L_2 \in \mathbb{R}^+, m \in \mathbb{N}$ such that $|\sigma(x)| \leq L_1 + L_2|x|^m$. Furthermore, $\sigma(\cdot)$ satisfies: (a) $\mathbb{E}[\sigma(z)He_j(z)] \neq 0$ for all $1 < j \leq k$, (ii) $\mathbb{E}[\sigma(\sigma(z))He_2(z)]\mathbb{E}[P_k(z)He_2(z)] > 0$ and iii) $\mathbb{E}[\sigma(\sigma(z))z] = 0$*

The last two conditions ensure that all neurons in $W_1$ recover spherical projections of $W^\star$. In the absence of the above conditions, we still expect recovery of $W^\star$ but with anisotropy across neurons. Such an anisotropy is expected to complicate the subsequent analysis. We show in Appendix C.7 the existence of a $\sigma(\cdot)$ satisfying the above set of conditions.

The next assumption, however, is only a technical one that arises only because we used $\{a_i^\star = 1\}$. It could be relaxed by taking Gaussian values, or by performing more gradient steps on $W_2$, but this would complicate the proof. We discuss this in detail in App. C.25:

**Assumption 3.** $\mathbb{E}_{z\sim\mathcal{N}(0,1)}[g^\star(z)He_j(z)] = 0$ *for $1 < j \leq k$*

Under the above assumptions, our main result now establishes hierarchical learning for the target of the form (1) by a three-layer network $f^\star(\mathbf{x})$ by first recovering $W^\star$ through the first-layer $W_1$, next recovering $h^\star(\mathbf{x})$ through the second layer pre-activations $\mathbf{h}_2(\mathbf{x}) = W_2\sigma(W_1\mathbf{x})$ and finally fitting $f^\star(\mathbf{x})$ upon training the last layer $\mathbf{w}_3$. The full formal statement of the result is provided in Appendix C.1.

**Theorem 1** (Informal). *Let $f^\star(\mathbf{x})$ be as in Eq. (1) with $\varepsilon_1 \in (0,1)$ and consider a three-layer model:*

$$\hat{f}_\theta(\mathbf{x}) = \mathbf{w}_3^\top\sigma(b_2 + W_2\sigma(W_1\mathbf{x} + b_1)), \tag{17}$$

*with $W_1 \in \mathbb{R}^{p_1 \times d}$, $W_2 \in \mathbb{R}^{p_2 \times p_1}$, $\mathbf{w}_3 \in \mathbb{R}^{p_3}$.*

*Let $\mathcal{L}_c(\theta)$ denote the correlation loss defined as $\mathcal{L}_{cl}(\theta) := -\hat{f}_\theta(\mathbf{x})f^\star(\mathbf{x})$. Under Ass. 1-3, for any $0 < \delta < \delta' < 1$, there exist time-steps $T_1 = \mathcal{O}(\text{polylog} d)$ such that with batch-size $n_1 = \Theta(d^{\varepsilon_1+1+\delta}), n_2 = \Theta(d^{k\varepsilon_1+\delta})$ and $p_2 = p_1 = \Theta(d^{k\varepsilon_1+\delta'})$, the following holds with high probability as $d \to \infty$:*

*(i) $T_1$ steps of neuron-wise spherical SGD on correlation-loss $\mathcal{L}_c(\theta)$ applied to $W_1$ with step-size $\eta = \tilde{\eta}\sqrt{p_2}\sqrt{d^{\varepsilon_1}}$ on independent batches of size $n_1$ results in $W_1$ learning random projections along*

$W^\star$ upto error $o_d(1)$. Concretely, there exists a sequence of random matrices $Z \in \mathbb{R}^{p_1 \times d^{\varepsilon_1}}$ with independent rows sampled uniformly on the unit sphere i.e $z_i \sim U(\mathcal{S}(1))$:

$$W_1 = Z(W^\star) + o(1), \tag{18}$$

as $d \to \infty, \tilde{\eta} \to 0$.

(ii) Subsequently, upon reinitializing $W_2 = \mathbf{0}_{d \times d}$ and $\mathbf{w}_3$ with entries $\mathcal{N}(0,1)$, a single pre-conditioned gradient step on correlation loss $\mathcal{L}_c(\theta)$ with step size $\eta_2 = \Theta(\sqrt{p_2})$ and using an independent size $n_2$ results in learning $h^\star$ upto error $o_d(1)$ with the preactivation $\mathbf{h}_2(\mathbf{x}) = W_2\sigma(W_1\mathbf{x}) \in \mathbb{R}^{p_2}$:

$$\mathbf{h}_2(\mathbf{x}) = c\mathbf{w}_3 h^\star(\mathbf{x}) + o_d(1), \tag{19}$$

where $c \neq 0$ denotes a constant and the $o_d(1)$ error is w.r.t the metric induced by $L_2(\mathcal{N}(\mathbf{0}, I_d))$.

(iii) Upon training $W_1, W_2$ as above, updating $\mathbf{w}_3$ with ridge-regression on $\Theta(d^\delta)$ samples results in approximating $f^\star(\mathbf{x})$ upto error $o_d(1)$ with the 3-layer predictor $\mathbf{w}_3^\top \sigma(W_2\sigma(W_1\mathbf{x}))$.

The details of the initialization projections and pre-conditioning steps are provided in App. C.5. The condition $p_2 = p_1$ is again solely to simplify the analysis and we expect the results to hold for $p_2 = \Theta(d^\delta), p_1 = \Theta(d^{k\varepsilon_1+\delta})$.

Since each row of $W_j^\star$ contains $d$ parameters, the complexity $n_1 \approx \Theta(d^{\varepsilon_1+1})$ matches the total number of parameters in $W_1^\star, \cdots, W_r^\star$, and is therefore expected to be the information-theoretic scaling of the sample-complexity required for the (strong) recovery of $W_1^\star, \cdots, W_r^\star$. Similarly, the complexity $n_2 = \Theta(d^{k\varepsilon_1})$ is the expected minimum sample-complexity required for the strong recovery of a degree-$k$ functions on a $d^{\varepsilon_1}$-dim. space.

**Proof sketch** — We provide the full proof of the above result in App. C, and highlight the most important steps below:

(i) **Composition of Hermite decompositions**: Building upon [84], we use the asymptotic Gaussianity of $h^\star(\mathbf{x})$ to relate the Hermite decomposition of $f^\star(\mathbf{x})$ to the one of $h^\star(\mathbf{x})$.

(ii) **Low-dimensional dynamics for** $W_1$: Using the compositional Hermite-decomposition above, following [19, 10, 1], we show that the evolution of $W_1$ during the training of the first layer can be described through an effective dynamics on the overlaps $W_1(W^\star)^\top$. Unlike the single/multi-index analysis of [19, 10, 1], the diverging dimensionality of $W^\star, W_1$ that appear in our approach, as well as the later use of the updated weights $W_2$, requires a careful control over the error terms. Concretely, we show that the components of $W_1$ along $W^\star$, as well as the error terms, maintain isotropy and hypercontractivity through the dynamics. Moreover, such divergent dimensionality $d^\epsilon$ of $W^\star$ leads to "strong recovery" of $W^\star$ by $W_1$. We refer to Appendix C.8 for details.

(iii) **Function-space decomposition of the** $2^{\text{nd}}$-

Figure 2: **Numerical simulation:** Generalization error versus $\kappa = \log n / \log d$ for $f^\star(\mathbf{x}) = \tanh(3\mathbf{a}^\star \cdot P_3(W^\star\mathbf{x}))/\sqrt{d^{\varepsilon_1=1/2}}$ with different training protocols: **(Top)** kernel ridge regression (orange points) only beats the random performance (purple solid line) starting from $n = d + (d-1)d/2$, and is limited to quadratic approximation (orange line). 2-layer net (green points), instead, starts to learn at $\kappa = 1.5$ (black dashed line) and can beat the quadratic limit (asymptotics is given by the green line). 3-layer net trained with layerwise training (blue markers) not only learn at $\kappa = 1.5$ (vertical line). but also surpasses the best possible 2-layer net error, illustrating the advantage of depth; **(Bottom)** comparison of layerwise training (blue) with joint training (red) of all the layers of a 3-layer net with standard backpropagation.

**layer pre-activations**: Gradient steps on $W_2$ extract statistics in features-space $\sigma(W_1\mathbf{x})$. Similar to [84, 66, 37], we show that these statistics appear in the updates for the pre-activations $h_2(\mathbf{x})$ as projections of a perturbed version of $f^\star$ on the conjugate Kernel defined by the first-layer:

$$\Delta\mathbf{h}_2(\mathbf{x}) \approx c\sigma(W_1\mathbf{x})^\top(\frac{1}{n}\sigma(W_1X^\top)^\top(\sigma(W_1X)^\top)^{-1}\sigma(W_1X)f^\star(X), \tag{20}$$

where $X \in \mathbb{R}^{n \times d}$ denotes the batch of data utilized in a gradient step and $c > 0$ denotes a constant.

(iv) **Concentration of the sample-covariance matrix**: In light of $(iii)$, the recovery of features

in $h_2(\mathbf{x})$ depends on the feature matrix $\sigma(W_1 X)$ being able to approximate and span the relevant functional subspace, which requires both sufficiently many samples and sufficiently many neurons. Building on the matrix-concentration analysis of [58], we show that the projections onto the $\sigma(W_1 X)$ up to degree-$k$ functions can be well approximated as long as $n, p_1 = \Theta(d^{k\varepsilon_1+\delta})$. Low-degree eigenfunctions concentrate faster since they span lower-dimensional subspaces.

**From SIGHT to MIGHT** — While we expect similar results to hold in generality, the theorem is only fully proven for the class of target in eq. (1). While a complete proof for MIGHT is a difficult task, we discuss additional ($r > 1$ and $\ell > 1$) results in this and subsequent paragraphs.

We first remark that part $(i)$ of Theorem 1 ( weak-recovery of $W^\star$), under suitable symmetry assumptions on $g^\star(\cdot)$, holds for arbitrary $r$, and thus for MIGHT functions $f^\star$ (and not only SIGHT ones) (see App. E). Establishing rigorously part $(ii)$ for $r > 1$ involves technical hurdles relating to the control in the Gaussian approximation of $h^\star$. We describe them in App. E.

MIGHT functions are interesting in illustrating the role of the information exponent in Ass. 1. It is easy to design counterexamples, for instance, the parity problem with $y = \text{sign}(h_1^\star h_2^\star h_3^\star)$ violates Ass. 1. We illustrate some of these numerically in App. 5 (See Fig. 10). We believe, however, that with reusing batches, the information exponent could be replaced with the much permissive generative one [32, 52, 11]. SIGHT and MIGHT functions are indeed generalizations of the multi-index functions, and the properties of the latter such as information [19] and generative exponents [28], and the notion of trivial, easy and hard directions [81]) should translate to the former.

**From MIGHT to Deeper MIGHT** — Depth introduces more difficulties for rigorous studies, but our mathematical analysis can be extended for more general constructions. By the tree-like hierarchical construction of features (Eq. (7)) for general depth, the components $\mathbf{h}_\ell^\star(\mathbf{x})$ remain independent and asymptotically Gaussian. Generalizing Thm. 1 for $L \geq 3$ in its full-generality requires however not only an extension of part $(ii)$ of Thm. 1 to $r > 1$, but also a careful control over the non-asymptotic rates for the tails of $\mathbf{h}_\ell^\star(\mathbf{x})$ and the associated kernels.

We instead prove a weaker, but useful, result corresponding to the hierarchical weak recovery of a single non-linear feature at a general level of depth $L \in \mathbb{N}$, under an idealized scenario of perfect spherical recovery of hidden features at level $L - 1$. We refer to App. D for the full formal statement and its proof, which exploits the independence of components of $h_{L-1}^\star(\mathbf{x})$ and the hyper-contractivity of the Gaussian measure:

**Theorem 2.** *For $L \in \mathbb{N}$, let $f^\star(\mathbf{x})$ denote a target as in Eq. (6) with $r = 1$, and let $\delta', \delta$ be arbitrary reals satisfying $0 < \delta < \delta' < 1$. Consider a model of the form $\hat{f}_\theta(\mathbf{x}) = \mathbf{w}_L^\top \sigma(W_{L-1}\sigma(W h_{L-1}^\star(\mathbf{x})))$ with $W \in \mathbb{R}^{p_{L-2} \times d^{\varepsilon_{L-2}}}$ having $p_{L-2} = \Theta(d^{k\varepsilon_\ell - 2 + \delta'})$ rows independently sampled as $\mathbf{w}_i \sim U(\mathbb{S}_{d^{\varepsilon_{L-2}}}(1))$. Under Ass. 1-3, after a single step of pre-conditioned SGD on $W_{L-1}$ with batch-size $\Theta(d^{k\varepsilon_\ell - 2 + \delta})$, step-size $\Theta(\sqrt{p_{L-1}})$, the pre-activations $h_{L-1}(\mathbf{x}) := W_{L-1}\sigma(W h_{L-1}^\star(\mathbf{x}))$ satisfy, for a constant $c > 0$:*

$$h_{L-1}(\mathbf{x}) = c\mathbf{w}_L h_L^\star(\mathbf{x}) + o_d(1), \tag{21}$$

## 4  General Conjecture for Efficient Hierarchical Learning

Building on the above results, we now propose a general structure for hierarchical learning and conjecture its relevance in broader settings, as we briefly alluded to in Section 1.6 under "Hierarchical data models". As highlighted in Assumption 1, our analysis requires the target nonlinearities $g^\star(\cdot)$ and $P_k(\cdot)$ to have low-degree components (in our case Information Exponents $1, 2$ respectively).

More generally, for any such compositional target to be learnable through gradient descent, we conjecture that for every depth level $\ell = 1, \cdots, L$, the intermediate representation $\mathbf{h}_\ell^\star(\mathbf{x})$ retains low-degree correlations with the target $y = f^\star(\mathbf{x})$ or transformations of $y$. Concretely, defining the following require that at every layer $\ell$: $\mathbb{E}\left[f^\star(\mathbf{x})(\mathbf{h}_\ell^\star(\mathbf{x}))^{\otimes k}\right] = \Theta(1)$ for some small $k \in \mathbb{N}$. More formally, one may introduce the *Compositional Information Exponent*:

**Definition 1** (Compositional Information Exponent). *Given a SIGHT or a MIGHT function $f^\star(\mathbf{x})$, we define the compositional information exponent at level $\ell$ as:*

$$\text{CIE}(\ell) = \inf\{k : \left\|\mathbb{E}[(\mathbf{h}_\ell(\mathbf{x}))^{\otimes k}f^\star(\mathbf{x})]\right\|_F = \Theta(1)\} \tag{22}$$

Therefore we conjecture that compositional target learnable through gradient descent have, at every layer $\ell$, low $\text{CIE}(\ell)$. Intuitively, even when the mapping from $(\mathbf{h}_\ell^\star(\mathbf{x}))$ to the final label $y = f^\star(\mathbf{x})$ is highly non-linear, the low-degree correlations with the intermediate representations $\mathbf{h}_\ell^\star(\mathbf{x})$ ensure that it can be recovered through gradient descent. This aligns with the general wisdom

on the structure of real data. The non-trivial correlation of image labels with low-degree features: edges, shapes, and colors, or in the case of text, the correlation between labels and certain word-frequencies or bi-gram counts (see for instance the discussion in [24]). Equivalently, such low-degree dependence can be interpreted as robustness of $y$ with respect to perturbations in $(\mathbf{h}_\ell^\star(\mathbf{x}))$, in which case the condition becomes to mantain a robust compositionality. This hypothesis generalises the classical notion of information exponent—originally formulated for the input layer—to all intermediate representations of a hierarchical model. Consequently, within the SIGHT and MIGHT classes, functions that are efficiently learnable by gradient descent are characterised by the presence of such low-degree alignments at each level. When this condition fails, for example, in parity functions whose first non-vanishing Hermite coefficient lies at large degrees, gradient descent fails to efficiently recover any intermediate features.

Beyond the information exponent, we believe this layer-wise information exponent condition can be replaced by a more generic one, at the price of a more complex analysis. For a start, with repeated passes and data reuse, it is natural to expect that the information exponent can be replaced by the generative exponent, as discussed for two-layer networks in [28, 32]. In such settings, the condition can be generalized to:
$$\left\| \mathbb{E}\left[ \mathcal{T}(y)(\mathbf{h}_\ell^\star(\mathbf{x}))^{\otimes k} \right] \right\|_F = \Theta(1),$$
for some transformation $\mathcal{T} : \mathbb{R} \to \mathbb{R}$ and small $k \in \mathbb{N}$.

Moreover, it may be possible to replace it by the more permissive "staircase" picture once one goes beyond layer-wise training analyses. Making this precise is an exciting, but challenging, direction for future work.

## 5 Numerical Illustrations

While our theorems provide a rigorous control of learning with a particular, well-conditioned, training procedure, we numerically test the validity of our theory towards describing realistic training routines with mini-batch updates, finite (and rather low) dimensional examples, using multi-pass (instead of a single one) for the second layer, etc. For concreteness, we consider $f^\star(\mathbf{x}) = \tanh\left( \frac{3\mathbf{a}^{\star\top} P_3(W^\star \mathbf{x})}{\sqrt{d^{\varepsilon_1 = 1/2}}} \right)$, a similar example as discussed in Section 3 and with, again, a polynomial $P_{k=3}$ with second and third Hermite polynomials. We show simulations in Fig. 2 and discuss here the most salient observations:

**(i)** First we compare the performance of kernel methods with those of a two-layer network. On the one hand, the former method should be able to fit the quadratic part of the target function as soon as $n = O(d^2)$ [60]. This is well observed, with a double descent peak when the number of data hits the number of features in a quadratic kernel, i.e. $n_{\text{peak}} = d(d-1)/2 + d + 1$. On the other hand, two-layer networks are capable of recovering $W^\star$ when $n = O(d^{1.5})$, therefore improving the test performance to quadratic and cubic fit when $\kappa \geq 1.5$.

**(ii)** We then train a three-layer network, with a **layerwise** approach resembling the procedure in Thm 1, where we train every layer in order, (first $W_1$, then $W_2$, etc.). We do not, however, follow the restrictions of the theorem and just perform a standard gradient descent (no reinitializing, no projection, using minibatch, etc.). Not only does the method starts to learn when $n > d^{1.5}$ but **it outperforms the 2-layer baseline** in agreement with Thm. 1.

**(iii)** Lastly, we consider the standard training procedure —refered to as **joint training**– with backpropagation through the network with mini-batch gradient descent. The routine performs similarly to the layerwise approach, illustrating the generality of the dimensionality reduction beyond the assumptions of Thm. 1.

We refer to App. F for details on the numerical implementations, along with the analysis of the quality of the learned representations as a function of the sample complexity (see Fig. 9).

**Conclusion—** We introduced a theoretical framework for understanding the computational advantages of deep neural networks over shallow models when learning high-dimensional hierarchical functions, where depth facilitates a progressive reduction of effective dimensionality. We hope our paper will spark interest in these directions.

**Acknowledgement—** We thank Alex Damian, Jason Lee, Bruno Loureiro, Yue M. Lu, Theodor Misiakiewicz, Eshaan Nichani, Tomaso Poggio, Zhichao Wang, Denny Wu, and Mathieu Wyart for insightful discussions. We acknowledge funding from the Swiss National Science Foundation grants SNSF SMArtNet (grant number 212049), OperaGOST (grant number 200021 200390) and DSGIANGO (grant number 225837).

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

## A  Illustration of SIGHT and MIGHT targets

We exemplify visually SIGHT (1) and MIGHT (4) functions in Fig. 3, and their deep version (6) in Fig. 4. These illustrations clarify how hierarchical compositions operate across layers and how depth progressively compresses the input through structured non-linear transformations. Specifically, they highlight the architectural transition from shallow models to deeper ones, where each layer reduces the effective dimensionality via localized polynomial projections. The tree-like structure of the deep targets, emphasizes the compositional nature of the learning task and motivates the layer-wise training regime analyzed in the main results.

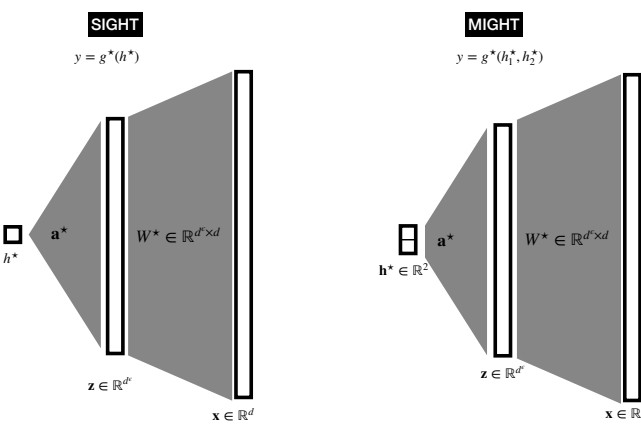

Figure 3: **SIGHT and MIGHT targets:** Illustration of Single and Multi Index Gaussian Hierarchical Targets, i.e., SIGHT in eq. (2) and MIGHT in eq. (4). **Left: A SIGHT function.** Here we first go from $\mathbf{x} \in \mathbb{R}^d$ to $\mathbf{z} \in \mathbb{R}^{d^\varepsilon}$. After applying the polynomial transformation pointwise (not shown), this is projected to create a scalar $h^\star \in \mathbb{R}$. One can then output the label $y = g^\star(h^\star)$. **Right: A MIGHT function.** Again, we go from $\mathbf{x} \in \mathbb{R}^d$ to $\mathbf{z} \in \mathbb{R}^{d^\varepsilon}$. After applying the polynomial transformation pointwise, we finally projecte on two values $h^\star_{4,1}$ and $h^\star_{4,2}$, from which we create $y$ as a two-index function $y = g^\star(h^\star_{4,1}, h^\star_{4,2})$.

## B  Depth Separation

Here, we further discuss the separation in sample complexity for deep versus shallow models complementing the exposition in the main text. Different works in approximation theory have established clear depth-separation results in expressive power. For example, [79] constructed a family of highly oscillatory functions (essentially obtained by iterative compositions of ReLU units) which a network of depth $L$ and constant width can express in contrast to two-layer networks. Similarly, [71] demonstrated a depth separation using simple geometric indicator functions that can be efficiently realized by a network with an extra hidden layer, but cannot be approximated to high accuracy by any two-layer network of polynomial size. It is important to note that "learning" in the context of these results refers to the ability of the architecture to approximate a fixed target function under a given input distribution. In contrast, the present work focuses on representation learning via gradient descent to recover hierarchical feature compositions in Gaussian data. The depth separation we highlight is not purely about static approximation power, but about data-efficient learning through hierarchical dimension reduction. A deep network can progressively extract and refine features across multiple layers, effectively performing stage-wise dimensionality reduction, such that each layer learns a meaningful intermediate representation of the data. Along these lines, [23] addressed under different data models similar questions and found that deep networks trained with gradient descent learn hierarchical features and progressively reduce dimensionality across layers. As highlighted in the main, the advances closest to our framework are due to [84, 66, 37], who established depth separation results between 2-layer and 3-layer networks under simpler

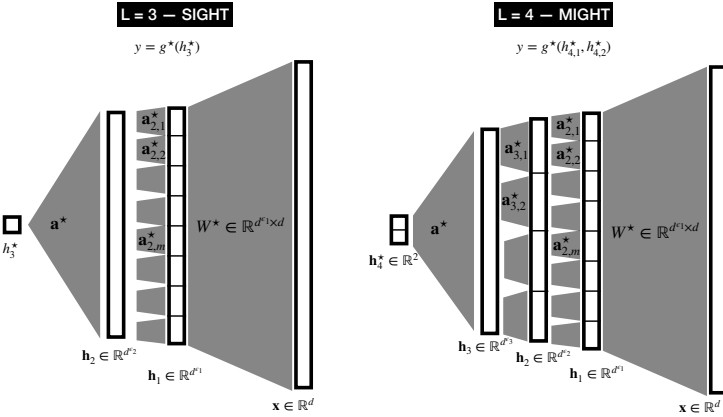

Figure 4: **Deep SIGHT and MIGHT:** Illustration of deep target functions. **Left: A SIGHT function with depth $L = 3$.** Here we first go from $\mathbf{x} \in \mathbb{R}^d$ to $\mathbf{h}_1 \in \mathbb{R}^{d^{\varepsilon_1}}$. After applying the polynomial transformation pointwise (not shown), we now divide $\mathbf{h}_1$ into $d^{\varepsilon_2}$ blocks of sizes $d^{\varepsilon_1 - \varepsilon_2}$. Each of these blocks is projected to create one of the components of $\mathbf{h}_1 \in \mathbb{R}^{d^{\varepsilon_2}}$. After another polynomial transformation (not shown) we finally project to a single value $h_3^{\star}$. We can then output the label $y = g^{\star}(h_3^{\star})$. **Right: A MIGHT function with depth $L = 4$.** Again, we go from $\mathbf{x} \in \mathbb{R}^d$ to $\mathbf{h}_1 \in \mathbb{R}^{d^{\varepsilon_1}}$. After applying the polynomial transformation pointwise (not shown), we now divide $\mathbf{h}_1$ into $d^{\varepsilon_2}$ blocks of sizes $d^{\varepsilon_1 - \varepsilon_2}$. Each of these blocks is projected to create one of the components of $\mathbf{h}_2 \in \mathbb{R}^{d^{\varepsilon_2}}$. We repeat this operation: we further divide $\mathbf{h}_2$ into $d^{\varepsilon_3}$ blocks of sizes $d^{\varepsilon_2 - \varepsilon_3}$ and each of these blocs is projected to create one of the components of $\mathbf{h}_3 \in \mathbb{R}^{d^{\varepsilon_3}}$. After another polynomial transformation (not shown) we finally project on two values $h_{4,1}^{\star}$ and $h_{4,2}^{\star}$ and create $y$ as a two-index function $y = g^{\star}(h_{4,1}^{\star}, h_{4,2}^{\star})$.

training setups, where the first layer remains fixed. In contrast, our analysis strengthens these separations by considering fully-trained architectures without fixed layers.

## B.1 Towards General Two-Layer Networks Lower Bound

In Section 2, we argued why a two-layer network, upon recovering $W^{\star}$ (up to noisy random rotations), is insufficient for learning SIGHT targets (see eq. (13)). Ideally, one would hope to show that such barrier introduced holds for two-layer networks trained through a general gradient-based algorithm. While obtaining unconditional lower-bounds on two-layer networks trained under gradient descent remains a challenging open problem, we briefly comment on why we conjecture our class of SIGHT targets to be hard to learn with polynomial sample complexity. Amongst lower-bounds closest to our class of targets, [33] established that targets of the form $g^{\star}(\mathbf{x}^{\top} A \mathbf{x})$, with $A = U \begin{pmatrix} 0 & I \\ I & 0 \end{pmatrix} U^{\top}$ for some orthogonal matrix $U \in \mathbb{R}^{d \times d}$ cannot be learned under polynomial time and sample-complexity by a two-layer network.

We next discuss how such targets fall under the setup of MIGHT (4). Consider the setting of MIGHT with $r = 2$, $k = 2$, $P_k(z) = z^2 - 1$ and:

$$g^{\star}(h_1^{\star}, h_2^{\star}) = h_1^{\star} - h_2^{\star}. \tag{23}$$

Since $h_j^\star = \frac{1}{\sqrt{d}} \sum_{i=1}^{d^{\epsilon_1}} (\langle \mathbf{w}_{j,i}^\star, \mathbf{x} \rangle)^2 - 1$, we obtain that:

$$h_1^\star - h_2^\star = \frac{1}{\sqrt{d}} \sum_{i=1}^{d^{\epsilon_1}} (\langle \mathbf{w}_{1,i}^\star, \mathbf{x} \rangle)^2 - (\langle \mathbf{w}_{2,i}^\star, \mathbf{x} \rangle)^2$$

$$= \mathbf{x}^\top \Big( \frac{1}{\sqrt{d}} \sum_{i=1}^{d^{\epsilon_1}} \big( \mathbf{w}_{1,i}^\star \mathbf{w}_{1,i}^{\star\top} - \mathbf{w}_{2,i}^\star \mathbf{w}_{2,i}^{\star\top} \big) \Big) \mathbf{x}$$

$$= \mathbf{x}^\top U \begin{pmatrix} 0 & I \\ I & 0 \end{pmatrix} U^\top \mathbf{x},$$

where $U \in \mathbb{R}^{d \times d}$ is an orthogonal matrix whose columns are suitable orthonormal combinations of the vectors $\{\mathbf{w}_{1,i}^\star\}_i \cup \{\mathbf{w}_{2,i}^\star\}_i$. We provide below a short derivation for the mapping.

Let $m = d^{\varepsilon_1}$ and define the block-rotation $R := \frac{1}{\sqrt{2}} \begin{pmatrix} I & I \\ I & -I \end{pmatrix}$, which satisfies $R^\top \begin{pmatrix} I & 0 \\ 0 & -I \end{pmatrix} R = \begin{pmatrix} 0 & I \\ I & 0 \end{pmatrix}$. Let $U_0 \in \mathbb{R}^{d \times 2m}$ collect the $2m$ orthonormal vectors $\{\mathbf{w}_{1,i}^\star\} \cup \{\mathbf{w}_{2,i}^\star\}$, and complete it with an orthogonal complement $U_\perp$ to form $\tilde{U} = [\, U_0 \mid U_\perp \,] \in \mathbb{R}^{d \times d}$. Let $U = \tilde{U} \operatorname{diag}(R, I_{d-2m})$, which is orthogonal. Then:

$$\frac{1}{\sqrt{d}} (h_1^\star(\mathbf{x}) - h_2^\star(\mathbf{x})) = \mathbf{x}^\top \Big( \frac{1}{\sqrt{d}} \sum_{i=1}^{m} (\mathbf{w}_{1,i}^\star \mathbf{w}_{1,i}^{\star\top} - \mathbf{w}_{2,i}^\star \mathbf{w}_{2,i}^{\star\top}) \Big) \mathbf{x} = \mathbf{x}^\top U \begin{pmatrix} 0 & I \\ I & 0 \end{pmatrix} U^\top \mathbf{x}.$$

This is exactly of the form $g^\star(\mathbf{x}^\top A \mathbf{x})$ with $A = \tilde{U} \begin{pmatrix} 0 & I \\ I & 0 \end{pmatrix} \tilde{U}^\top$.

## C   Proofs of the main Results

### C.1   Full Statement of Theorem 1

**Theorem 3.** *Let $f^\star(\mathbf{x})$ be as in Eq. (1) with $\varepsilon_1 \in (0,1)$ and consider a three-layer model:*

$$\hat{f}_\theta(\mathbf{x}) = \mathbf{w}_3^\top \sigma(b_2 + W_2 \sigma(W_1 \mathbf{x} + b_1)), \tag{24}$$

*with $W_1 \in \mathbb{R}^{p_1 \times d}, W_2 \in \mathbb{R}^{p_2 \times p_1}, \mathbf{w}_3 \in \mathbb{R}^{p_3}$.*

*Let $\mathcal{L}_c(\theta)$ denote the correlation loss defined as $\mathcal{L}_{cl}(\theta) := -\hat{f}_\theta(\mathbf{x}) f^\star(\mathbf{x})$. Under Ass. 1-3, for any $0 < \delta < \delta' < 1$, with batch-size $n_1 = \Theta(d^{\varepsilon_1 + 1 + \delta}), n_2 = \Theta(d^{k\varepsilon_1 + \delta})$ and $p_2 = p_1 = \Theta(d^{k\varepsilon_1 + \delta'})$, the following holds with high probability as $d \to \infty$:*

***Recovery by layer*** 1: *For each $i \in [p_1]$, let $\mathbf{w}_i^1$ denote the $i_{th}$ neuron of $W_1$. Suppose that $\mathbf{w}_i^1$ is updated as in Algorithm C.5 through spherical SGD on correlation loss $\mathcal{L}_c(\theta)$, using step size $\eta = \tilde{\eta}\sqrt{d^{\varepsilon_1} p_2}$. Let the value at the $t_{th}$ iterate be denoted by $\mathbf{w}^{1,t}$, i.e.:*

$$\tilde{\mathbf{w}}_i^{1,t} = \mathbf{w}^{1,t} + (\mathbf{I}_d - (\mathbf{w}_i^{1,t})(\mathbf{w}_i^{1,t})^\top) \eta \frac{1}{n} \sum_{i=1}^{n} f^\star(\mathbf{x}_i) \tilde{\sigma}'(\langle \mathbf{w}^{1,t}, \mathbf{x}_i \rangle) \mathbf{x}_i$$

$$\mathbf{w}^{1,t+1} = \frac{1}{\|\tilde{\mathbf{w}}^{1,t}\|_2} \tilde{\mathbf{w}}^{1,t} \tag{25}$$

*Let $P_{W^\star} \in \mathbb{R}^{d \times d}$ denote the projection operator onto the subspace spanned by $W^\star$ and define:*

$$\mathbf{u}_i^\star = \frac{P_{W^\star} \mathbf{w}_i^{1,0}}{\left\| P_{W^\star} \mathbf{w}_i^{1,0} \right\|} \tag{26}$$

*Let $\tau_i$ denote the stopping times defined by:*

$$\tau_\kappa^i := \{\inf t : |\langle \mathbf{u}_i^\star, \mathbf{w}_i^{1,t} \rangle| \geq \kappa\}. \tag{27}$$

*then, for any $\kappa < 1, \exists$ a constant $C_\kappa > 0$ and $\tilde{\eta} > 0$ such that w.h.p as $d \to \infty$:*

  •

$$\max_i \tau_\kappa^i \leq C_\kappa \log d \tag{28}$$

- $\forall i \in [p_1]$:
$$\mathbf{w}_i^{1,\tau_\kappa} = \kappa^+ \mathbf{u}_i^\star + \sqrt{1-\kappa^+}\mathbf{u}_i + o_d(1), \tag{29}$$
where $\kappa^+ > 0$ and $\mathbf{u}_i \sim U(\mathcal{S}^{d-1}(1))$.

**Recovery by layer** 2: *Suppose that $W_2$ is re-initialized to $W_2 = \mathbf{O}_{d \times d}$ while $\mathbf{w}_3$ is re-initialized with entries drawn from $\mathcal{N}(0,1)$. Let $Z = \sigma(X(W_1)^\top) \in \mathbb{R}^{n_2 \times p_2}$ and consider a single pre-conditioned update of the form:*
$$W_2 \leftarrow \left(\frac{1}{n}Z^\top Z + \lambda_2\right)^{-1} \nabla_{W_2}\mathcal{L}_c. \tag{30}$$

*There exists $\lambda_2 \in \mathbb{R}^+$ with $\lambda_2 = \Theta(\frac{N}{n})$ and step size $\eta_2 = \Theta(\sqrt{p_2})$ such that the pre-activation $\mathbf{h}_2(\mathbf{x}) = W_2\sigma(W_1\mathbf{x}) \in \mathbb{R}^{p_2}$ satisfies:*
$$\mathbf{h}_2(\mathbf{x}) = c\mathbf{w}_3 h^\star(\mathbf{x}) + \mathcal{O}_\prec\left(\frac{1}{\sqrt{d^{\min(\delta,\delta'-\delta)}}}\right). \tag{31}$$

**Remark**: *Here, we introduced the regularization parameter $\lambda_2 > 0$ since we assume that $p_1 \gg n_2$ (overparameterized setting) and thus $Z^\top Z$ is singular. Alternatively, one could consider the underparameterized setting $p_1 \ll n_2$ without the need for an additional regularization.*

**Recovery by layer-3**: *Let $X \in \mathbb{R}^{n_3 \times d}$ denote a matrix with rows containing $n_3$ independent samples from the input distribution $\mathcal{N}(0,\mathbf{I}_d)$. Let $H \in \mathbb{R}^{n_3 \times p_3}$ denote the corresponding pre-activation matrix with rows $\{W_2\sigma(W_1\mathbf{x})) - f^\star(\mathbf{x}_i), \text{ for } i \in [n_3]\}$. For $n_3 = \Theta(d^\delta)$, $\exists \lambda > 0$ such that the ridge-regression predictor $\hat{\mathbf{w}}_\lambda$ given by:*
$$\hat{\mathbf{w}}_\lambda = (\frac{1}{n}H^\top H + \lambda I)^{-1}H^\top f(X), \tag{32}$$

*satisfies:*
$$\mathbb{E}_{\mathbf{x} \sim \mathcal{N}(0,I)}\left[\left\|\hat{\mathbf{w}}_\lambda^\top \sigma(W_2\sigma(W_1\mathbf{x})) - f^\star(\mathbf{x})\right\|_2^2\right] = o_d(1). \tag{33}$$

## C.2 Proof Sketch

We prove each of the three parts of Theorem 1 in succession. We outline the proof for each of these parts below:

**Part $(i)$**:

  (i) The asymptotic composition of Hermite polynomials allows us to decompose the Hermite decomposition of $f^\star(\mathbf{x})$ along Hermite polynomials applied to $W^\star\mathbf{x}$.

  (ii) The leading order term in the Hermite-decomposition $f^\star(\mathbf{x})$ lies along $\mathrm{He}_2(W^\star\mathbf{x})$, which contributes a linear drift to the dynamics of each neuron in $W_1$, with the direction of the drift for neuron $i$ given by $\mathbf{u}_i^\star = W^\star(W^\star)^\top\mathbf{w}_i^{1,0}$, i.e, the initial direction of $\mathbf{w}_i^1$ projection onto $W^\star$.

  (iii) We show that the neuron $\mathbf{w}_i^1$ remains approximately isotropic w.r.t the rows of $W^\star$.

  (iv) Under the above isotropy and due to $d^{\varepsilon_1} \gg 1$, we show that the above linear term dominates throughput the weak-recovery and subsequent states of the dynamics.

  (v) As a consequence, each neuron in $W_1$ evolves primarily along $\mathbf{u}_i^\star$, with the noise controlled through the choice of batch-size. A stopping-time based analsyis then yields $\mathbf{w}_i^{(t)} \to \mathbf{u}_i^\star$.

  (vi) For subsquent use in part $(ii)$ however, we require finer control over the distribution of $\mathbf{w}_i^1$ and its residual terms.

  (vii) Inductively, we show that the distribution of $\mathbf{w}_i^1$ conditioned on a suitable stopping-time is approximately uniform on the unit sphere along $W^\star$ and maintains hypercontractivity.

**Part $(ii)$**:

  (i) Through results established in Part $(i)$, we show that the distribution of the updated weights $W_1$ approximately maintains hypercontractivity for the eigenfunctions of the random-features Kernel associated to the features $\sigma(XW_1^\top)$. This ensures the concentration of the associated sample covariances.

(ii) Upon establishing concentration and spherical approximation along the subspace corresponding to $W^\star$, through an analysis similar to [58], we show that the feature matrix $Z = \sigma(XW^\top)$ contains $\Theta(d^k)$ spikes with diverging eigenvalues and an isotropic bulk with eigenvalues $O(1)$.

(iii) Under $n, p_2 \gg \Theta(d^{k\varepsilon_1})$, we show that these spikes suffice for the pre-conditioned update

$$-\eta(\frac{1}{n}\sigma(W_1 X^\top)^\top(\sigma(W_1 X)^\top)^{-1}\nabla_{W_2}\mathcal{L},$$

to approximate $f^\star(x)$ upto degree $k$-components. As a result, we obtain the recovery of $h^\star(\mathbf{x})$ through $h^2(\mathbf{x})$

**Part** $(iii)$: Finally, fitting the target $f^\star(\mathbf{x})$ upon training $\mathbf{w}_3$ follows through universality of the random features Kernel associated with $\sigma(\cdot)$ and perturbation of the Kernel regression operators.

### C.3  Preliminaries

#### C.3.1  Stochastic Domination

Throughout the analysis, much of our probabilistic error bounds will take the following form, which are standard for functions of random variables with finite Orlicz-norm such as sub-Gaussian/sub-Exponential random variables:

$$\Pr\left[|X|_d \geq C\frac{(\log d)^k}{d^m}\right] \leq e^{-c(\log d)^m}, \tag{34}$$

for some constants $m > 1, k > 0, c > 0$. A slightly weaker form of the bound takes the form:

$$\Pr\left[|X|_d \geq C\frac{1}{d^{m-\delta}}\right] \leq \frac{1}{d^k}, \tag{35}$$

for any $\delta > 0$ and $k \in \mathbb{N}$. To concisely represent such bounds, we use the following notation:
**Definition 2.** *[Stochastic dominance [54]] We say that a sequence of real or complex random variables $X_d$ in a normed space is stochastically dominated by another sequence $Y_d$ in the same space if for all $\varepsilon > 0$ and $k$, the following holds for large enough $d$:*

$$\Pr[\|X\|_d > d^\varepsilon\|Y\|_d] \leq d^{-k}. \tag{36}$$

*We denote the above relation through the following notation:*

$$X = \mathcal{O}_\prec(Y). \tag{37}$$

*Through a union bound, we obtain that $\mathcal{O}_\prec$ is closed under addition, multiplication, i.e $X_1 = O_\prec(Y_1)$ and $X_2 = O_\prec(Y_2)$ imply that:*

$$X_1 + X_2 = O_\prec(Y_1 + Y_2), \tag{38}$$

*and:*

$$X_1 X_2 = O_\prec(Y_1 Y_2), \tag{39}$$

*Furthermore, due to the flexibility of setting an arbitrarily large $k$ in Eq. (36), we observe that stochastic dominance is closed under unions of polynomially many events in $d$.*

*We will often exploit this while taking unions over $p = \mathcal{O}(d)$ neurons and $n = \mathcal{O}(d)$ samples. Furthermore, $\prec$ absorbs polylogarithmic factors i.e:*

$$X = \mathcal{O}_\prec(Y) \implies X = \mathcal{O}_\prec(\text{polylog}(d)Y), \tag{40}$$

*subsumes exponential tail bounds of the form:*

$$\Pr[X_d > tY_d] \leq e^{-t^\alpha}, \tag{41}$$

*for some $\alpha > 0$, as well as polynomial tails of arbitrarily large degree:*

$$\Pr[X_d > tY_d] \leq \frac{C_k}{t^k}, \tag{42}$$

*for some sequence of constants $C_k$ dependent on $k$.*

The above bounds directly translate to the following control over moments:

**Proposition 1.** *Let $X_d, Y_d$ denote two sequences of random variables with:*

$$X = \mathcal{O}_{\prec}(Y), \tag{43}$$

*then for any $q \in \mathbb{N}$ and $\delta > 0$:*

$$\mathbb{E}\left[\|X\|^p\right]^{1/p} \leq d^\delta \mathbb{E}\left[\|Y\|^p\right]^{1/p} \tag{44}$$

**Proposition 2.** *The above proposition follows directly through the following decomposition:*

$$\mathbb{E}\left[\|Y\|^p\right]^{1/p} = \mathbb{E}\left[\|Y\|^p \, \mathbf{1}_{\|X\| \leq d^\delta \|Y\|}\right]^{1/p} + \mathbb{E}\left[\|Y\|^p \, \mathbf{1}_{\|X\| > d^\delta \|Y\|}\right]^{1/p}, \tag{45}$$

*where $\mathbf{1}$ denotes the indicator function. Using the property $\mathbb{E}[Z] = \int_{s=0}^{\infty} \Pr[Z > s] ds$, the second term is bounded by $\frac{1}{d^k}$ for any $k$ and large enough $d$.*

**Asymptotic notation:** In light of the above proposition, throughout the subsequent sections, we use the notation $\tilde{\mathcal{O}}$ to denote deterministic asymptotic bounds upto factors $d^\delta$ for arbitrarily small $\delta > 0$ i.e:

$$f(d) = \tilde{\mathcal{O}}(g(d)), \tag{46}$$

if for any $\delta > 0$, $f(d) \leq d^\delta g(d)$ for large enough $d$

Through a standard application of the Lindeberg exchange technique [26, 82], we further have the following useful estimate:

**Lemma 1** (Non-asymptotic CLT -bound)**.** *Let $X_1, \ldots, X_n \in \mathbb{R}$ be $n$ i.i.d random variables satisfying $X_i = O_{\prec}(1)$. Then, for any function $q : \mathbb{R} \to \mathbb{R}$ with $q \in \mathcal{C}^3(\mathbb{R})$, $\|q'''\|_{\infty} < \infty$ and any $\delta > 0$:*

$$\left|\mathbb{E}\left[q\left(\frac{1}{\sqrt{d}}\left(\sum_{i=1}^{d} X_i\right)\right)\right] - \mathbb{E}_{z \sim \mathcal{N}(0,1)}\left[q(z)\right]\right| \leq c_1 \sqrt{d} |\mathbb{E}\left[X\right]| + c_2 |\mathbb{E}\left[X\right]_1^2 - 1| + \frac{c_3}{d^{1/2-\delta}} \mathbb{E}\left[|X|^3\right], \tag{47}$$

*where $c_1, c_2$ denote constants dependent only on $q$.*

Through standard truncation arguments over the tail of $\frac{1}{\sqrt{d}}(\sum_{i=1}^{d} X_i)$, the above bound extends to all polynomials $\mathbb{R} \to \mathbb{R}$ of finite degree.

### C.3.2 Orthogonal Polynomials and Spherical Harmonics

**Hermite Polynomials**: A key role in our analysis is played by the decomposition of square integrable function with respect to the Gaussian measure in terms of the Hermite polynomials [44].

**Definition 3** (Hermite decomposition)**.** *Let $f : \mathbb{R}^m \to \mathbb{R}$ be a function that is square integrable w.r.t the Gaussian measure. There exists a family of tensors $(C_j(f))_{j \in \mathbb{N}}$ such that $C_j(f)$ is of order $j$ and for all $\mathbf{x} \in \mathbb{R}^m$,*

$$f(\mathbf{x}) = \sum_{j \in \mathbb{N}} \langle C_j(f), \mathcal{H}_j(\mathbf{x}) \rangle \tag{48}$$

*where $\mathcal{H}_j(\mathbf{x})$ is the $j$-th order Hermite tensor [44].*

**Gegenbauer and Associated Laguerre polynomials** Let $\mathbf{w} \sim U(\mathcal{S}^{d-1}(\sqrt{d}))$ denote a random variable distributed uniformly on the sphere in $\mathbb{R}^d$ of radiues $\sqrt{d}$. Let $\mu_d$ denote the associated push-forward measure of the projection $\sqrt{d}\langle \mathbf{w}, \mathbf{e}_1 \rangle$. The Gegenbauer polynomials $Q_\ell^d(\cdot)$ [39] for $\ell \in \mathbb{N}$ form an orthonormal basis w.r.t $L^2(\mu_d)$ with $Q_\ell^d(\cdot)$ being a polynomial of degree $\ell$. Therefore, for any $f \in L^2(\mu_d)$ and $v \in \mathbb{R}^d$ with $\|v\| = 1$, the following decomposition exists:

$$f(\sqrt{d}\langle \mathbf{v}, \mathbf{w} \rangle) = \sum_{k=0}^{\infty} \nu_{d,k} Q_k^d(\sqrt{d}\langle \mathbf{v}, \mathbf{w} \rangle) \tag{49}$$

Next, suppose that $\mathbf{x} \sim \mathcal{N}(\mathbf{0}, \mathbf{I}_d)$. Let $\tau_d$ denote the associated pushforward measure of $\|\mathbf{x}\|^2$. Then, the associated Laguerre polynomials $l_k^d(\cdot)$ form an orthonormal basis w.r.t $\tau_d$ [9].

**Spherical Harmonics** Recall that any inner-product Kernel can be diagonalized w.r.t $L_2(U(\mathcal{S}^{d-1}(\sqrt{d})))$ along the basis of spherical Harmonics $\{Y_{\ell,k}\}_{\ell \in [B(d,k)], k \in \mathbb{N}}$, where $B(d,k)$ denotes the number of spherical harmonics of degree $k$, satisfying $B(d,k) = \Theta(d^k)$:

$$K(\mathbf{x}, \mathbf{x}') = \sum_{k=0}^{\infty} \lambda_k \sum_{l=1}^{n_k} Y_{l,k}(\mathbf{x}) Y_{l,k}(\mathbf{x}'), \tag{50}$$

where $\lambda_k$ denotes the eigenvalue of $K$ w.r.t the $k$-degree spherical harmonics $Y_{l,k}(\mathbf{x})$. [40].

The Spherical Harmonics are related to the Gegenbauer polynomials through the following identity:

**Proposition 3.** *For any* $\mathbf{w}_1, \mathbf{w}_2 \sim U(\mathbb{S}^{d-1}(\sqrt{d}))$*:*

$$Q_k^d(\mathbf{w}_1, \mathbf{w}_2) = \frac{1}{B(d,k)} \sum_{\ell=1}^{B(d,k)} Y_{\ell,k}(\mathbf{w}_1) Y_{\ell,k}(\mathbf{w}_2) \tag{51}$$

We next recall that the Gaussian measure $\mathcal{N}(\mathbf{0}, \mathbf{I}_d)$ admits the following tensor product decomposition:

$$\mathcal{N}(\mathbf{0}, \mathbf{I}_d) = \chi^2(\|\mathbf{x}\|^2) \otimes U(\mathbb{S}^{d-1}(\sqrt{d})), \tag{52}$$

where $U(\mathbb{S}^{d-1}(\sqrt{d}))$ denotes the uniform measure on sphere of radius $\sqrt{d}$

The above tensor product decomposition naturally relates the Hermite orthonormal basis w.r.t the Gaussian measure against the product of radial functions and Gegenbauer polynomials. In particular, we have the following relation:

**Proposition 4.** *For any* $k \in \mathbb{N}$*, the* $k_{th}$*-degree Hermite polynomial lies in the subspace spanned by functions of the form:*

$$f(\frac{\|\mathbf{x}\|^2 - 1}{\sqrt{d^{\varepsilon_1}}}) Y_{\ell,j}(\sqrt{d}\mathbf{x}/\|\mathbf{x}\|), \tag{53}$$

*with* $0 < j \le k$*.*

*Proof.* Recall that $Y_{\ell,j}(\sqrt{d}\mathbf{x}/\|\mathbf{x}\|)$ are homogenous polynomials of degree $j$. Upon restriction to the sphere of radius $\|x\|$, $\mathrm{He}_k(\mathbf{x})$ is a polynomial of degree at-most $k$. Therefore, by Fubini's theorem, we obtain:

$$\mathbb{E}\left[f(\frac{\|\mathbf{x}\|^2 - 1}{\sqrt{d^{\varepsilon_1}}}) Y_{\ell,j}(\sqrt{d}\mathbf{x}/\|x\|) \mathrm{He}_k(\mathbf{x})\right] = 0, \tag{54}$$

for $j > k$. $\qquad \square$

**Proposition 5.** *For any* $k > 2$ *and polynomial* $q(x)$*:*

$$\mathbb{E}\left[\frac{\frac{1}{\sqrt{d^{\varepsilon_1}}} n \sum_{i=1}^{\sqrt{d^{\varepsilon_1}}} \langle \mathbf{w}_i^\star, \mathbf{x} \rangle^2 - 1}{\sqrt{d^{\varepsilon_1}}} q(\frac{1}{\sqrt{d^{\varepsilon_1}}} n \sum_{i=1}^{\sqrt{d^{\varepsilon_1}}} \mathrm{He}_k(\langle \mathbf{w}_i^\star, \mathbf{x} \rangle))\right] = \mathcal{O}(\frac{1}{\sqrt{d^{\varepsilon_1}}}) \tag{55}$$

*Proof.* The above is a direct consequence of Lemma 1 applied to the random variables $(\mathrm{He}_2(\langle \mathbf{w}_i^\star, \mathbf{x} \rangle), \mathrm{He}_k(\langle \mathbf{w}_i^\star, \mathbf{x} \rangle)) \in \mathbb{R}^2$, whose higher-moments are bounded by Gaussian hypercontractivity (Lemma 5). $\qquad \square$

We utilize Gegenbauer polynomials and spherical Harmonics primarily due to the absence of results on eigenvectors of inner-product Kernel matrices under polynomial scalings. This is also the primary bottleneck towards the extension of our theory to multiple layers. Essentially, our analysis relies on showing the concentration of the sample-covariance matrix to the population covariance matrix along the degree-$k$ components.

### C.3.3 Spectral Norm of a tensor

**Definition 4.** *For a symmetric positive-definite tensor* $T \in \mathbb{R}^{d \otimes k}$ *of order* $k$*, we define the spectral norm of* $T$ *as follows:*

$$\|T\|_2 = \sup_{x \in \mathbb{R}^d, \|x\|=1} |\langle x^{\otimes k}, T \rangle| \tag{56}$$

### C.3.4 Hermite-tensors and Gaussian-inner Products

We denote by $\text{He}_k$ for $k \in \mathbb{N}$ the normalized Hermite-polynomials forming an orthonormal basis w.r.t $L^2(\gamma)$. For any $f \in L^2(\gamma)$, we have:

$$f(z) = \sum_{k=0}^{\infty} \mu_k \text{He}_k(z). \tag{57}$$

The Hermite tensors result in the following generalization of the above decomposition:

**Proposition 6.** *Let $\gamma_m$ denote the $m$-dimensional Gaussian measure. For any $f, g : \mathbb{R}^m \to \mathbb{R} \in \ell^2(\mathbb{R}^m, \gamma_m)$, let $C_k(f)$ denote the $k_{th}$-order Hermite-tensor, defined as:*

$$C_k(f) := \mathbb{E}_{z \sim \gamma_m} \left[ f(\mathbf{z}) \, \text{He}_k(\mathbf{z}) \right] = \mathbb{E}_{z \sim \gamma_m} \left[ \nabla^k f(z) \right], \tag{58}$$

*where $\text{He}_k(\mathbf{z})$ denotes the $k_{th}$-order Hermite tensor on $\mathbb{R}^m$. Then:*

$$\langle f, g \rangle_\gamma = \sum_{k \in \mathbb{N}} \langle C_k(f), C_k(g) \rangle. \tag{59}$$

### C.3.5 Compact Self-Adjoint Operators

We collect here the following well-known properties of bounded linear operators on a Hilbert space $L^2(\mu)$ [16]:

**Proposition 7.** *Let $A : L^2(\mu, \Omega) \to L^2(\mu, \Omega)$ denote a bounded-linear operator on a hilbert space $L^2(\mu)$. Then:*

  (i) *If $A$ is compact, self-adjoint then $A$ can be diagonalized along a countable-basis of eigenvectors.*

  (ii) *Suppose that $\mu$ is $\sigma$-finite, then any integral operator $I(x,y) : \omega \times \omega \to \mathbb{R}$ with $\|I(x,y)\|_{L^2(\mu) \times L^2(\mu)} < \infty$ is compact*

  (iii) *For a symmetric integral operator $\|I(x,y)\|_{L^2(\mu) \times L^2(\mu)} < \infty$:*

$$\int I(x,y) d\mu(\omega \times \omega) = \sum_k \lambda_k, \tag{60}$$

  *where $\{\lambda_k\}$ denote the eigenvalues associated with $I(\cdot, \cdot)$.*

### C.3.6 Concentration in Orlicz-spaces

**Definition 5.** *For any $\alpha \in \mathbb{R}$, define $\psi_\alpha(x) = e^{x^\alpha} - 1$. The* Orlicz norm *for a real random variable $X$; $\|X\|_{\psi_\alpha}$ is defined as*

$$\|X\|_{\psi_\alpha} = \inf \left\{ t > 0 \; : \; \mathbb{E}\left[ \psi_\alpha \left( \frac{|X|}{t} \right) \right] \leq 1 \right\} \tag{61}$$

Random variables exhibiting suitable bounds on orlicz norms of finite-order exhibit the following concentration inequality:

**Theorem 4** (Theorem 6.2.3 in [50]). *Let $X_1, \ldots, X_n$ be $n$ independent random variables with zero mean and second moment $\mathbb{E}X_i^2 = \sigma_i^2$. Then,*

$$\left\| \sum_{i=1}^n X_i \right\|_{\psi_\alpha} \leq K_\alpha \log(n)^{1/\alpha} \left( \sqrt{\sum_{i=1}^n \sigma_i^2} + \max_i \|X_i\|_{\psi_\alpha} \right) \tag{62}$$

### C.4 Useful Preliminary Results

A central result underlying our analysis for part $(ii)$ of Theorem 1, based on [60] is the following matrix-concentration bound for matrices with independent heavy-tailed rows:

**Lemma 2** (Theorem 5.48 in [83]). *Let $A \in \mathbb{R}^{n \times p}$ be a random matrix with independent rows $a_i \in \mathbb{R}^p$ with covariance $\mathbb{E}\left[ a_i a_i^\top \right] = \Sigma_a$ and $\mathbb{E}\left[ \max_{i \leq n} \|a\|_i^2 \right] \leq m$. Then:*

$$\mathbb{E}\left[ \left\| \frac{1}{n} A^\top A - \Sigma_a \right\|_2 \right] \leq \max(\|\Sigma_a\|_2^{1/2} \delta, \delta^2), \tag{63}$$

*where $\delta = C\sqrt{m \frac{\log(\min(n,p))}{n}}$.*

**Lemma 3** (Weyl's inequality). *For any $A, B \in \mathbb{R}^{m \times n}$, for all $i \in \mathbb{N}$ with $i \leq \min(m, n)$:*

$$|\sigma_i(A) - \sigma_i(B)| \leq \|A - B\| \tag{64}$$

**Lemma 4** (Resolvent Identity). *Let, $A, B \in \mathbb{R}^{p \times p}$ be two invertible matrices, then:*

$$A^{-1} - B^{-1} = A^{-1}(B - A)B^{-1}. \tag{65}$$

Our next central tool that will be utilized frequently throughout our analysis is the hypercontractivity w.r.t the Gaussian measure:

**Lemma 5** (Gaussian Hypercontractivity, Proposition 5.48. in [15]). *For any polynomial $q : \mathbb{R}^d \to \mathbb{R}$ of degree $k$ and any $p \in \mathbb{N}, p \geq 2$:*

$$\|q(z)\|_{p,\gamma^d} \leq (p-1)^k \|q(z)\|_{2,\gamma^d}, \tag{66}$$

*where $\gamma^d$ denotes the standard Gaussian measure on $\mathbb{R}^d$ and $\|q(z)\|_{p,\gamma}$ denotes the $p$-norm:*

$$\|q(z)\|_{p,\gamma} := \mathbb{E}_{z \sim \gamma}\left[|q(z)|^p\right]^{\frac{1}{p}} \tag{67}$$

**Proposition 8.** *Let $\mathbf{z} \sim \mathcal{N}(0, \mathbf{I}_d)$ denote a $d$-dimensional Gaussian vectors. Suppose that $X_1, \cdots, X_k$ denote i.i.d random variables obtained by applying a fixed polynomial of degree $p \in \mathbb{N}$ to distinct subsets of coordinates of $\mathbf{z}$. Then:*

$$\frac{1}{\sqrt{k}}\left(\sum_{i=1}^{k} X_i\right) = \mathcal{O}_{\prec}\left(\sqrt{|\mathbb{E}\left[X\right]^2| + k|\mathbb{E}\left[X\right]|^2}\right). \tag{68}$$

*Proof.* Since $q(z) = \frac{1}{\sqrt{k}}\left(\sum_{i=1}^{k} X_i\right)$ is a polynomial in $z$ with finite degree $p$, Lemma 5 implies that its higher-order moments are bounded as $\|q(x)\|_p \leq C_p \|q(x)\|_2$. The result then follows by noting that:

$$\mathbb{E}\left[q(z)^2\right]^{1/2} = \sqrt{\mathbb{E}\left[X\right]^2 + 2(k-1)|\mathbb{E}\left[X\right]|^2} \tag{69}$$

$\square$

**Lemma 6** (Discrete Gronwall). *Let $a_t, b_t, c_1^t, c_2^t$ be non-negative sequences satisfying:*

$$a_{t+1} \geq a_t + c_1^t a_t + c_2^t b_t, \tag{70}$$

*then, for any $t \in \mathbb{N}$:*

$$a_{t+1} \geq \prod_{s=1}^{t}(1 + c_1^s)a_0 + \sum_{j=1}^{t-1}\prod_{s=1}^{j}(1 + c_1^s)c_2 b_j \tag{71}$$

*We analogously have the corresponding upper bound i.e*

$$a_{t+1} \leq a_t + c_1^t a_t + c_2^t b_t, \tag{72}$$

*implies that:*

$$a_{t+1} \leq \prod_{s=1}^{t}(1 + c_1^s)a_0 + \sum_{j=1}^{t-1}\prod_{s=1}^{j}(1 + c_1^s)c_2 b_j \tag{73}$$

## C.5 Full Algorithm

We describe the full algorithmic routine used in Theorem 1 in Algorithm 1.

## C.6 Leveraging Asymptotic Gaussianity

A crucial property of the non-linear feature $h^\star(\mathbf{x})$ that we leverage is its asymptotic Gaussianity, not only w.r.t their marginals but w.r.t propagation to the lower-level features. Specifically, building on [84], we show that the high Hermite-degree functions of $h_m^\star(\mathbf{x})$ do not propagate projections along low Hermite-degree functions of $W^\star \mathbf{x}$. To show this, we provide an inductive proof inspired by the combinatorial approach developed in [84], wherein the (entropically) dominant contributions in $(h_m^\star(\mathbf{x}))^k$ arise from terms having the lowest degrees in $\text{He}_j(\langle \mathbf{w}^\star, \mathbf{x} \rangle)$.

---

**Algorithm 1** Layer-Wise Training for a Three-Layer Network

---

**Input:** Training data $\mathcal{D}$, mini-batch sizes $n_1, n_2$, learning rates $\eta_1, \eta_2$, ridge regularization $\lambda$, iteration steps $T_1$.

**Initialize:**
$\quad W_i^{(1)} \stackrel{\text{i.i.d}}{\sim} U(\mathcal{S}^{d-1}(1))$ for $i \in [p_1]$
$\quad W^{(2)} \leftarrow \mathbf{I}_{p_2 \times d}$
$\quad (b^{(1)}, b^{(2)}) \leftarrow \mathbf{0}_{p_1 \times p_2}$.

**Layer 1 updates (correlation loss with spherical projections):**
$\hat{f}(\mathbf{x}) := \hat{f}_\theta(\mathbf{x}) = \mathbf{w}_3^\top \sigma(W_2 \sigma(W_1 \mathbf{x}))$
$\mathcal{L} : \mathbb{R}^d \times \mathbb{R} \to \mathbb{R} \leftarrow \mathcal{L}(\mathbf{x}, y) := -y\hat{f}(\mathbf{x})$ **(Set loss $\mathcal{L}$ to correlation loss)**
**for** $t = 1$ to $T_1$ **do**
$\quad$ Sample mini-batch $X, \mathbf{y} \subset \mathcal{D}$ of size $n_1$
$\quad$ **For each neuron $j$ in layer 1:**
$\quad\quad \tilde{W_j^{(1)}} \leftarrow W_j^{(1)} - \eta_1 \nabla_{W_j^{(1)}} \mathcal{L}(X, y)(\mathbf{I}_d - (W_j^{(1)})(W_j^{(1)})^\top)$
$\quad\quad W_j^{(1)} \leftarrow \frac{1}{\left\| \tilde{W_j^{(1)}} \right\|} \tilde{W_j^{(1)}}$
**end for**

**Fix layer 1, update layer 2:**
**Re-initialize** $W^{(2)} \to \mathbf{0}_{p_2 \times p_1}$
Sample mini-batch $X, \mathbf{y} \subset \mathcal{D}$ of size $n_2$
$W_2 \leftarrow \left( \frac{1}{n} \sigma(X(W_1)^\top)^\top (\sigma(X(W_1)^\top) + \lambda \right)^{-1} \nabla_{W_2} \mathcal{L}$

**Fix layers 1,2, solve for $W^{(3)}$ via ridge regression:**
Sample mini-batch $X, \mathbf{y} \subset \mathcal{D}$ of size $n_3$
**Form design matrix $H$:**
$\quad$ **For each $(x, y)$ in $\mathcal{D}$:**
$\quad\quad h_1 \leftarrow \sigma(W^1 x + b^{(1)})$
$\quad\quad h_2 \leftarrow \sigma(W^2 h_1 + b^{(2)})$
$\quad\quad H_{(x,:)} \leftarrow [h_2]^\top, \quad Y_x \leftarrow y$
**Solve:**
$\quad W^{(3)} \leftarrow (H^\top H + \lambda I)^{-1} H^\top Y$

---

**Proposition 9.** *For any $\varepsilon_1 > 0$, and $k \in \mathbb{N}$, let $h_\star(\mathbf{x})$ for $\mathbf{x} \in \mathbb{R}^d$ denote a non-linear feature of the form:*

$$h_\star(\mathbf{x}) = \frac{1}{\sqrt{d^{\varepsilon_1}}} \sum_{i=1}^{d^{\varepsilon_1}} \mathrm{P}_k(\langle \mathbf{w}_i^\star, \mathbf{x} \rangle), \tag{74}$$

*where $\mathrm{P}_k$ denote polynomials of degree $k$ satisfying $\mathbb{E}_{z \sim \mathcal{N}(0,1)}[\mathrm{P}_k(z)] = 0$ and $\mathbb{E}_{z \sim \mathcal{N}(0,1)}[\mathrm{P}_k(z)z] = 0$*

*Denote by $S$ the set of indices in $[d^{\varepsilon_1}]$ and by $\Gamma_m(S)$ the set of all $m$-permutations in $S$ consisting of distinct values.*

*Then, the following holds for any $m \in \mathbb{N}$:*

$$\mathrm{He}_m(h_\star(\mathbf{x})) = \frac{1}{\sqrt{m d^{m\varepsilon_1}}} \sum_{s \in \Gamma_m(S)} \prod_{s_i} \mathrm{P}_k(\langle \mathbf{w}_{s_i}^\star, \mathbf{x} \rangle) + r_m(\mathbf{x}) = \mathcal{O}_\prec(1), \tag{75}$$

*where $r_m(\mathbf{x})$ satisfies:*

*(i)*

$$r_m(\mathbf{x}) = \mathcal{O}_\prec\left( \frac{1}{\sqrt{d^{\varepsilon_1}}} \right). \tag{76}$$

*(ii) For any $k \in \mathbb{N}$ and $v \in \mathbb{R}^d$:*

$$\left\| \mathbb{E}\left[ C_k(r_m(\mathbf{x})) He_{k-1}(\langle \mathbf{v}, \mathbf{x} \rangle) \mathbf{x} \right] \right\|_2 = \tilde{\mathcal{O}}\left( \frac{1}{\sqrt{d^{\varepsilon_1}}} (\max_{i \in d^{\varepsilon_1}} |\langle \mathbf{w}_i^\star, \mathbf{v} \rangle|)^{k-1} \right), \tag{77}$$

*for some $\tilde{\delta} > 0$,*

*where recall that $\tilde{\mathcal{O}}$ subsumes factors of the form $d^\delta$ for arbitrarily small $\delta > 0$.*

*The above set of properties characterize, in particular the projections onto Hermite-polynomials of $\mathbf{x}$ of non-linear functions applied to $h^\star(\mathbf{x})$*

*Proof.* The proof proceeds by induction. Similar to [84], the central idea is to utilize the fact that the Hermite-degree is additive for products of terms dependent on orthogonal subspaces. The entropically-dominant terms in $He_p(h_\star(\mathbf{x}))$ arise from products of $\langle \mathbf{w}_i^\star, \mathbf{x} \rangle$, $\langle \mathbf{w}_j^\star, \mathbf{x} \rangle$ for $i \neq j$ contributing a leading Hermite-degree of $dk$.

We show inductively that Equation 75 holds for any $m \in \mathbb{N}$.

The base case $m = 1$ holds trivially. Suppose that the statement holds for some $m \in \mathbb{N}$. Recall that the (normalized) Hermite polynomials satisfy the following recursion:

$$He_{m+1}(x) = x \sqrt{\frac{m}{m+1}} He_m(x) - m \sqrt{\frac{m-1}{m+1}} He_{m-1}(x). \tag{78}$$

Applying the above relation with $x = h_\star(\mathbf{x})$ yields:

$$He_{m+1}(h_\star(\mathbf{x})) = \sqrt{\frac{m}{m+1}} \left( \frac{1}{\sqrt{d^{\varepsilon_1}}} \sum_{i=1}^{d^{\varepsilon_1}} P_k(z_i) \right) He_m(h_\star(\mathbf{x})) - m \sqrt{\frac{m-1}{m+1}} He_{m-1}(h_\star(\mathbf{x})) \tag{79}$$

The induction hypothesis on $He_m(h_\star(\mathbf{x}))$, then implies:

$$He_m(h_\star(\mathbf{x}))$$
$$= \sqrt{\frac{m}{m+1}} \frac{1}{\sqrt{d^{\varepsilon_1}}} \sum_{i=1}^{d^{\varepsilon_1}} P_k(\langle \mathbf{w}_i^\star, \mathbf{x} \rangle) \left( \frac{1}{\sqrt{m d^{m\varepsilon_1}}} \sum_{s \in \Gamma_m(S)} \prod_{s_i} P_k(\langle \mathbf{w}_{s_i}^\star, \mathbf{x} \rangle) + r_m(\mathbf{x}) \right) - m \sqrt{\frac{m-1}{m+1}} He_{m-1}(h_\star(\mathbf{x})) \tag{80}$$

The first term splits into two components depending on whether $i \in s$ or $i \notin s$:

$$He_m(h_\star(\mathbf{x})) = \sum_{i=1}^{d^{\varepsilon_1}} (P_k(\langle \mathbf{w}_i^\star, \mathbf{x} \rangle))^2 \left( \frac{1}{\sqrt{(m+1)d^{(m+1)\varepsilon_1}}} \sum_{s \in \Gamma_{m-1}(S \setminus i)} \prod_{s_i} P_k(\langle \mathbf{w}_{s_i}^\star, \mathbf{x} \rangle) \right) +$$
$$+ \left( \frac{1}{\sqrt{(m+1)d^{(m+1)\varepsilon_1}}} \sum_{s \in \Gamma_{m+1}(S)} \prod_{s_i} P_k(\langle \mathbf{w}_{s_i}^\star, \mathbf{x} \rangle) \right) - m \sqrt{\frac{m-1}{m+1}} He_{m-1}(h_\star(\mathbf{x})) + \mathcal{O}_\prec\left( \frac{1}{\sqrt{d^{\varepsilon_1}}} \right), \tag{81}$$

where we used that $\sum_{i=1}^{d^{\varepsilon_1}} \frac{1}{\sqrt{d^{\varepsilon_1}}} P_k(\langle \mathbf{w}_i^\star, \mathbf{x} \rangle) r_m(\mathbf{x}) = \mathcal{O}_\prec\left( \frac{1}{\sqrt{d^{\varepsilon_1}}} \right)$ through the closure under-muliplication of $\mathcal{O}_\prec(\cdot)$ and Lemma 8. The second term is exactly the desired expression for $He_m(h^\star(\mathbf{x}))$ in Equation 75.

Next, we rewrite the first term as:

$$\underbrace{\sum_{i=1}^{d^{\varepsilon_1}} \left( \frac{1}{\sqrt{(m+1)d^{(m+1)\varepsilon_1}}} \sum_{s \in \Gamma_{m-1}(S/i)} \prod_{s_i} P_k(\langle \mathbf{w}_{s_i}^\star, \mathbf{x} \rangle) \right)}_{T_1}$$

$$+ \underbrace{\sum_{i=1}^{d^{\varepsilon_1}} \frac{1}{\sqrt{(m+1)d^{(m+1)\varepsilon_1}}} ((P_k(\langle \mathbf{w}_i^\star, \mathbf{x} \rangle))^2 - 1) \sum_{s \in \Gamma_{m-1}(S/i)} \prod_{s_i} P_k(\langle \mathbf{w}_{s_i}^\star, \mathbf{x} \rangle)}_{T_2}.$$

By the induction hypothesis, $T_1$ cancels with $-m\sqrt{\frac{m-1}{m+1}}\,\mathrm{He}_{m-1}(h_\star(\mathbf{x}))$ upto an error $\mathcal{O}_\prec(\frac{1}{\sqrt{d^\varepsilon}})$. It remains to show that $T_2$ is stochastically dominated as $\mathcal{O}_\prec(\frac{1}{\sqrt{d^\varepsilon}})$. To achieve this, we note by Gaussian hypercontractivity (Lemma 5), it suffices to bound the second-moment of $T_2$. We have:

$$\mathbb{E}\left[T_2^2\right] = \frac{1}{d^{(m+1)\varepsilon_1}}\sum_{i=1}^{d^{\varepsilon_1}}((\mathrm{P}_k(\langle\mathbf{w}_i^\star,\mathbf{x}\rangle))^2-1)\sum_{s\in\Gamma_{m-1}(S/i)}\prod_{s_i}\mathrm{P}_k(\langle\mathbf{w}_{s_i}^\star,\mathbf{x}\rangle))^2$$

$$+\frac{1}{d^{(m+1)\varepsilon_1}}\sum_{i\neq j=1}^{d^{\varepsilon_1}}\prod_{k=l}(\mathrm{P}_k(\langle\mathbf{w}_k^\star,\mathbf{x}\rangle)^3-\mathrm{P}_k(\langle\mathbf{w}_k^\star,\mathbf{x}\rangle))\sum_{s\in\Gamma_{m-2}(S/(i,j))}(\prod_{s_i}\mathrm{P}_k(\langle\mathbf{w}_{s_i}^\star,\mathbf{x}\rangle)^2),$$

where in the last line we used the fact that the cross-terms vanish for terms with $\mathrm{He}_k(\langle\mathbf{w}_{s_i}^\star,\mathbf{x}\rangle)$ appearing once. The desired bound is obtained by noting that by the inductive hypothesis:

$$\sum_{s\in\Gamma_{m-1}(S/i)}\prod_{s_i}\mathrm{He}_k(\langle\mathbf{w}_{s_i}^\star,\mathbf{x}\rangle) = \mathcal{O}_\prec(\sqrt{d^{(m-1)\varepsilon_1}}). \tag{82}$$

Therefore the first term contributes $d^{\varepsilon_1}$ terms of order $\mathcal{O}_\prec(d^{(m-1)\varepsilon_1})$ while the second term consists of $d^{2\varepsilon_1}$ terms of order $\mathcal{O}_\prec(d^{(m)\varepsilon_1})$. Therefore, both the terms are entropically sub-dominant compared to the factor $\frac{1}{d^{(m+1)\varepsilon_1}}$, yielding:

$$T_2 = \mathcal{O}_\prec(\frac{1}{\sqrt{d^{\varepsilon_1}}}) \tag{83}$$

It remains to show statement $(ii)$ (Equation 77). We first consider the residual term:

$$\sum_{i=1}^{d^{\varepsilon_1}}\frac{1}{\sqrt{d^{\varepsilon_1}}}\mathrm{P}_k(\langle\mathbf{w}_i^\star,\mathbf{x}\rangle)r_m(\mathbf{x}) \tag{84}$$

Recall that for any $v\in\mathbb{R}^d$ and any $r(\mathbf{x})$:

$$\mathbb{E}\left[\langle\nabla^k r(\mathbf{x})\mathbf{v}^{\otimes k}\rangle\right] = \mathbb{E}\left[r(\mathbf{x})\mathrm{He}_k(\langle\mathbf{x},\mathbf{v}\rangle)\right]. \tag{85}$$

Therefore, by induction and the closure of stochastic domination under multplication, the above term satisfies $(ii)$.

For the remaining term $T_2$, $(ii)$ holds by noting that by Proposition 6, for each $i\in\sqrt{d^{\varepsilon_1}}$, $\|\mathbb{E}\left[C_k(r_m(\mathbf{x}))\mathrm{He}_{k-1}(\langle\mathbf{v},\mathbf{x}\rangle)\langle\mathbf{x},\mathbf{w}_i^\star\rangle\right]\|_2$ is a polynomial in $\{\langle\mathbf{w}_i^\star,\mathbf{v}\rangle\}_{i\in\sqrt{d^{\varepsilon_1}}}$ of degree at-least $k-1$. Since $\{\langle\mathbf{x},\mathbf{w}_i^\star\rangle\}$ are orthonormal functions:

$$\sum_{i=1}^{\sqrt{d^{\varepsilon_1}}}\|\mathbb{E}\left[C_k(r_m(\mathbf{x}))\mathrm{He}_{k-1}(\langle\mathbf{v},\mathbf{x}\rangle)\langle\mathbf{x},\mathbf{w}_i^\star\rangle\right]\|_2^2 \le \|\mathbb{E}\left[C_k(r_m(\mathbf{x}))\mathrm{He}_{k-1}(\langle\mathbf{v},\mathbf{x}\rangle)\right]\|^2 = \tilde{\mathcal{O}}(\frac{1}{d^{\varepsilon_1}})$$
$$\tag{86}$$
$$\square$$

### C.7 Existence of activation satisfying Assumptions 2, 3

Consider any $\sigma:\mathbb{R}\to\mathbb{R}$ and a constant $c>0$. Observe that the activation $\tilde{\sigma}:\mathbb{R}\to\mathbb{R}$ defined as:
$$\tilde{\sigma}(x) := \sigma(x)-cx, \tag{87}$$
satisfies:
$$\tilde{\sigma}((\tilde{\sigma})(x)) = \sigma(\sigma(x)-cx)-c(\sigma(x)-cx). \tag{88}$$

Set $\sigma(x)$ as a bounded-analytic function with $\mathbb{E}\left[\sigma(z)\mathrm{He}_k(z)\right]\neq 0$, for instance $\sigma(z)=\tanh(z+a)-bz$, for some $a,b\neq 0$ such that such that $\mathbb{E}\left[\sigma(z)\mathrm{He}_k(z)\right]\neq 0$ for all $k\in\mathbb{N}$. Furthermore, we may further set $a,b\in\mathbb{R}$ such that $\mathbb{E}_{z\sim\mathcal{N}(0,1)}\left[\sigma(\sigma(z))\right]<0$, for instance by setting $b\approx 0$ and $a\approx 0, a<0$.

Then, by Equation 88, the condition $\mathbb{E}_{z\sim\mathcal{N}(0,1)}\left[\tilde{\sigma}((\tilde{\sigma})(z))z\right]=0$ corresponds to the following equation on $c$:

$$g(c) = \mathbb{E}_{z\sim\mathcal{N}(0,1)}\left[\tilde{\sigma}((\tilde{\sigma})(z))z\right] = \mathbb{E}_{z\sim\mathcal{N}(0,1)}\left[\sigma(\sigma(z)-cz)-c(\sigma(z))+c^2z)z\right]$$
$$= \mathbb{E}_{z\sim\mathcal{N}(0,1)}\left[\sigma(\sigma(z)-cz)\right]-c\mathbb{E}_z\left[(\sigma(z))z\right]+c^2 = 0.$$

By the choice of $\sigma$, $g(0)<0$ while the boundedness of $\sigma$ further implies that $g(c)\to\infty$ as $c\to\infty$. Hence $\exists c\in\mathbb{R}$ such that $g(c)=0$. On the other hand, note that $\tilde{\sigma}$

## C.8  Feature Learning by the First Layer

In this section, we analyze the dynamics of $W_1$ (part $(i)$ of Theorem 1). In fact, for subsequent usage in the dynamics of $W_2, \mathbf{w}_3$, we require a stronger characterization of $(i)$ of Theorem 1. To state the precise result, we first set up the required notation. Let $\mathcal{D}_t = \{X_t, \mathbf{y}_t\}$ denote the batch of samples at time-step $t$ for $t \in \mathbb{N}$. Observe that under the correlation loss, and with $W_2 = \mathbb{I}$, each neuron $w_i$ for $i \in [p]$ evolves independently. In-fact, the dynamics is equivalent to that of a two-layer network with modified activation $\tilde{\sigma} = \sigma(\sigma(\cdot)))$

Therefore, the gradient descent dynamics on $W_1$ defines a stochastic mapping:

$$\mathbf{w}^0 \rightarrow \mathbf{w}^{(t)}, \tag{89}$$

applied to a random variable $\mathbf{w}^0 \sim U(\mathcal{S}^{d-1}(1))$.

Let $\{\mathcal{F}_t\}_{t\in\mathbb{N}}$ denote the filtration generated by $\mathcal{D}_1, \mathcal{D}_2, \cdots$. Let $U^\star \in \mathbb{R}^d$ denote the subspace spanned by the teacher weights $W^\star$. Define $\mathbf{u}^\star := \frac{P_{U^\star}\mathbf{w}_0}{\|P_{U^\star}\mathbf{w}_0\|}$ to be the unit-vector along $P_{W^\star}w_0$. Our analysis proceeds by establishing the following:

(i) The dynamics of $\mathbf{w}^{(t)}$ is dominated by drift along the initial direction $u^\star$.

(ii) The overlap of $\mathbf{w}^{(t)}$ along $W^\star$ grows linearly upto reaching a threshold $\kappa > 0$ and subsequently $\mathbf{w}^{(t)}$ reaches overlap $\kappa$ in a constant number of iterations.

(iii) The distribution of $\mathbf{w}^{(t)}$ maintains isotropy and regularity of tails.

To see intuitively why the dynamics of $\mathbf{w}^{(t)}$ is dominated by the drift along $u^\star$, consider the following heuristic sketch:

$$\frac{d\mathbf{w}^{(t)}}{dt} = -\nabla_{\mathbf{w}^{(t)}} \mathcal{L}_c$$
$$= c\mathbb{E}\left[h^\star(\mathbf{x})\sigma'(\mathbf{w}^{(t)})\mathbf{x}\right] + \mathbb{E}\left[r(\mathbf{x})\sigma'(\mathbf{w}^{(t)})\mathbf{x}\right] + \text{higher-order terms},$$

where we substituted the decomposition in Proposition 9.

Next, we note that the degree $j$ contribution in $\mathbb{E}\left[h^\star(\mathbf{x})\sigma'(\mathbf{w}^{(t)})\mathbf{x}\right]$ is of the form:

$$\mathbb{E}\left[\frac{1}{\sqrt{d^{\epsilon_1}}}\sum_{i=1}^{d^{\epsilon_1}} \mathrm{He}_j(\langle \mathbf{w}_i^\star, \mathbf{x}\rangle)\sigma'(\mathbf{w}^{(t)})\mathbf{x}\right] = \frac{1}{\sqrt{d^{\epsilon_1}}}\sum_{i=1}^{d^{\epsilon_1}}(\langle \mathbf{w}_i^\star, \mathbf{w}^{(t)}\rangle)^{j-1}\mathbf{w}_i^\star. \tag{90}$$

For $j = 2$, the above term results in a drift along $\mathbf{u}_i^\star$, while for $j > 2$, the contributions are suppressed as long as $\langle \mathbf{w}_i^\star, \mathbf{w}^{(t)}\rangle = \mathcal{O}_{\prec}(\frac{1}{\sqrt{d^{\epsilon_1}}})$. Analogously, the contributions from higher-order terms are suppressed as long as $\mathbf{w}^{(t)}$ doesn't align with individual directiosn $\mathbf{w}_i^\star$.

We now move on to the full proof. Let $\kappa > 0$ be fixed We introduce the following hitting time:

$$\tau_\kappa := \{\inf t : |\langle \mathbf{u}^\star, \mathbf{w}^{(t)}\rangle| \geq \kappa\}. \tag{91}$$

Let $\mathcal{F}^\star$ denote the product sigma-algebra w.r.t $\{\mathcal{F}_t\}$. Since $\tau_\kappa$ is measurable w.r.t $\sigma(\mathcal{F}^\star \cup \mathcal{F}(\mathbf{w}_0))$, the random variable $\mathbf{w}^{\tau_\kappa}$ then admits a regular conditional distribution w.r.t $\mathcal{F}^\star$, $\mu_\kappa(|\mathcal{F}^\star)$ [48].

Suppose that each neuron for $i \in [p_1]$ in Algorithm 1 is stopped at $\tau_\kappa$ as defined above.

Let $\mathbf{e}_1, \cdots \mathbf{e}_{d-d^{\epsilon_1}}$ denote a fixed basis for the complement of $W^\star$. The main result of this section establishes points $(i) - (iii)$ described above and constitutes the formal statement for part $(i)$ of Theorem 1:

**Theorem 5.** *For any $0 < \kappa < 1$, let $\mu_\kappa(\cdot|X_1, X_2, \cdots)$ denote the regular conditional measure over $w_\kappa^\tau$ conditioned on the sequence of datasets $X_1, X_2, \cdots$ associated with the natural filtration $\mathcal{F}_t$. Then, for any $k \in \mathbb{N}$, there exists a sequence of "high-probability" events $\mathcal{E} \in \cup_{t\geq 1}\{\mathcal{F}_t\}$ such that:*

(i) $\Pr[\mathcal{E}] \geq 1 - \frac{C_k}{d^k}$ *for some $C_k > 0$ and large enough $d$.*

(ii) *For any $X_1, X_2, \dots \in \mathcal{E}$, the random variable $\mathbf{w} \sim \mu_\eta(\cdot | X_1, X_2, \dots)$, satisfies the following with probability $1 - Ce^{-C \log d^2}$ as $d \to \infty$:*

$$\mathbf{w}_\kappa^\tau = \kappa^+ \mathbf{u}^\star + \mathbf{u}_\perp + \mathbf{v}, \tag{92}$$

*where $\kappa^+ \geq \kappa$ and:*

(a) $\mathbf{u}_\perp \in U^\star, \mathbf{v} \in U_\perp^\star$.

(b) $\|\mathbf{u}_\perp\| = \mathcal{O}_\prec(\frac{1}{d^\delta})$.

(c)

$$\sup_{i \in d^{\varepsilon_1}} |\langle \mathbf{w}_\kappa^\tau, \mathbf{w}_i^\star \rangle| = O_\prec(\frac{1}{\sqrt{d^{\varepsilon_1}}}). \tag{93}$$

*and*

$$\sup_{j \in [d - d^{\varepsilon_1}]} |\langle \mathbf{w}^{\tau_\kappa}, \mathbf{e}_j \rangle| = O_\prec(\frac{1}{\sqrt{d}}). \tag{94}$$

(d) *For any (deterministic) $\mathbf{w}^\star \in U^\star$:*

$$|\langle \mathbf{w}^\star, \mathbf{w}^{\tau_\kappa} \rangle| = O_\prec(\frac{1}{\sqrt{d^{\varepsilon_1}}}), \tag{95}$$

*and for any $\mathbf{w}_\perp \in U_\perp^\star$:*

$$|\langle \mathbf{w}_\perp, \mathbf{w}_\kappa^\tau \rangle| = O_\prec(\frac{1}{\sqrt{d}}). \tag{96}$$

(e)

$$\|\mathbf{v}\| - |\langle \mathbf{w}^0, \mathbf{v} \rangle| = \mathcal{O}_\prec(\frac{1}{d^\delta}), \tag{97}$$

*where $\mathbf{w}^0 \sim U(\mathbb{S}^{d-1}(1))$ denotes the initialization of the neuron.*

Properties $(c)$ stipulates that $\mathbf{w}_\kappa^\tau$ remains approximately isotropic with well-behaved tails along a fixed basis of $U^\star$ and its complement. This is important for ensuring concentration of well-behaved functions of $\mathbf{w}_\kappa^\tau$ in part $(ii)$. Maintaining this property throughout the dynamics further leads to a control over the higher-order terms.

**Corollary 1.** *Let $\mathbf{w}_\eta^\tau$ be as defined in Theorem 5. Then, $\exists \, \delta > 0$ and choice of step-size $\eta = \tilde{\eta}\sqrt{d^{\varepsilon_1}}$ for some $\tilde{\eta} > 0$ such that:*

$$\left\| \mathbb{E}\left[\mathbf{w}_\kappa^\tau (\mathbf{w}_\kappa^\tau)^\top\right] - \kappa \frac{1}{\sqrt{d^{\varepsilon_1}}}(W^\star)^\top(W^\star) - \sqrt{1 - \kappa^2} \frac{1}{\sqrt{d - d^{\varepsilon_1}}}(I - (W^\star)^\top(W^\star)) \right\| = \mathcal{O}(\frac{1}{d^\delta}) \tag{98}$$

### C.9 Form of the Update

Under the initialization $W_2 = \mathbf{I}_p, b_1, b_2 = \mathbf{0}, w_3 = \mathbf{1}$, for any $i \in [p_1]$, the update to any neuron $w$ in $W_1$ can be expressed as:

$$\tilde{\mathbf{w}}^t = \mathbf{w}^t + (\mathbf{I}_d - (\mathbf{w}_t)(\mathbf{w}_t)^\top)\eta \frac{1}{n}\sum_{i=1}^n f^\star(\mathbf{x}_i)\tilde{\sigma}'(\langle \mathbf{w}^t, \mathbf{x}_i \rangle)\mathbf{x}_i$$

$$\mathbf{w}^{t+1} = \frac{1}{\|\tilde{\mathbf{w}}^t\|_2}\tilde{\mathbf{w}}^t \tag{99}$$

In what follows, we denote the gradient update as:

$$\mathbf{g}^t := \frac{1}{n}\sum_{i=1}^n f^\star(\mathbf{x}_i^t)\tilde{\sigma}'((\langle \mathbf{w}^t, \mathbf{x}_i \rangle)\langle \mathbf{w}^{(t)}, \mathbf{x}_i^t \rangle))\mathbf{x}_i^t, \tag{100}$$

and its corresponding spherical version as:

$$\mathbf{g}_\perp^t := \mathbf{g}^t(\mathbb{I} - (\mathbf{w}^{(t)})(\mathbf{w}^{(t)})^\top), \tag{101}$$

where we recall that $\left\|\mathbf{w}^{(t)}\right\| = 1$ by the spherical constraint.

Applying the Hermite decomposition to $f^\star(\mathbf{x})$ and utilizing the composition of Hermite-coefficients established in Proposition 9 results in the following expansion for the gradient:

**Lemma 7.** *Let $C_k^\star$ for $k \in \mathbb{N}$ denote the $k_{th}$-order Hermite tensor of $f^\star(z)$ and let $\{c_k\}_{k=1}^\infty$ be the Hermite coefficients of $\tilde{\sigma}(\cdot)$. Then:*

$$\mathbb{E}[\mathbf{g}^\perp] = (\mathbf{I}_d - (\mathbf{w}_t)(\mathbf{w}_t)^\top)\frac{1}{\sqrt{p}}\left(\sum_{k=0}^\infty c_{k+1} C_{k+1}^\star \times_{1\dots k} (\mathbf{w}^t)^{\otimes k}\right) \tag{102}$$

*Proof.* The above is a direct consequence of Stein's Lemma applied to $\mathbb{E}[\mathbf{g}^t]$:

$$\mathbb{E}\left[\mathbf{x}\tilde{\sigma}'(\langle \mathbf{w}, \mathbf{x}\rangle)f^\star(\mathbf{x})\right] = \mathbb{E}\left[\nabla_{\mathbf{x}}\sigma'(\langle \mathbf{w}, \mathbf{x}\rangle)f^\star(\mathbf{x})\right] + \mathbb{E}\left[\tilde{\sigma}'(\langle \mathbf{w}, \mathbf{x}\rangle)\nabla_{\mathbf{x}}f^\star(\mathbf{x})\right]$$
$$= \mathbf{w}\mathbb{E}\left[\tilde{\sigma}''(\langle \mathbf{w}, \mathbf{z}\rangle)f^\star(\mathbf{z})\right] + \mathbb{E}\left[\sigma'(\langle \mathbf{w}, \mathbf{x}\rangle)\nabla_{\mathbf{x}}f^\star(\mathbf{x})\right].$$

The first term vanishes under the orthogonal projection $(\mathbf{I}_d - (w^{(t)})(w^{(t)})^\top)$ while the second term results in Equation 102. $\square$

Combining the above with the recursive Hermite decomposition of $f^\star(\mathbf{x})$ through Proposition 9 yields the following form for the expected updates:

**Proposition 10.** *Let $\{\mu_k\}_{k=1}^\infty, \{c_k\}_{k=1}^\infty, \{c_k^\star\}_{k=1}^\infty$ denote the Hermite coefficients of $g^\star, \tilde{\sigma}, P_\star^k$ respectively. Then:*

$$\mathbb{E}\left[\mathbf{g}_t^\perp\right] = (\mathbf{I}_d - (\mathbf{w}_t)(\mathbf{w}_t)^\top)\Big(\frac{1}{\sqrt{d^{\varepsilon_1}}}c_2 c_2^\star \mu_1 v^\top (W^\star)^\top (W^\star)\mathbf{w} + \mu_1 \sum_{j=3}^k c_j^\star c_j \frac{1}{\sqrt{d^{\varepsilon_1}}}\sum_{i=1}^{d^{\varepsilon_1}}(\langle \mathbf{w}_i^\star, \mathbf{w}\rangle)(\langle \mathbf{w}_i^\star, \mathbf{v}\rangle)$$

$$+ \sum_{m=k+1}^\infty \mu_m c_m \frac{1}{\sqrt{md^{m\varepsilon_1}}}\sum_{s\in\Gamma(S,m),j\in[k]^s}\prod_{i=1}^{m-1}c_{j_i}^\star(\langle \mathbf{w}_{s_i}^\star, \mathbf{w}\rangle)^{j_1}\langle \mathbf{w}_{s_m}^\star, \mathbf{v}\rangle + \sum_{m=1}^\infty c_m \mathbb{E}\left[r_m(\mathbf{x})\tilde{\sigma}'(\langle \mathbf{w}, \mathbf{x}\rangle)\langle \mathbf{x}, \mathbf{v}\rangle\right]\Big),$$

$$\tag{103}$$

*where $r_m(\mathbf{x})$ denotes the remainder for the degree-$m$ Hermite term in Equation 75.*

*Proof.* Proposition 9 applied to $f^\star(\mathbf{x})$ yields:

$$f^\star(\mathbf{x}) = \sum_{m=1}^\infty \mu_m \frac{1}{\sqrt{md^{m\varepsilon_1}}}\sum_{s\in\Gamma_m(S)}\prod_{s_i}P_k(\langle \mathbf{w}_{s_i}^\star, \mathbf{x}\rangle) + \sum_{m=1}^\infty \mu_m r_m(\mathbf{x}). \tag{104}$$

Next, by expanding $P_k(\langle \mathbf{w}_{s_i}^\star, \mathbf{x}\rangle) = \sum_{j=1}^k c_k^\star \mathrm{He}_j(\langle \mathbf{w}_{s_i}^\star, \mathbf{x}\rangle)$, the first term in the RHS can be further decomposed as:

$$\sum_{m=k+1}^\infty \mu_m \frac{1}{\sqrt{d^{m\varepsilon_1}}}\sum_{s\in\Gamma_m(S)}\prod_{s_i}P_k(\langle \mathbf{w}_{s_i}^\star, \mathbf{x}\rangle) = \sum_{m=k+1}^\infty \mu_m \frac{1}{\sqrt{d^{m\varepsilon_1}}}\sum_{s\in\Gamma(S,m),j\in[k]^s}\prod_{i=1}^m c_{j_i}^\star \mathrm{He}_{j_1}(\langle \mathbf{w}_{s_i}^\star, \mathbf{w}\rangle).$$

$$\tag{105}$$

Equation 103 then follows by noting that any term of the form $\prod_{i=1}^m c_{j_i}^\star \mathrm{He}_{j_1}(\langle \mathbf{w}_{s_i}^\star, \mathbf{w}\rangle)$ appears in $C_\ell^\star$ with $\ell = \sum_{i=1}^m j_i$. $\square$

The magnitude of the gradient updates is bounded through the following Lemma:

**Proposition 11.** *Let $g_{\perp,i}^t := \sigma f^\star(\mathbf{x})\sigma'(\sigma'(\langle \mathbf{w}, \mathbf{x}\rangle))\mathbf{x}(I - \frac{1}{\varepsilon^2}\mathbf{w}\mathbf{w}^\top)$ denote the spherical gradient for neuron $i$ at time-step $t$. Then $g_\perp$ satisfies:*

*(i)*

$$\left\|\mathbf{g}_\perp^t\right\|^2 = \left\|\mathbb{E}\left[\mathbf{g}_\perp^t\right]\right\|^2 + \mathcal{O}_\prec(\frac{1}{d^{\varepsilon_1+\delta}}), \tag{106}$$

*(ii) For any $v \in \mathbb{R}^d$ with $\|v\| = 1$:*

$$\langle \mathbf{g}_\perp^t, \mathbf{v}\rangle - \mathbb{E}\left[\langle \mathbf{g}_\perp^t, \mathbf{v}\rangle\right] = \mathcal{O}_\prec(\frac{1}{\sqrt{n_1}}) = \mathcal{O}_\prec(\frac{1}{\sqrt{d^{1+\delta}}\sqrt{d^{\varepsilon_1}}}) \tag{107}$$

*(iii)*

$$\left\|P_{U^\star}\mathbf{g}_\perp^t\right\| - \mathbb{E}\left[\left\|P_{U_\perp^\star}\mathbf{g}^t\right\|\right] = \mathcal{O}_\prec(\frac{1}{\sqrt{d^{1+\delta}}}) \tag{108}$$

*Proof.* We begin by writing:

$$\left\|\mathbf{g}_\perp^t\right\|^2 = \left\|\mathbb{E}\left[\mathbf{g}_\perp^t\right]\right\|^2 + \left\|\mathbf{g}_\perp^t - \mathbb{E}\left[\mathbf{g}_\perp^t\right]\right\|^2. \tag{109}$$

By the assumption on $\sigma$, the composed activation $\tilde{\sigma}$ and its derivatives are polynomially bounded. Therefore, applying standard concentration results for for independent random variables with bounded orlicz norm (Theorem 4), we obtain:

$$\mathbf{g}_\perp^t - \mathbb{E}\left[\mathbf{g}_\perp^t\right] = \mathcal{O}_\prec(\sqrt{\frac{d}{n}}) = \mathcal{O}_\prec(\frac{1}{\sqrt{d^{\delta+\varepsilon_1}}}) \tag{110}$$

which yields Equation 106.

Applying the same result (Theorem 4) to projections of $\mathbf{g}_\perp^t$ along $\mathbf{v}$ and $P_{U^\star}$ then yields Equations 107 and 108. $\qquad\square$

## C.10 Initial Overlaps

Before proceeding with the analysis, we collect the following result on the concentration of the initial overlaps along $W^\star$ for the first-layer neurons at initialization:

**Lemma 8.** *For* $\mathbf{w} \sim \mathcal{N}(0, \frac{1}{d}\mathbf{I})$,

$$|\sqrt{d^{1-\varepsilon_1}}(\left\|P_U^\star\mathbf{w}\right\|_2 - 1| = \mathcal{O}_\prec(\frac{1}{\sqrt{d^{\varepsilon_1}}}) \tag{111}$$

*Proof.* Since $\mathbf{w}^0 \sim U(\mathcal{S}^{d-1}(1))$, the squared overlap norm $\frac{d}{d^{\varepsilon_1}}\left\|W^\star\mathbf{w}_{1,i}^0\right\|_2^2$ is an average over $d^{\varepsilon_1}$ sub-exponential random variables. Therefore, a standard application of Bernstein's inequality [83] yields an error probability of $1 - ce^{-\log d^2}$. The proposition then follows through a union bound over the $p_1$ neurons. $\qquad\square$

## C.11 Difference Inequality

Let $P_{U^\star}^\star := (W^\star)^\top(W^\star)$ denote the projector onto the subspace spanned by $W^\star$. For $i \in [p_1]$, define $\mathbf{u}_i^\star$ to be the unit-vector along the projection of $\mathbf{w}_i^0$ along $W^\star$:

$$\mathbf{u}^\star := \frac{1}{\left\|W^\star\mathbf{w}_i^{(0)}\right\|}P_{U^\star}^\star\mathbf{w}^0 \tag{112}$$

Further define $m^t = \langle\mathbf{u}^\star, \mathbf{w}_{1,i}^{(t)}\rangle$ and $m_\perp^t = \left\|(P_{U^\star}^\star - \mathbf{u}^\star(\mathbf{u}^\star)^\top)\mathbf{w}^{(t)}\right\|$, denoting the projections of $\mathbf{w}^{(t)}$ along $\mathbf{u}_i^\star$ and its complement in the span of $W^\star$ and:

$$m_\times^t := \sqrt{d^{\varepsilon_1}}\max_{i\in\sqrt{d^{\varepsilon_1}}}|\langle\mathbf{w}^{(t)}, \mathbf{w}_i^\star\rangle|. \tag{113}$$

Additionally, we track the residual component in $\mathbf{w}^{(t)}$ lying in $U_\perp^\star$ but orthogonal to $\mathbf{w}_0$:

$$\mathbf{r}_\perp^t := (\mathbb{I} - \mathbf{w}^{(0)}(\mathbf{w}^{(0)})^\top)P_{U_\perp^\star}\mathbf{w}^{(t)} \tag{114}$$

Our analysis relies on showing that the dynamics of $\mathbf{w}^{(t)}$ is dominated by a linear drift along $\mathbf{u}^\star$. This requires a control over the following additional terms:

  (i) Residual linear drift along $U_\perp^\star$: This term is controlled through a bound on $m_\perp^t$.

  (ii) Contributions from higher-order terms: These are controlled through a bound on $m_\times^t$.

  (iii) Noise in the gradient updates: This is supressed through the choice of batch-size $n = \Theta(d^{1+\varepsilon_1+\delta})$

Recall that $n_1 = \Theta(d^{k\varepsilon_1 + \delta})$. Let $\tilde{\delta}$ be any arbitrary value satisfying $0 < \tilde{\delta} < \delta$ For any $\eta > 0$, we define the following stopping times:

$$\tau_\kappa^+ = \inf(t : |m^t| \geq 1 - \kappa) \tag{115}$$

$$\tau_{\tilde{\delta}}^- = \inf(t : |m^t| \leq d^{-\tilde{\delta}} \min(m_\perp^t, m_\times^t, \frac{1}{\sqrt{d^{1-\varepsilon_1}}})) \tag{116}$$

The stopping time $\tau_\kappa^+$ simply accounts for the overlap reaching the desired value $\kappa$. The stopping time $\tau_{\tilde{\delta}}^-$ ensures that for $t \leq \tau_{\tilde{\delta}}^-$, the three residual contributions in $g_\perp^t$ listed above, namely the drift along $U_\perp^\star$, higher-order terms and the gradient noise remain supressed.

Note that by definition, $m_{i,\perp}^0 = 0$ while $m_i^0 = \frac{1}{\sqrt{d^{\varepsilon_1}}}$ by Lemma 8 . While both $m_{i,\perp}^0, m_i^0$ grow exponentially, we will show that for any $0 < \tilde{\delta} < \delta$, there exists a small enough $\tilde{\eta}$ such that with step size $\eta = \tilde{\eta} d^{\varepsilon_1}$, $\tau_\kappa^+ > \tau_{\tilde{\delta}}^-$ with high-probability. Concretely, under small enough step-size, both $m_i^0, m_{i,\perp}^0$ grow under approximately identical linear dynamics, ensuring that the initial lead in the magnitude of $m_i^0$ is maintained till weak-recovery over the contributions from the remaining directions.

**Proposition 12.** *Let $c = \mu_1 c_2^\star \tilde{c}_2$ and $\eta = \tilde{\eta}\sqrt{d^{\varepsilon_1} p_2}$ Define $\tau^\star := \tau_\kappa^+ \wedge \tau_{\tilde{\delta}}^-$ For any $\tilde{\delta} < \delta_\perp < \delta, \kappa$ and $k \in \mathbb{N}$, and any constants $c_m^+, c_\perp^+, c_\times^+, c_m^-, c_\perp^-, c_\times^-$ such that $c_m^- < c < c_m^+$, $c_\perp^- < c < c_\perp^+$, $c_\times^- < c < c_\times^+$ there exists constants $C_1, C_2, C_3, C_4$ and $\tilde{\eta}$ such that for large enough $d$:*

*(i)*

$$\mathbb{P}(m^{t+1} \geq m^t + \tilde{\eta}c_m^- m_t - \tilde{\eta}c_m^+ m_t^2 - \tilde{\eta}^2 C_1 m_t^3, \quad \forall t < \tau^\star) \geq 1 - \frac{1}{d^k} \tag{117}$$

$$\mathbb{P}(m^{t+1} \leq m^t + \tilde{\eta}c_m^+ m_t - \tilde{\eta}c_m^- m_t^2, \quad \forall t < \tau^\star) \geq 1 - \frac{1}{d^k} \tag{118}$$

*(ii)*

$$\mathbb{P}(m_\perp^{t+1} \leq \left( m_\perp^t + \tilde{\eta}c_\perp^+ m_\perp^t - \tilde{\eta}c_\perp^- m^t m_\perp^t + \tilde{\eta}C_2(m_\times^t) + \tilde{\eta}\frac{C_3}{\sqrt{d^{1-\varepsilon_1+\delta_\perp}}}, \forall t < \tau^\star\right)) \geq 1 - \frac{1}{d^k} \tag{119}$$

*(iii)*

$$\mathbb{P}(m_\times^t \leq m_\times^t + \tilde{\eta}c_\times^- m_\times^t - \tilde{\eta}c_\times^+ m_\times^t m^t + \tilde{\eta}\frac{C_4}{\sqrt{d^{1+\delta_\perp}}}, \forall t < \tau^\star) \geq 1 - \frac{1}{d^k} \tag{120}$$

Before establishing the above proposition, we first show how it implies Theorem 5

### C.12 Proof of Theorem 5

Suppose that $t \leq \tau^\star$. Applying a union-bound over time-steps to $(i)$ in Proposition 12 then implies that with probability at-least $t(1 - \frac{1}{d}^k)$, for all $t \leq \tau^\star$, we have with high-probability:

$$m^{t+1} \geq m^t + \tilde{\eta}c_m^- m_t - \tilde{\eta}c_m^+ m_t^2 - \tilde{\eta}^2 C_1 m_t^3. \tag{121}$$

Since $\tau^\star \leq \tau_\kappa^+$, we further have that $m_t \leq \kappa$ for all $t \leq \tau^\star$ and thus $\tilde{\eta}^2 C_1 m_t^3 < \tilde{\eta}^2 C_1 \kappa m_t^2$. Equation 121 then implies:

$$m^{t+1} \geq (1 + \tilde{\eta}c_m^- - (\tilde{\eta}c_m^+ + \tilde{\eta}^2 C_1 \kappa)m^t)m^t, \tag{122}$$

which inductively implies the following intermediate bound:

$$m^{t+1} \geq \prod_{s=1}^t (1 + \tilde{\eta}c_m^- - (\tilde{\eta}c_m^+ + \tilde{\eta}^2 C_1 \kappa)m^s)m^0, \tag{123}$$

Since $m_t^2 \leq \kappa m_t$, the above simplifies to:

$$m^{t+1} \geq m^t + (\tilde{\eta}c_m^+(1 - \kappa) - \tilde{\eta}^2 C_1 \kappa^2)m^t \tag{124}$$

Therefore, $\forall t < \tau^\star$, we have:

$$m^t \geq (1 + \tilde{\eta}c_m^+(1 - \kappa) - \tilde{\eta}^2 C_1 \kappa^2)^t m^0. \tag{125}$$

By Lemma 8, we have:

$$|m^0| = \frac{1}{d^{1/2(1-\varepsilon_1)}} + \mathcal{O}_\prec(\frac{1}{d}), \tag{126}$$

Since for small enough $\tilde{\eta}$, $c_m^+(1 - \kappa) - \tilde{\eta}^2 C_1 \kappa^2 > 0$, Equation 125 implies:

$$\tau^\star \leq c_{\kappa,\varepsilon} \log d, \tag{127}$$

for some constant $c_{\kappa,\varepsilon} > 0$.

Next, consider the orthogonal component $m_\perp^t = 0$. Note that $m_\perp^0 = 0$ by definition. Part $(ii)$ of Proposition 12 along with the discrete Gronwall inequality (Lemma 6) implies:

$$m_\perp^t \leq \sum_{j=1}^{t-1} \prod_{s=1}^{j} (1 + \tilde{\eta}c_\perp^+ - \tilde{\eta}c_\perp^- m^s)(\tilde{\eta}C_2(m_\times^t)^2 + \tilde{\eta}\frac{C_3}{\sqrt{d^{1-\varepsilon_1+\delta_\perp}}})$$

$$\leq \sum_{j=1}^{t-1} \prod_{s=1}^{j} (1 + \tilde{\eta}c_\perp^+ - \tilde{\eta}c_\perp^- m^s)(\tilde{\eta}C_2\frac{1}{d^{2\tilde{\delta}}}(m^t)^2 + \tilde{\eta}\frac{C_3}{\sqrt{d^{1-\varepsilon_1+\delta_\perp}}}),$$

where we used that $m_\times^t \leq \frac{1}{d}^{\tilde{\delta}}$ since $t \leq \tau_{\tilde{\delta}}^-$.

Our goal next is to compare the above bound against the lower-bound given by Equation 123. Since $|\log(1 + a) - \log(1 + b)| \leq |a - b|$ for $a, b > 0$, we have:

$$\frac{(1 + \tilde{\eta}c_m^- - (\tilde{\eta}c_m^+ + \tilde{\eta}^2 C_1 \kappa)m^s)}{(1 + \tilde{\eta}c_\perp^+ - \tilde{\eta}c_\perp^- m^s)} \leq \exp(\tilde{\eta}|c_m^- - c_\perp^+| + |c_m^+ - c-\perp| + \tilde{\eta}^2 C_1 \kappa)) \tag{128}$$

Therefore, we obtain the corresponding bound:

$$m_\perp^t \leq t \exp(\tilde{\eta}|c_m^- - c_\perp^+| + |c_m^+ - c-\perp| + \tilde{\eta}^2 C_1 \kappa)) \left( \frac{1}{d^{2\tilde{\delta}}} C_2 m_0 + \frac{C_3}{\sqrt{d^{1-\varepsilon_1+\delta_\perp}}}) \right) \tag{129}$$

For any $0 < \delta_\perp < \tilde{\delta}$, we may set $|c_m^+ - c_\perp^-|, |c_m^- - c_\perp^+|$ and $\tilde{\eta}$ small enough so that for any $t \leq c_{\kappa,\varepsilon} \log d$:

$$(t\tilde{\eta}|c_m^- - c_\perp^+| + |c_m^+ - c-\perp| + \tilde{\eta}^2 C_1 \kappa) + \log C_2 < \delta_\perp - \tilde{\delta}, \tag{130}$$

Implying:

$$m_\perp^t \leq \frac{m_t}{d^{\tilde{\delta}}} \tag{131}$$

Similarly by setting $|c_m^+ - c_\times^-|, |c_m^- - c_\times^+|$ small enough enough we have by part (iii) in Proposition 12:

$$m_\times^t \leq \frac{m_t}{d^{\tilde{\delta}}} \tag{132}$$

Therefore, while the dynamics of $m^t, m_\perp^t, m_\times^t$ evolves at arbitrarily close rates. The initial advantage in $m^t$ over $m_\perp^t, m_\times^t$ through initalization ensures that the hitting time for $m^t$ arrives first, ensuring that:

$$\tau_\kappa < \tau_{\tilde{\delta}}^- \tag{133}$$

By the definition of $\tau^\star$, this establishes all claims in Theorem 1 apart from $(e)$. To obtain $e$, note that by the form of the updates, $r_\perp^t$ is updated solely through the gradient noise $\mathbf{g}_\perp^t - \mathbb{E}[\mathbf{g}_\perp^t]$ and normalization. Therefore, Lemma 11 implies that:

$$r_\perp^t = \mathcal{O}_\prec(t\frac{1}{d^{\sqrt{\delta}}}). \tag{134}$$

### C.12.1 The conditioning input set

To complete the proof of Theorem 5, it remains to specify the high-probability set $\mathcal{E}_{\kappa,\tilde{\delta}}$. For any $\tilde{\delta} > 0$ and $\kappa \in \mathbb{R}$, consider the event:

$$\mathcal{E}_{\kappa,\tilde{\delta}} : \cap_{t \leq \tau_\kappa^+} \left[ |\langle \mathbf{g}^t, \mathbf{v} \rangle - \mathbb{E}\left[\langle \mathbf{g}^t, \mathbf{v} \rangle\right]| \geq \frac{d^{\tilde{\delta}}}{d^{1+\varepsilon+\delta}} \right] \tag{135}$$

Equations 107, 110 and a union bound imply that for any $k \in \mathbb{N}$:

$$\Pr[\mathcal{E}_{\kappa,\tilde{\delta}}] \geq 1 - \frac{1}{d^k}, \tag{136}$$

where the probability is w.r.t the joint measure over $w_0, X_1, \cdots X_n$

By the law of total expectation:

$$\Pr[\mathcal{E}_{\kappa,\tilde{\delta}}^{\mathsf{C}}] \geq \frac{1}{d^k} \mathbb{E}\left[\mathbf{1}[\Pr[\mathcal{E}_{\kappa,\tilde{\delta}}^{\mathsf{C}}|\{X_i\}_{i\in\mathbb{N}}] \geq \frac{1}{d^k}\right], \tag{137}$$

implying, for any $k > \tilde{k} \in \mathbb{N}$:

$$\mathbb{E}\left[\mathbf{1}[\Pr[\mathcal{E}_{\kappa,\tilde{\delta}}^{\mathsf{C}}|\{X_i\}_{i\in\mathbb{N}}] \geq \frac{1}{d^k}\right] \leq \frac{1}{d^k}, \tag{138}$$

taking the interesection over $k \in \mathbb{N}$ the required conditioning set $\mathcal{E}$ in Theorem 5.

### C.13 Proof of Proposition 12

Consider the "effective" activation:
$$\tilde{\sigma}(x) := \sigma'(\sigma'(x)) \tag{139}$$

Let $\tilde{c}_2 = \mathbb{E}_{z \sim \mathcal{N}(0,1)}\left[\tilde{\sigma}(z)\text{He}_2(z)\right]$. Define the constant $c = \mu_2 c_1^\star \tilde{c}_2$. By assumption 3, $c > 0$.

**part** $(i)$: Applying Proposition 10, we obtain the following decomposition for the update $\mathbf{g}_t^\perp$:

$$\mathbf{g}_t^\perp = (\mathbf{I} - \mathbf{w}^{(t)}(\mathbf{w}^{(t)})^\top)\frac{1}{\sqrt{d^{\varepsilon_1}}}\mu_1 c_2^\star \tilde{c}_2 P_{U^\star}\mathbf{w}^{(t)}$$

$$+(\mathbf{I}-\mathbf{w}^{(t)}(\mathbf{w}^{(t)})^\top)\underbrace{\sum_{j=3}^{k}\mu_1 c_j^\star c_j \frac{1}{\sqrt{d^{\varepsilon_1}}}\sum_{i=1}^{d^{\varepsilon_1}}(\langle \mathbf{w}_i^\star, \mathbf{w}\rangle)^j \mathbf{w}_i^\star + \sum_{m=k+1}^{\infty}\mu_m c_m \frac{1}{\sqrt{md^{m\varepsilon_1}}}\sum_{s\in\Gamma(S,m),j\in[k]^s}\prod_{i=1}^{m-1}c_{j_i}^\star(\langle\mathbf{w}_{s_i}^\star,\mathbf{w}\rangle)^{j_1}\mathbf{w}_{s_m}^\star}_{\Delta_1}$$

$$+(\mathbf{I}-\mathbf{w}^{(t)}(\mathbf{w}^{(t)})^\top)\underbrace{\sum_{m=1}^{\infty}\mathbb{E}\left[\mu_m r_m(\mathbf{x})\tilde{\sigma}'(\langle\mathbf{w},\mathbf{x}\rangle)\mathbf{x}\right]}_{\Delta_2}$$

$$+(\mathbf{I}-\mathbf{w}^{(t)}(\mathbf{w}^{(t)})^\top)\underbrace{\mathbf{g}_t^\perp - \mathbb{E}\left[\mathbf{g}_t^\perp\right]}_{\Delta_3}$$

$$\tag{140}$$

Since $m_t = \langle\mathbf{w}^{(t)}, \mathbf{u}^\star\rangle$, $\|\mathbf{w}^t\| = 1$ the first term simplifies to:

$$\frac{1}{\sqrt{d^{\varepsilon_1}}}\mu_1 c_2^\star \tilde{c}_2(\mathbf{u}^\star)^\top P_{U^\star}\mathbf{w}^{(t)} - \frac{1}{\sqrt{d^{\varepsilon_1}}}\mu_1 c_2^\star \tilde{c}_2\langle\mathbf{w}^{(t)}, \mathbf{u}^\star\rangle\mathbf{w}^{(t)}P_{U^\star}\mathbf{w}^{(t)} = cm_t - cm_t^2. \tag{141}$$

Next, for $\Delta_1$, we separately consider the component along $\mathbf{w}_i^\star$ for each $i \in \sqrt{d^{\varepsilon_1}}$:

$$\langle\Delta_1, \mathbf{w}_i^\star\rangle = \sum_{j=3}^{k}\mu_1 c_j^\star c_j \frac{1}{\sqrt{d^{\varepsilon_1}}}(\langle\mathbf{w}_i^\star, \mathbf{w}\rangle)^j + \mu_m c_m \frac{1}{\sqrt{md^{m\varepsilon_1}}}\sum_{s\in\Gamma(S,m),s_m=i,j\in[k]^s}\prod_{i=1}^{m-1}c_{j_i}^\star(\langle\mathbf{w}_{s_i}^\star, \mathbf{w}\rangle)^{j_1}$$

$$\tag{142}$$

Each term of the form $\langle \mathbf{w}^\star_{s_i}, \mathbf{w} \rangle$ is uniformly bounded as $\frac{m^t_\times}{\sqrt{d^{\varepsilon_1}}}$. Since $|\Gamma(S, m), s_m = i| = \Theta(d^{(m-1)\varepsilon_1})$, we obtain:

$$|\langle \Delta_1, \mathbf{w}^\star_i \rangle| \leq \frac{1}{\sqrt{d^{\varepsilon_1}}} \sum_{j \in \mathbb{N}} c_j (m^t_\times)^j \tag{143}$$

for some constants $c_j$ with $\sup_j |c_j| < \infty$. Therefore a geometric-series bound (applicable since $m^t_\times < 1$) yields:

$$|\langle \Delta_1, \mathbf{w}^\star_i \rangle| \leq \frac{C}{\sqrt{d^{\varepsilon_1}}} m^t_\times, \tag{144}$$

for some contant $C > 0$. Summing the above bound over $i \in \sqrt{d^{\varepsilon_1}}$, results in the bound:

$$\|\Delta_1\|_2 \leq \frac{C}{\sqrt{d^{\varepsilon_1}}} m^t_\times \tag{145}$$

Next, for $\Delta_2$, we first apply the Hermite decomposition of $\tilde{\sigma}'$ to obtain:

$$\mathbb{E}\left[ r_m(x) \tilde{\sigma}'(\langle \mathbf{w}, \mathbf{x} \rangle) \mathbf{x} \right] = \sum_{j=1}^\infty \tilde{c}_j \mathbb{E}\left[ r_m(x) \mathrm{He}_{j-1}(\langle \mathbf{w}, \mathbf{x} \rangle) \mathbf{x} \right] \tag{146}$$

By Assumption 2, $\tilde{c}_1 = 0$ while $(ii)$ in Proposition 9 implies that the terms corresponding to $j > 2$ are bounded as $\frac{1}{\sqrt{d^{j\varepsilon_1}}} (m^t_\times)^j$

We therefore obtain:

$$\|\Delta_2\| \leq \frac{\tilde{C} m^t_\times}{\sqrt{d^{\varepsilon_1}}}, \tag{147}$$

for some constant $\tilde{C} > 0$. Lastly, Lemma 7 implies that:

$$|\langle \Delta_3, \mathbf{u}^\star \rangle| = \mathcal{O}_\prec \left( \frac{1}{d^{k\varepsilon + \delta}} \right). \tag{148}$$

Since $t < \tau^-_{\tilde{\delta}}$, the above bounds on $\Delta_1, \Delta_2, \Delta_3$ can be absorbed within arbitrarily small constants compared to $m$:

$$|\langle \Delta_1, \mathbf{u}^\star \rangle| + |\langle \Delta_2, \mathbf{u}^\star \rangle| + |\langle \Delta_3, \mathbf{u}^\star \rangle| \leq C m_t, \tag{149}$$

for arbitrarily small constant $C > 0$.

This results in the bound:

$$m^{t+1} \geq \frac{m^t + c \frac{\eta}{\sqrt{d^{\varepsilon_1}}} m^t - \eta \tilde{c} (m^t)^2}{\sqrt{1 + \eta^2 \|g^t\|^2}}, \tag{150}$$

where $\tilde{c} \leq c + \tilde{\varepsilon}$ for arbitrarily small $\tilde{\varepsilon}$. Next, we use the inequality $\sqrt{1+t}^{-1} \geq (1 - \frac{t}{2})$ for $t \geq 0$ to obtain:

$$\left( \sqrt{1 + \eta^2 \|g^t\|^2} \right)^{-1} \geq 1 - \frac{\eta^2}{2} \|g^t\|^2)$$
$$\geq 1 - \left( \frac{1}{2} \eta^2 c_g^2 m^t_2 \right), \tag{151}$$

where in the last line we applied the control over the squared gradient norm in Lemma 11 and $c_g < c$ can again be set arbitrarily close to $c$. Combining with Equation 150 yields part $(i)$.

Next, for part $(ii)$, introduce the operator:

$$P_{U^\perp} := (\mathbb{I} - (\mathbf{u}^\star)(\mathbf{u}^\star)^\top) P_{U^\star}, \tag{152}$$

corresponding to the projection onto the orthogonal complement of $\mathbf{u}^\star$ in $U^\star$. Let $\mathbf{u}^t_\perp := (\mathbb{I} - (\mathbf{u}^\star)(\mathbf{u}^\star)^\top) P_{U^\star} \mathbf{w}^{(t)}$.

$$\mathbf{g}_t P_{U^\perp} = \frac{(\mathbf{I} - \mathbf{w}^{(t)}(\mathbf{w}^{(t)})^\top)}{\sqrt{d^{\varepsilon_1}}} \mu_1 c_2^\star \tilde{c}_2 P_{U^\perp}(W^\star)^\top (W^\star)\mathbf{w}^t + (\mathbf{I} - \mathbf{w}^{(t)}(\mathbf{w}^{(t)})^\top)P_{U^\perp}\Delta_1 +$$
$$+ (\mathbf{I} - \mathbf{w}^{(t)}(\mathbf{w}^{(t)})^\top)P_{U^\perp}\Delta_2 + (\mathbf{I} - \mathbf{w}^{(t)}(\mathbf{w}^{(t)})^\top)P_{U^\perp}\Delta_3 \tag{153}$$

Since $\|P_{U^\perp}\mathbf{w}^t\| = m_\perp^t$ and $\|P_{U^\star}\mathbf{w}^t\| = m^t$ , the first term simplifies to:

$$\left\| \frac{(\mathbf{I} - \mathbf{w}^{(t)}(\mathbf{w}^{(t)})^\top)}{\sqrt{d^{\varepsilon_1}}} \mu_1 c_2^\star \tilde{c}_2 P_{U^\perp}(W^\star)^\top (W^\star)\mathbf{w}^t \right\| \leq \frac{\mu_1 c_2^\star \tilde{c}_2}{\sqrt{d^{\varepsilon_1}}} \left\| P_{U^\perp}(W^\star)^\top (W^\star)\mathbf{w}^t \right\|$$
$$- \frac{\mu_1 c_2^\star \tilde{c}_2}{\sqrt{d^{\varepsilon_1}}} \left\| (\mathbf{w}^{(t)}(\mathbf{w}^{(t)})^\top)P_{U^\perp}(W^\star)^\top (W^\star)\mathbf{w}^t \right\|$$
$$= cm_\perp^t - cm^t m_\perp^t.$$

By Equation 145, we have:

$$\left\| (\mathbf{I} - \mathbf{w}^{(t)}(\mathbf{w}^{(t)})^\top)P_{U^\perp}\Delta_1 \right\| \leq \|\Delta_1\| \leq \frac{C}{\sqrt{d^{\varepsilon_1}}} m_\times^t, \tag{154}$$

for some constant $C > 0$.

Similarly, by Equation 147, we obtain a bound:

$$\left\| (\mathbf{I} - \mathbf{w}^{(t)}(\mathbf{w}^{(t)})^\top)P_{U^\perp}\Delta_2 \right\| \leq \frac{\tilde{C}}{\sqrt{d^{\varepsilon_1}}} m_\times^t. \tag{155}$$

The above combine to result in the term $\tilde{\eta} C_2(m_\times^t)$ in Equation 126. Finally, by Equation 108 in Proposition 11, $\Delta_3 P_{U^\perp}$ is bounded as:

$$\left\| (\mathbf{I} - \mathbf{w}^{(t)}(\mathbf{w}^{(t)})^\top)P_{U^\perp}\Delta_3 \right\| = \mathcal{O}_\prec \left( \frac{1}{\sqrt{d^{1-\varepsilon_1+\tilde{\delta}}}} \right) \tag{156}$$

yielding the last term in Equation 126.

Analogously, part $(iii)$ follows by considering the terms $\langle \mathbf{w}_i^\star, \Delta_j \rangle, i \in \sqrt{d^{\varepsilon_1}}$ for $j = 1, 2, 3$ for $(iii)$ respectively, with the bound on $\|\mathbf{g}^t\|^2$ remaining the same.

## C.14 Feature Learning by the Second Layer

To motivate our setup for the training of $W_2$, we start with a heuristic discussion of the dynamics of gradient updates in the absence of pre-conditioning and projections. Throughout the subsequent discussions, we denote $n_2$ and $p_1$ by $n, p$ respectively. The presentation of formal results towards the proof of part $(ii)$ of Theorem 1 starts from Section C.18.

## C.15 Updates in Feature Space and Projection Onto the Kernel

Under correlation loss $\mathcal{L}_c = -f^\star(\mathbf{x})\hat{f}(\mathbf{x})$, the gradient update for a neuron $\mathbf{w}_{i,2}, i \in [p]$ in the second layer has the following form:

$$\mathbf{w}_{i,2}^{t+1} = \mathbf{w}_{i,2}^t - \eta \nabla_{W_2^t}\mathcal{L} = \mathbf{w}_{i,2}^t + \eta \sum_{\mu=1}^n (f^\star(\mathbf{x}_\mu))w_{i,3}\sigma'(\langle \mathbf{w}_{i,2}^t, \sigma(W_1(\mathbf{x})) \rangle)\sigma(W_1(\mathbf{x}_\mu)) \in \mathbb{R}^{p_1}.$$
$$\tag{157}$$

Under the approximation $\hat{f}(\mathbf{x}) \approx 0$, the updated pre-activation out of at a fixed input $\mathbf{x}$ are thus given by:

$$\langle \mathbf{w}_{i,2}^t, \sigma(W_1(\mathbf{x})) \rangle \approx \langle \mathbf{w}_{i,2}^t, \sigma(W_1(\mathbf{x})) \rangle + \eta \langle \sum_{\mu=1}^n f^\star(\mathbf{x}_\mu)a_i\sigma'(\langle \mathbf{w}_{i,2}^t, \sigma(W_1(\mathbf{x}_\mu)) \rangle)\sigma(W_1(\mathbf{x}_\mu)), \sigma(W_1(\mathbf{x})) \rangle$$

$$= \langle \mathbf{w}_{i,2}^t, \sigma(W_1(\mathbf{x})) \rangle + \eta \sum_{\mu=1}^n f^\star(\mathbf{x}_\mu)a_i\sigma'(\langle \mathbf{w}_{i,2}^t, \sigma(W_1(\mathbf{x}_\mu)) \rangle)\langle \sigma(W_1(\mathbf{x}_\mu)), \sigma(W_1(\mathbf{x})) \rangle.$$
$$\tag{158}$$

Letting $h_{2,i}^t(\mathbf{x}) := \langle \mathbf{w}_{i,2}^t, \sigma(W_1(\mathbf{x})) \rangle$, we obtain:

$$h_{2,i}^{t+1}(X) \approx h_{2,i}^t(X) + \eta w_{3,i} Z Z^\top (f^\star(X) \sigma'(h_{2,i}^t(X))), \tag{159}$$

with $X \in \mathbb{R}^{n_2 \times d}$ the data matrix and $Z$ denotes the feature-mapping $\sigma(W_1 X^\top)$.

We see that in the limit $n, p_1 \to \infty$, the above update results in a projection of $f^\star$ on the following Kernel (integral operator):

$$K_1(\mathbf{x}, \mathbf{x}') = \mathbb{E}_{\mathbf{w} \sim \mu_1} \left[ \sigma(\langle \mathbf{w}, \mathbf{x} \rangle) \sigma(\langle \mathbf{w}, \mathbf{x}' \rangle) \right], \tag{160}$$

where $\mu_1$ denotes the distribution of the rows of $W_1$ obtained upon feature learning in part $(i)$.

## C.16 The Role of Preconditioning

In light of Equation 159, we obtain a dynamics of the form:

$$h_{2,i}^{t+1}(\mathbf{x}) \approx h_{2,i}^t(\mathbf{x}) + \eta w_{3,i} \mathbb{E}_{\mathbf{x}'} \left[ K_1(\mathbf{x}, \mathbf{x}') f^\star(\mathbf{x}') \sigma'(h_{2,i}^t(\mathbf{x}')) \right] + \text{noise}, \tag{161}$$

where $K_1(\mathbf{x}, \mathbf{x}') f^\star(\mathbf{x}) \sigma'(h_{2,i}^t(\mathbf{x}))$ denotes the projection of $f^\star(\mathbf{x}) \sigma'(h_{2,i}^t(\mathbf{x}))$ onto the Kernel $K_1$. Through a central limit theorem-based heurestic, we expect the noise to be of order $\mathcal{O}(\frac{1}{\sqrt{n}} + \frac{1}{\sqrt{p}})$ [66]. However, the decay in $K_1$'s spectrum, entails that the degree-$k$ components in $K_1 f^\star(\mathbf{x}) \sigma'(h_{2,i}^t(\mathbf{x}))$ are of order $\mathcal{O}(\frac{1}{d^k})$. Comparing $\mathcal{O}(\frac{1}{\sqrt{n}} + \frac{1}{\sqrt{p}})$ and $\mathcal{O}(\frac{1}{d^k})$, one expects a sample-complexity of $d^{2k}$ for recovering $h^\star(\mathbf{x})$ through a single (non-preconditioned) gradient step. For quadratic features, this is precisely the sample-complexity obtained in [66].

A possible way to get around the additional sample complexity would be to re-use a single batch of size $\mathcal{O}(d^{k_\varepsilon})$ for up to $\mathcal{O}(d^{k_\varepsilon})$ steps, ensuring that the projection on the Kernel is well-approximated at each step while the number of steps are enough for the dynamics described by Equation 159 to approximate ridge-regression, which effectively has the same effect as pre-conditioning through the removal of the learned components. However, analyses of gradient descent with the re-use of batches for a large number of iterations is expected to be challenging due to the accumulation of additional correlations and memory terms [32].

Therefore, to allow a simplified "online" analysis we opt to include additional pre-conditioning in the updates, which effectively removes the extra $\frac{1}{d^{k_\varepsilon}}$ factor from Equation 159.

**Remark**: Under the additional assumption that $\mathbb{E}_{z \sim \mathcal{N}(0,1)} \left[ \sigma(z) z \right] = 0$, [66] improved the sample-complexity for recovery of quadratic features from $\mathcal{O}(d^4)$ to $\mathcal{O}(d^2)$. Such an assumption on $\sigma(\cdot)$ however, appears insufficient towards reducing the general $\mathcal{O}(d^{2k})$ sample complexity to $\mathcal{O}(d^k)$ for general degree $k$ components.

## C.17 Main Result for part (ii)

This section deals with the recovery of the non-linear features $h^\star(\mathbf{x})$.
**Theorem 6.** *Let $W_2^1$ denote the updated layer $2$ weights after a single pre-conditioned gradient step with batch-size $n_2$, with initialization $W_2^0 = \mathbf{0}_{p_2 \times p_1}$ as in Algorithm 1:*

$$W_2^1 = -\eta \left( \frac{1}{n} \sigma(W_1 X^\top)^\top (\sigma(W_1 X)^\top + \lambda_2 \right)^{-1} \nabla_{W_2} \mathcal{L}, \tag{162}$$

*where $W_1 \in \mathbb{R}^{p_1 \times d}$ denotes the updated weight matrix with independent rows obtained as per Theorem 5. The updated pre-activations $\mathbf{h}_2^1(\mathbf{x}) = W_2^1 \sigma(W_1 \mathbf{x})$ then satisfy:*

$$h_{2,i}^1(\mathbf{x}) = \eta w_{3,i} \sigma'(0) h^\star(\mathbf{x}) + r(\mathbf{x}), \tag{163}$$

*where $w_3$ is the readout scalar weight and the remainder $r(\mathbf{x})$ satisfies:*

$$r(\mathbf{x}) = \mathcal{O}_\prec \left( \frac{1}{\sqrt{d^{\min(\delta, \delta' - \delta)}}} \right). \tag{164}$$

## C.18 Structure of the Pre-conditioned Update

Let $Z$ denote the feature matrix $Z = \sigma(XW_1^\top)$ applied to an independent data-matrix $X \in \mathbb{R}^{n_2 \times d}$ using the updated weights $W_1$ obtained in part $(i)$. Throughout the section, we assume that the threshold parameter $\kappa > 0$ in Theorem 5 is fixed to some dimension-independent value and occasionally consider the limit $\kappa \to 0$ (but after $d \to 0$). Denote by $Z(\mathbf{x})$ the same mapping applied to a fixed point $\mathbf{x} \in \mathbb{R}^d$.

The proposition below expresses a pre-conditioned gradient update on $W_2$ as a "Kernel-ridge regression like" update to $h_2^t(\mathbf{x})$.

**Proposition 13.** *Suppose that $W_2$ is re-initialized to $\mathbf{0}$. The updated pre-activations $h_2^t(\mathbf{x})$ satisfy for $i \in [p_2]$:*

$$h_{2,i}^1(\mathbf{x}) = \frac{\eta}{n} w_{3,i} Z(\mathbf{x})^\top (\frac{1}{n} Z^\top Z + \lambda_2 I)^{-1} Z^\top (f^\star(X)) \sigma'(0). \tag{165}$$

## C.19 Decomposition into Radial and Spherical Kernels

Let $l_k^d, Q_k^d$ for $k \in \mathbb{N}$ denote the associated Laguerre and Gegenbauer polynomials in dimension $d$. Recall that $U^\star$ denotes the span of $W^\star$.

From Theorem 5, for any $i \in [p_1]$, the updated neuron $\mathbf{w}_i^1$ can be decomposed as:

$$\mathbf{w}_i^1 = \mathbf{u}_i + \mathbf{v}_i, \tag{166}$$

where $\|\mathbf{w}_i^1\| = 1$, $\mathbf{u}_i \in U^\star$ and $\mathbf{v}_i \in U_\perp^\star$.

For any $\mathbf{x} \in \mathbb{R}^d$, denote by $\mathbf{x}^\star, \mathbf{x}^\perp$, its components along $U^\star$ and $U_\perp^\star$ respectively.

Next, we decompose the inner-product $\langle \mathbf{w}_i, \mathbf{x} \rangle$ as:

$$\langle \mathbf{w}_i, \mathbf{x} \rangle = \|\mathbf{x}^\star\| \langle \mathbf{u}, \frac{\mathbf{x}^\star}{\|\mathbf{x}^\star\|} \rangle + \|\mathbf{x}^\perp\| \langle \mathbf{v}, \frac{\mathbf{x}_\perp}{\|\mathbf{x}_\perp\|} \rangle. \tag{167}$$

By the Gaussianity of $\mathbf{x}$, the random variables $\|\mathbf{x}^\star\|, \langle \mathbf{u}, \frac{\mathbf{x}^\star}{\|\mathbf{x}^\star\|} \rangle, \|\mathbf{x}^\perp\|, \langle \mathbf{v}, \frac{\mathbf{x}_\perp}{\|\mathbf{x}_\perp\|} \rangle$ are mutually independent. The variables $\|\mathbf{x}^\star\|^2$ and $\|\mathbf{x}_\perp\|^2$ are distributed as $\chi^2$ variables with $d^{\epsilon_1}$ and $d - d^{\epsilon_1}$ degrees of freedom respectively and hence admit the associated Laguerre polynomials as an orthonormal basis (Section C.3.2). Therefore, $\langle \mathbf{w}, \mathbf{x} \rangle$ admits an orthonormal basis given by the tensor product of associated Laguerre and Gegenbauer polynomials.

By expanding $\sigma$ along this bases of associated Laguerre and Gegenbauer polynomials, the activation $\sigma(\langle \mathbf{w}, \mathbf{x} \rangle + b)$ can then be decomposed as:

$\sigma(\langle \mathbf{w}_i, \mathbf{x} \rangle + b)$

$$= \sum_{k_1, j_1, k_2, j_2 = 0}^{\infty} a_{j_1, k_1, j_2, k_2}^d (b, \|\mathbf{u}_i\|, \|\mathbf{v}_i\|) l_{j_1}^{d^{\epsilon_1}} (\|\mathbf{x}^\star\|^2) l_{j_2}^{d - d^{\epsilon_1}} (\|\mathbf{x}_\perp\|^2) Q_{k_1}^{d^{\epsilon_1}} (d^{\epsilon_1} \langle \mathbf{u}, \frac{\mathbf{x}^\star}{\|\mathbf{x}^\star\|} \rangle) Q_{k_2}^{d - d^{\epsilon_1}} (d - d^{\epsilon_1} \langle \mathbf{v}, \frac{\mathbf{x}_\perp}{\|\mathbf{x}_\perp\|} \rangle),$$
$$\tag{168}$$

where $l_k^d, P_k^d$ denote the associated Laguerre and Gegenbauer polynomials in dimension $d$ respectively. The above convergence holds in $L_2$ w.r.t $\mathbf{x} \sim \mathcal{N}(0, I_d)$.

**Proposition 14.** *For all $k \in \mathbb{N}$:*

$$\lim_{\kappa \to 0} \lim_{d \to \infty} a_{k,0,0,0}^d(b) \sqrt{d^{\epsilon_1}}^k = \mu_k(b) \tag{169}$$

*Proof.* Let $d_\perp := d - d^{\epsilon_1}$, then:

$$\sigma(\langle \mathbf{w}_i, \mathbf{x} \rangle + b) = \sigma \left( (\sqrt{1 + \frac{\|\mathbf{x}\|_\star^2 - d^{\epsilon_1}}{d^{\epsilon_1}}}) \sqrt{d_1^{\bar{\epsilon}}} \langle u, \frac{\mathbf{x}^\star}{\|\mathbf{x}\|^\star} \rangle + \sqrt{1 + \frac{\|\mathbf{x}\|_\perp^2 - d_\perp}{d_\perp}} \sqrt{(d_\perp)} \langle \mathbf{v}, \frac{\mathbf{x}_\perp}{\|\mathbf{x}_\perp\|} + b \rangle \right) \tag{170}$$

Let $r^\star = \frac{\|\mathbf{x}\|_\star^2 - 1}{\sqrt{d}}$ and $r^\perp = \frac{\|\mathbf{x}\|_\perp^2 - 1}{\sqrt{d}}$. Subsequently, the result follows through the Taylor expansion $\sqrt{1+z} = 1 + \frac{z}{2} + o(z^2)$ w.r.t $r^\star$, while noting that $\sqrt{d_1^{\bar{\epsilon}}} \langle u, \frac{\mathbf{x}^\star}{\|\mathbf{x}\|^\star} \rangle \to \mathcal{N}(0, \tilde{\kappa})$ and $\sqrt{(d_\perp)} \langle \mathbf{v}, \frac{\mathbf{x}_\perp}{\|\mathbf{x}_\perp\|} \rangle \to \mathcal{N}(0, 1 - \tilde{\kappa})$ by Theorem 5 for some $\tilde{\kappa} > \kappa$.

The Taylor expansion implies that the coefficient of $r^\star$ converges to $\mu_k(b)\frac{1}{\sqrt{d^{\varepsilon_1}}^k}$. On the other hand, the coefficient must also equal $\sum_{j\geq k} a^d_{j,0,0,0} c_{jk}$, where $c_{jk}$ denotes the coefficient of $z^k$ in the $k_{th}$ associated Laguerre polynomials. $\square$

In light of the above decomposition, we introduce the following sequence of radial Kernels, with $k_1 = k_2 = j_2 = 0$:

$$K_0^d(\mathbf{x}_1, \mathbf{x}_2) = \sum_{j=0}^\infty \sum_{j'=0}^\infty \mathbb{E}\left[ a^d_{j,k,0,0}(b, \|\mathbf{u}_i\|, \|v_i\|) a^d_{j',k,0,0}(b, \|\mathbf{u}_i\|, \|\mathbf{v}_i\|) \right] l_j^{d^{\varepsilon_1}}(\|\mathbf{x}_1^\star\|^2) l_{j'}^{d^{\varepsilon_1}}(\|\mathbf{x}_2^\star\|^2)$$

(171)

**Proposition 15.** *Under Assumption 1, $h_{1,2}^\star$, $K_0^d$ admits uniformly continuous eigenfunctions $\phi_{1,d}, \phi_{2,d}$ with associated eigenvalues $\lambda_1 = \Theta(1), \lambda_2 = \Theta(\frac{1}{\sqrt{d^{\varepsilon_1}}})$ such that $r^\star(\mathbf{x}) = \frac{1}{\sqrt{d^{\varepsilon_1}}}(\|\mathbf{x}^\star\|^2 - 1)$ satisfies:*

$$r^\star(\mathbf{x}) = \mathcal{P}_{\text{span}\{\phi_{1,d},\phi_{2,d}\}} r^\star(\mathbf{x}) + \mathcal{O}(\frac{1}{\sqrt{d^{\varepsilon_1}}}).$$

(172)

where $\mathcal{P}$ denotes projection in $L^2(\mu(\mathbf{x}))$.

*Proof.* By proposition 14, $a_k^d$ converge a.s to deterministic limits $a_k^d$ as $d \to \infty$. We obtain the following limiting expression for $K_k(\mathbf{x}_1, \mathbf{x}_2)$:

$$K_0(\mathbf{x}_1, \mathbf{x}_2) = \mathbb{E}_b\left[a_0^2\right] + \mathbb{E}_b\left[a_0 a_1\right] l_1(\|\mathbf{x}_1^\star\|^2) l_1(\|\mathbf{x}_2^\star\|^2) + \mathbb{E}_b\left[a_0 a_1\right] l_0(\|\mathbf{x}_1^\star\|^2) l_1(\|\mathbf{x}_2^\star\|^2)$$
$$+ \mathbb{E}_b\left[a_0 a_1\right] l_1(\|\mathbf{x}_1^\star\|^2) l_0(\|\mathbf{x}_2^\star\|^2) + \cdots$$

(173)

Note that:

$$\mathbb{E}\left[l_0(\|\mathbf{x}_1^\star\|^2) K_0(\mathbf{x}_1, \mathbf{x}_2) l_0(\|\mathbf{x}_2^\star\|^2)\right] = \mathbb{E}_b\left[a_0^2\right] \xrightarrow[d\to\infty, \kappa\to 0]{} \mathbb{E}_b\left[\mu_0^2(b)\right].$$

(174)

By assumption on $\sigma$, $\mu_0^2(b)$ is analytic in $b$ and hence non-zero almost sure w.r.t $b \sim \mathcal{N}(0,1)$. Therefore, by the variational characterization of eigenvalues for compact self-adjoint operators [16], we obtain:

$$\lambda_1(K_0(\mathbf{x}_1, \mathbf{x}_2)) > \mathbb{E}_b\left[a_0(b) a_0(b)\right] - \mathcal{O}_\prec(\frac{1}{\sqrt{d}^{k\varepsilon_1}})$$

(175)

implying:

$$\lambda_1(K_0(\mathbf{x}_1, \mathbf{x}_2)) = \Theta_d(1),$$

(176)

Similarly, we obtain:

$$\mathbb{E}\left[l_1(\|\mathbf{x}_1^\star\|^2) K_0(\mathbf{x}, \mathbf{x}_2) l_1(\|\mathbf{x}_1^\star\|^2)\right] = \mathbb{E}_b\left[a_1^2\right] = \Theta(\frac{1}{d^{\varepsilon_1}}),$$

(177)

implying:

$$\lambda_2(K_0(\mathbf{x}, \mathbf{x}')) = \Theta_d(\frac{1}{d^{\varepsilon_1}}).$$

(178)

Analogously, since $\sigma$ is analytic, with probability 1, $a_{j,0} \neq 0, \forall j \in \mathbb{N}$, we obtain that the $j_{th}$ eigenvalue for $K_0(\mathbf{x}, \mathbf{x}')$ satisfies:

$$\lambda_j(K_0(\mathbf{x}, \mathbf{x}')) = \Theta_d(\frac{1}{d^{\frac{j}{2}\varepsilon_1}}).$$

(179)

Equation 172 then follows by noting that:

$$\mathbb{E}\left[r^\star(\mathbf{x}) K_0(\mathbf{x}, \mathbf{x}') r^\star(\mathbf{x}')\right] = \Theta_d(\frac{1}{d^{\varepsilon_1}})$$

(180)

The continuity of the eigenfunctions then follows since we have:

$$\phi_{j,d}(\mathbf{x}) = \mathbb{E}_{\mathbf{x}'}\left[K(\mathbf{x}, \mathbf{x}') \phi_{j,d}(\mathbf{x}')\right],$$

(181)

Since $K(\mathbf{x}, \mathbf{x}')$ is uniformly continuous in $\mathbf{x}$. $\square$

## C.20 Decomposition of the Feature Matrix

For $j_i, k_i \in \mathbb{N}$, let $\theta^d_{j_1,k_1,j_2,k_2}$ denote the eigenfunctions of the radial Kernel $K^d_k$ for $k \in \mathbb{N}$, defined as (Generalizing Equation 171):

$$K^d_{k_1,k_2}(\mathbf{x}_1,\mathbf{x}_2) = \sum_{j_1,j_1'}^{\infty}\sum_{j_2,j_2'}^{\infty} \mathbb{E}\left[a^d_{j_1,k_1,j_2,k_2}a^d_{j_1',k_1,j_2',k_2}\right] l^{d^{\varepsilon_1}}_{j_1}(\|\mathbf{x}_1^\star\|^2)l^{d-d^{\varepsilon_1}}_{j_2}(\|\mathbf{x}_1^\perp\|^2)l^{d^{\varepsilon_1}}_{j_1'}(\|\mathbf{x}_2^\star\|^2)l^{d-d^{\varepsilon_1}}_{j_2'}(\|\mathbf{x}_2^\perp\|^2)$$
(182)

Analogously, let $\kappa^d_{j_1,k_1,j_2,k_2}$ denote the eigenfunctions of the associated companion Kernel defined on the weights:

$$\mathcal{K}^d_{k_1,k_2}(\mathbf{w}_1,\mathbf{w}_2) = \sum_{j=0,j'=0}^{\infty} a^d_{j,k_1,j',k_2}(\mathbf{w}_1)a^d_{j,k_1,j',k_2}(\mathbf{w}_2)$$
(183)

Define:

$$\psi_{j_1,j_2,k_1,k_2}(\mathbf{x}) = \theta^d_{j_1,k_1,j_2,k_2}(r^\star,r_\perp)Y_k(\mathbf{x}^\star/r^\star)Y_k(\mathbf{x}_\perp/r_\perp).$$
(184)

And for the conjugate:

$$\phi_{j_1,j_2,k_1,k_2}(\mathbf{w}) = \kappa^d_{j_1,k_1,j_2,k_2}(b,\|u\|,\|v\|)Y_k(\mathbf{u}/\|\mathbf{u}\|)Y_k(\mathbf{v}/\|\mathbf{v}\|).$$
(185)

With a slight abuse of notation, we denote $\theta^d_{j,0,0,0}$ and $\kappa^d_{j,0,0,0}$ by $\theta^d_j$ and $\kappa^d_j$ respectively. These correspond to eigenvalues for the zeroth-order radial Kernel along $U^\star$.

We partition the indices $j_1, j_2, k_1, k_2$ into three disjoint sets:

$$\mathcal{S}_1 = \{j_1, j_2, k_1, k_2 : k_1 = j_2 = k_2 = 0, j_2 \le 2k\} \cup \{j_1, j_2, k_1, k_2 : j_1 = j_2 = k_2 = 0, k_1 \le k\}$$
$$\mathcal{S}_2 = \{j_1, j_2, k_1, k_2 : j_2, 2j_1 + 2j_2/\varepsilon_1 + k_1 + k_2/\varepsilon_1 = k\}\,\mathcal{S}_1$$
$$\mathcal{S}_3 = \{j_1, j_2, k_1, k_2, \in \mathbb{N}^4\}\backslash(\mathcal{S}_1 \cup \mathcal{S}_2).$$

The above partitioning is motivated as follows:

(i) $\mathcal{S}_1$ corresponds the set of eigenfunctions whose projections can be approximated via $Z$ with $n_2, p_2 = \Theta(d^{k\varepsilon_1})$ and are relevant towards learning $f^\star(\mathbf{x})$.

(ii) $\mathcal{S}_2$ corresponds to the set of eigenfunctions whose projections can be approximated by $Z$ but do not contribute to the learning of $f^\star(\mathbf{x})$.

(iii) $\mathcal{S}_3$ corresponds to the high-degree set of eigenfunctions for which the number of samples, neurons $n_2, p_2$ are insufficient towards being approximated through $Z$.

Let $\Theta^d_j, \mathfrak{K}^d_j$ denote matrices with rows $\theta^d_{j_1,k_1,j_2,k_2}(r^\star,r_\perp)$ and $\kappa^d_{j_1,k_1,j_2,k_2}(b,\|\mathbf{u}\|,\|\mathbf{v}\|)$ respectively. Similarly, let $\Psi_{j_1,j_2,k_1,k_2}(X), \Phi_{j_1,j_2,k_1,k_2}(W)$ denote matrices with rows $\psi_{j_1,j_2,k_1,k_2}(\mathbf{x})$ and $\phi_{j_1,j_2,k_1,k_2}(\mathbf{x})$

Expressing Equation 168 in matrix form and applying Proposition 3 to expand each term $Q^d_k(\cdot)$, we obtain the following decomposition:

$$Z = \sum_{j=1}^{2k}\Theta^d_j(\mathbf{r}^\star)D^r_j(\mathfrak{K}^d_j(\mathbf{b},\|\mathbf{u}\|))^\top + \sum_{j=1}^{k}Y_j(X^\star)D^{\mathcal{S}_1}_j Y_j(U)$$

$$+ \sum_{j_1,j_2,k_1,k_2\in\mathcal{S}_2}\Psi_{j_1,j_2,k_1,k_2}(X)D^{\mathcal{S}_2}_{j_1,j_2,k_1,k_2}\Phi_{j_1,j_2,k_1,k_2}(W)^\top$$
(186)

$$+ \sum_{j_1,j_2,k_1,k_2\in\mathcal{S}_3}\Psi_{j_1,j_2,k_1,k_2}(X)D^{\mathcal{S}_3}_{j_1,j_2,k_1,k_2}\Phi_{j_1,j_2,k_1,k_2}(W)^\top,$$

where $D^r_j, D^s_j$ denote diagonal matrices with entries $(b^d_j)^2, (a^d_j)^2$ respectively. We denote the above three-components corresponding to $\mathcal{S}_1, \mathcal{S}_2, \mathcal{S}_3$ as $Z_1, Z_2, Z_3$ respectively.

### C.21 Approximation of Eigenfunctions

Let $M = |S_1| \cup |S_2|$. Since $B(d, k) = \Theta(d^k)$ (section C.3.2), we obtain $M = \Theta(d^{k\varepsilon})$. We next show that the above partitioning of eigenfunctions translates to a "spike"+bulk structure for $Z$, with the spikes arising from components corresponding to $S_1, S_2$ allowing the reconstruction of the corresponding eigenfunctions through the sample-covariance. The higher-degree components $S_3$, on the other hand, coalesce into a bulk.

For each $\mathbf{x} \in \mathbb{R}^d$, let $\psi_{S_1 \cup S_2}(\mathbf{x}) \in \mathbb{R}^M$ denote the combined vector of components along eigenfunctions indexed by $S_1, S_2$, i.e:

$$\psi_{S_1 \cup S_2}(\mathbf{x}) := \big( \big( \psi^d_{j_1, j_2, k_1, k_2}(\mathbf{x}) \big)_{j_1, j_2, k_1, k_2 \in S_1 \cup S_2} \big). \tag{187}$$

Analogously, define:

$$\phi_{S_1 \cup S_2}(\mathbf{w}) := \big( \big( \phi^d_{j_1, j_2, k_1, k_2}(\mathbf{w}) \big)_{j_1, j_2, k_1, k_2 \in S_1 \cup S_2} \big). \tag{188}$$

These properties are summarized in the following proposition, which constitutes the central result of this section:

**Proposition 16.** *There exists a sequence $c_d$ with $c_d = \mathcal{O}(1)$ such that*

*(i)*

$$\left\| \frac{1}{n} \sum_{\mu=1}^n \psi_{S_1 \cup S_2}(\mathbf{x}_\mu) \psi_{S_1 \cup S_2}(\mathbf{x}_\mu)^\top - \mathbb{I}_M \right\| = \mathcal{O}_{\prec}(\frac{1}{d^{\frac{\delta}{2}}}) \tag{189}$$

*(ii)*

$$\left\| \frac{1}{p} \sum_{i=1}^p \phi_{S_1 \cup S_2}(\mathbf{w}_i) \phi_{S_1 \cup S_2}(\mathbf{w}_i))^\top - \mathbb{I}_M \right\| = \mathcal{O}_{\prec}(\frac{1}{d^{\frac{\delta}{2}}}) \tag{190}$$

*(iii)*

$$\frac{1}{p} Z_3 Z_3^\top = c_d \mathbb{I}_d + \mathcal{O}_{\prec}(\frac{1}{d^{\min(\frac{\delta' - \delta}{2}, \frac{\delta}{2})}}). \tag{191}$$

*(iv)*

$$\| Z_3 \Phi_{S_1 \cup S_2}(W) \|_F = \mathcal{O}_{\prec}(\frac{p_2}{d^{\frac{\delta}{2}}}) \tag{192}$$

Before proceeding with the proof, we highlight the key-takeaways from the above result. Points $(i), (ii)$ imply that the matrices $Z_1, Z_2$ contribute $M$ spikes to $Z$ with left, right singular vectors aligned with $\psi_{S_1 \cup S_2}(\mathbf{x})$ and $\phi_{S_1 \cup S_2}(\mathbf{w})$ respectively. Points $(ii), (iv)$ imply that the high-degree components $Z_3$ contribute an approximately isotropic bulk, that doesn't interfere with the spikes along $\phi_{S_1 \cup S_2}(\mathbf{w})$. Note that $(iv)$ is necessary since the large rank of $Z_3$ could cause the corresponding components to collectively interfere with the low-degree components.

The crucial consequence is that the spikes in $Z_1, Z_2$ allow effective reconstruction of the components along $S_1 \cup S_2$. In contrast, the failure of $Z_3$ to estimate the covariance structure along $S_3$ prevents the recovery of such high-degree components.

*Proof.* Equation 189 is a direct consequence of Lemma 2 and the hyper-contractivity of the spherical measure. Equation 190 however, requires additional control over the error in $\mathbf{w}$.

We start with showing that the covariance is well-approximated in expectation Let $\mathbf{v} \in \mathbb{R}^n, \|\mathbf{v}\| = 1$ denote an arbitrary fixed unit vector. Then:

$$\mathbf{v}^\top \mathbb{E} \big[ \phi_{S_1 \cup S_2}(\mathbf{w}) \phi_{S_1 \cup S_2}(\mathbf{w})^\top - \mathbb{I}_M \big] \mathbf{v} = \mathbb{E} \left[ \sum_{s \in S_1 \cup S_2} v_s^2 \phi_i^2(\mathbf{w}) \right] - 1, \tag{193}$$

since $\psi_i^2$ are uniformly lipschitz on $S_d$, applying a taylor expansion on $\mathbf{w}$ around $\mathbf{u}^\star$ yields:

$$\mathbb{E} \left[ \sum_{s \in S_1 \cup S_2} v_s^2 \psi_s^2(\mathbf{w}) \right] = \mathbb{E} \left[ \sum_{s \in S_1 \cup S_2} v_s^2 \psi_s^2(\tilde{\mathbf{w}}) \right] + \mathcal{O}(\frac{1}{d^\delta}) \tag{194}$$

where we used that $h_v(\mathbf{w}) = \mathbb{E}\left[\sum_{s \in s_1 \cup s_2} v_s^2 \phi_i^2(\mathbf{w})\right]$ is an even polynomial in $\mathbf{w}$. Therefore, $\mathbb{E}\left[\nabla h_v(\mathbf{w})\right] = 0$ while $\left\|\mathbb{E}\left[\nabla^2 h_v(\mathbf{w})\right]\right\| \leq C$ for some constant $C > 0$. Corollary 1 then ensures that the second order-term is bounded as $\mathcal{O}(\frac{1}{d^\delta})$. Taking supremum over $v$ for $\|v\| = 1$, we obtain:

$$\left\|\mathbb{E}\left[\phi_{S_1 \cup S_2}(\mathbf{w})\phi_{S_1 \cup S_2}(\mathbf{w})^\top - \mathbb{I}_M\right]\right\| = \mathcal{O}(\frac{1}{d^\delta}), \tag{195}$$

for some $\delta > 0$. We move on to establishing the concentration of $\Phi_{S_1}(\mathbf{w})$. By Equation 28 in [63], spherical harmonics $Y_{m,k}$ of degree $k \in \mathbb{N}$ admit a basis with the following representing along the cartersian coordinates:

$$Y_\alpha(\mathbf{w}) = C_\alpha^{1/2} h_\alpha(w_1, w_2) \prod_{j=1}^{d-2} \left\{ (w_1^2 + \cdots + w_{d-j+1}^2)^{\alpha_j/2} \tilde{Q}_{\alpha_j}^{(d_j)} \left( \frac{w_{d-j+1}}{\sqrt{w_1^2 + \cdots + w_{d-j+1}^2}} \right) \right\}, \tag{196}$$

where $\alpha \in \mathbb{N}^d$ contains at-most $\ell$-non-zero entries. Therefore, $Y_\alpha(\mathbf{w})$ is a polynomial in at-most $\ell$ coordinates in $\mathbf{w}$ along with the $\ell$ projection norms $r_j = \sqrt{\sum_{i=1}^j w_j^2}$. Applying part $ii, c$ of Theorem 5 then implies that:

$$\sup_{i \in d^{\varepsilon_1}} \left| \frac{1}{i} \sum_{j=1}^i (\langle \mathbf{w}_\kappa^\tau, \mathbf{w}_i^\star \rangle)^2 \right| = \mathcal{O}_\prec(\frac{1}{d^{\varepsilon_1}}). \tag{197}$$

Subsequently:

$$|Y_\alpha^{d^{\varepsilon_1}}(\mathbf{u}_i) - Y_\alpha^{d^{\varepsilon_1}}(\tilde{\mathbf{u}}_i^\star)| = \mathcal{O}_\prec(\frac{1}{d^\delta}). \tag{198}$$

and:

$$|Y_\alpha(\mathbf{u}_i) - Y_\alpha(\tilde{\mathbf{u}}_i^\star)| = \mathcal{O}_\prec(\frac{1}{d^\delta}). \tag{199}$$

Taking a union bound over the $\Theta(d^{k\varepsilon_1})$ values of $\alpha$ yields:

$$\mathbb{E}\left[\max_\alpha Y_\alpha^2(\mathbf{w})\right] = \tilde{\mathcal{O}}(1). \tag{200}$$

For the radial components recall that $\|u\|, \|v\| = \mathcal{O}_\prec(1)$ while the radial eigenfunctions are continuous.

We conclude that:

$$\mathbb{E}\left[\max_{i \in [n]} \|\psi_{S_1 \cup S_2}\|^2\right] = \tilde{\mathcal{O}}(M) \tag{201}$$

Setting $\tilde{\delta} < \delta, \delta'$ and recalling that, $n_2 = \Theta(d^{k\varepsilon_1 + \delta}), p_2 = \Theta(d^{k\varepsilon_1 + \delta'})$ while $|S_1 \cup S_2| = \Theta(d^{k\varepsilon})$, we may apply Lemma 2 to obtain:

$$\left\|\phi_{S_1 \cup S_2}(\mathbf{w})\phi_{S_1 \cup S_2}(\mathbf{w})^\top - \mathbb{E}\left[\phi_{S_1 \cup S_2}(\mathbf{w})\phi_{S_1 \cup S_2}(\mathbf{w})^\top\right]\right\| = \mathcal{O}_\prec(\frac{1}{d^{\frac{\delta}{2}}}) \tag{202}$$

where we absorbed the $d^{\tilde{\delta}}$ factor into the $\frac{1}{p}$ factor in the bound in Lemma 2 (Equation 63).

The proof of $(iii)$ similarly follows from Propositions 4, 8 in [60]. We outline the central steps. First, via the expansion of $\sigma(\cdot)$ given by Equation 168, for any $\mathbf{x}$, $\Psi_{S_3}(\mathbf{x})^\top \Phi_{S_3}(\mathbf{w})$ can be expressed as $\sigma(\langle \mathbf{w}, \mathbf{x} \rangle + b) - \Psi_{S_1 \cup S_2}(\mathbf{x})^\top \Phi_{S_1 \cup S_2}(\mathbf{w})$. Through Equation 196, $\Psi_{S_3}(\mathbf{x})^\top \Phi_{S_3}(\mathbf{w})$ therefore depends on $\mathbf{w}$ only through a finite number of coordinates in $U^\star$. Analogous to Equation 202 above, applying Lemma 2 and using $p >> n$, we obtain that:

$$\left\|\frac{1}{p} Z_3 Z_3^\top - G_3(X, X)\right\| = \mathcal{O}_\prec(\frac{1}{d^{\delta' - \delta}}) \tag{203}$$

where $G_3(X, X)$ denotes the gram-matrix associated to the Kernel:

$$K_3(\mathbf{x}, \mathbf{x}') = \sum_{s \in S_3} \lambda_s \psi_s(\mathbf{x}) \psi_s(\mathbf{x}'), \tag{204}$$

applied to the data-matrix $X \in \mathbb{R}^{n \times d}$.

The gram-matrix $G_3(X^\star, X^\star)$ now corresponds exactly to the spherical distribution on $U^\star$, with decay identical to the case of spherical data in [58]. Therefore, proposition 8 in [58] applies, which entails that the off-diagonal contributions from $G_3(X^\star, X^\star)$ are neglible in operator norm. This results in the bound:

$$\|G_3(X^\star, X^\star) - c_d \mathbf{I}\| = \mathcal{O}_{\prec}(\frac{1}{d^\delta}) \tag{205}$$

for some constant $c > 0$, implying $(iii)$ and $(iv)$. □

## C.22 Properties of the Feature-covariance Matrix

Having established Proposition 16 and the concentration of the top eigenvectors, the setting of $Z$ is now reduced to the spike + "bulk" structure in the proof of Theorem 1 in [60] with $\Theta(d^{k \varepsilon_1})$ spikes arising from the eigenfunctions $S_1, S_2$ corresponding to near-identity sample-covariances and a remaining bulk with uniformly-bounded operator norm. A consequence of such a structure is that the top singular vectors of $Z$ align closely with these "spikes". This ensures that projections onto $Z$ "reproduce" functions in $S \cup S_2$

Therefore, the proofs of Propositions 6, 7 in [60], based on perturbation inequalities for singular values, singular vectors, result in the following estimates for $Z$:

**Proposition 17.** $\frac{1}{\sqrt{p}} Z$ admits a singular value decomposition

$$\frac{1}{\sqrt{p}} Z = U_1 S_1 V_1^\top + U_2 S_2 V_2^\top + U_3 S_3 V_3^\top, \tag{206}$$

such that:

    (i) $\sigma_{min}(S_1 \cup S_2) = \Theta_d(1)$

    (ii) $\|S_3 - c_3 \mathbf{I}\| = o_{d,p}(1)$, for some constant $c_3 > 0$.

    (iii) $\Psi_{S_1 \cup S_2}^\top U_3 = o_d(\sqrt{n})$ and $\Phi_{S_1 \cup S_2}^\top V_3 = o_d(\sqrt{p})$

The proof of the above Proposition follows directly through Proposition 6 in [60]. The above result exactly charaterizes the projections of functions onto pre-conditioned features:

**Proposition 18.** For any $g : \mathbb{R}^d \to \mathbb{R}$ such that the projections onto radial components of degree $> 2$ are $o_d(1)$, for any $\lambda_2 = \Theta(\frac{p}{n})$:

$$Z(\mathbf{x})(\frac{1}{n} Z^\top Z + \lambda_2 I)^{-1} Z^\top g(X) = \mathcal{P}_{S_1} g(\mathbf{x}) + \mathcal{O}_{\prec}(\frac{1}{d^{\delta' - \delta}}). \tag{207}$$

The proof of part $(ii)$ of Theorem 1 is then completed by showing that under Assumption 1, the projection onto $S_1$ is exactly along $h^\star(\mathbf{x})$:

**Proposition 19.** Under Assumption 1:

$$\mathcal{P}_{S_1} g(\mathbf{x}) = \mu_1 h^\star(\mathbf{x}) + o_d(1). \tag{208}$$

*Proof.* By Assumption 3 and the composition of Hermite decompositions (Lemma 9), the non-vanishing terms along the radial component $\|x\|^2 - 1$ consists of total input-degree-2 and $2k$ while the remaining terms on the complement of $h_\ell^\star$ have degree at least $3(k+1) > k$. $S_1$ therefore consists exactly of the subspace with effective degree $k$. □

## C.23 Proof of Proposition 18

Let $\hat{g}(\mathbf{x}) = Z(\mathbf{x})^\top (\frac{1}{n} Z^\top Z + \lambda_2 I)^{-1} Z^\top g(X)$. Proposition 18 is equivalent to $\|\hat{g}(\mathbf{x}) - g(\mathbf{x})\|_2^2 = o_d(1)$. Expanding, we obtain:

$$\|\hat{g}(\mathbf{x}) - \mathcal{P}_k g(\mathbf{x})\|_2^2 = \|g(\mathbf{x})\|^2 - 2\langle \hat{g}(\mathbf{x}), \mathcal{P}_{S_1} g(\mathbf{x}) \rangle + \|\hat{g}(\mathbf{x})^2\|. \tag{209}$$

It therefore suffices to show that:

$$\langle \hat{g}(\mathbf{x}), \mathcal{P}_{S_1} g(\mathbf{x}) \rangle \to \|\mathcal{P}_{S_1} g(\mathbf{x})\|^2, \tag{210}$$

and:

$$\left\| \hat{g}(\mathbf{x})^2 \right\| \to \left\| \mathcal{P}_{S_2} g(\mathbf{x}) \right\|^2 . \tag{211}$$

Let $g_{S_1}$ denote the vector with components:

$$g_{S_1} := \left[ \mathbb{E}\left[ g(\mathbf{x}) \Psi_s(\mathbf{x}) \right] : s \in S_1 \right], \tag{212}$$

Let $\Lambda_{\leq 2, k}$ denote the diagonal matrix with the corresponding eigenvalues.

Then the above terms can be expressed as:

$$\langle \hat{g}(\mathbf{x}), \mathcal{P}_k g(\mathbf{x}) \rangle = g_{S_1} D_{S_1} \Phi_{S_1}^\top (\frac{1}{n} Z Z^\top + \lambda_2 I)^{-1} Z^\top g(X), \tag{213}$$

and:

$$\left\| \hat{g}(\mathbf{x})^2 \right\| = g(X)^\top Z (\frac{1}{n} Z Z^\top)^{-1} \Sigma (\frac{1}{n} Z Z^\top + \lambda_2 I)^{-1} Z^\top g(X), \tag{214}$$

where $\Sigma$ denotes the feature covariance:

$$\Sigma = \frac{1}{p} \mathbb{E}\left[ Z(\mathbf{x}) Z(\mathbf{x}))^\top \right] = \frac{1}{p} \sum_{j_1, j_2, k_1, k_2} \Phi_{j_1, j_2, k_1, k_2}(W) \Phi_{j_1, j_2, k_1, k_2}(W)^\top \tag{215}$$

To compute the above terms, we use Proposition 17 to estimate certain intermediate quantities similar to Proposition 7 in [60]:

**Proposition 20.** *Under the setup of Theorem 1, with the decomposition of eigenfunctions specified by Equation 186 :*

(i)

$$\psi_{S_1}^\top Z (\frac{1}{n} Z^\top Z + \lambda_2 I)^{-1} \phi_{S_1} D_{S_1} = \mathbb{I}_{m_1} + \mathcal{O}_\prec (\frac{1}{d^{\frac{\delta}{2}}}) \tag{216}$$

$$D_{S_1} \phi_{S_1}^\top Z (\frac{1}{n} Z^\top Z + \lambda_2 I)^{-1} Z^\top \frac{1}{p_2} f_{S_3} = \mathcal{O}_\prec (\frac{1}{d^{\frac{\delta}{2}}}) \tag{217}$$

$$\left\| \psi_{S_1}^\top Z (\frac{1}{n} Z^\top Z + \lambda_2 I)^{-1} \phi_{S_1} D_{S_1} \right\| = \mathcal{O}_\prec (\frac{1}{d^{\frac{\delta}{2}}}) \tag{218}$$

(ii)

$$\left\| \Psi_{S_2} f^\star(X) \right\| = \mathcal{O}_\prec (\frac{1}{d^{\frac{\delta}{2}}}) \tag{219}$$

Under the above proposition, the terms given by Equations 213, 214 simplify as follows:

$$g_{S_1} D_{S_1} \Phi_{S_1}^\top (\frac{1}{n} Z^\top Z + \lambda_2 I)^{-1} Z^\top g(X) = g_{S_1} D_{S_1} \Phi_{S_1}^\top (\frac{1}{n} Z^\top Z + \lambda_2 I)^{-1} Z_{1,2}^\top g_{1,2}(X)$$
$$+ g_{S_1} D_{S_1} \Phi_{S_1}^\top (\frac{1}{n} Z^\top Z)^{-1} Z_{1,2}^\top g_3(X) + g_{S_1} D_{S_1} \Phi_{S_1}^\top (\frac{1}{n} Z^\top Z + \lambda_2 I)^{-1} Z_3^\top g(X)$$

By Equation 216, the first term converges to $\left\| \mathcal{P}_{S_1} g(\mathbf{x}) \right\|^2$ while the other two terms are bounded as $\mathcal{O}_\prec (\frac{1}{d^\delta})$ by Equations 217,218 respectively.

Similarly,

$$g(X)^\top Z (\frac{1}{n} Z Z^\top)^{-1} \Sigma (\frac{1}{n} Z Z^\top)^{-1} Z^\top g(X) = g(X)^\top Z (\frac{1}{n} Z Z^\top)^{-1} \Sigma_{1,2} (\frac{1}{n} Z Z^\top)^{-1} Z^\top g(X)$$
$$+ g(X)^\top Z (\frac{1}{n} Z Z^\top)^{-1} \Sigma_3 (\frac{1}{n} Z Z^\top)^{-1} Z^\top g(X)$$

By Equation 218, the second term is bounded as $\mathcal{O}_\prec(1)$. This completes the proof of part $(ii)$ of Theorem 5.

## C.24 Proof of part (iii): Fitting the Target

Upon the completion of part $(ii)$, the second-layer pre-activations $h_2(\mathbf{x}) = W_2\sigma(W_1\mathbf{x})$ are approximately equivalent to those of a random feature-mapping applied to the scalar input $h^\star(\mathbf{x})$, with the random weights of the feature mapping given by $\tilde{w} = cw_3$, with $c = \eta\sigma'(0)$ as in Proposition 13. Hence, we introduce the Kernel $K(\cdot,\cdot) : \mathbb{R}^2 \to \mathbb{R}$:

$$K(z_1, z_2) := \mathbb{E}_{w \sim \mathcal{N}(0,1)}\left[\sigma(cwz_1 + b)\sigma(cwz_2 + b)\right]. \tag{220}$$

For $Z \in \mathbb{R}^n$, we further denote by $K(Z, Z)$ the corresponding gram-matrix $K(Z, Z) \in \mathbb{R}^{n \times n}$, with entries

$$K(Z, Z)_{i,j} = K(z_i, z_j). \tag{221}$$

Let $\mathcal{H}_K$ denote the RKHS corresponding to the Kernel $K$. Let $H^\star \in \mathbb{R}^{n \times p_1}$ further denote the matrix with rows $h^\star(\mathbf{x}_\mu)$.

Since the moments of $h^\star(\mathbf{x})$ are uniformly bounded in $d$, we obtain:
**Proposition 21.** *[[25]] For any $\delta > 0$, and large enough $d$, $\exists$ constants $c, C$ such that with $\lambda = \Theta(\sqrt{n})$:*

$$\left\|k(h^\star, H^\star)(\frac{1}{n}K(H^\star, H^\star) + \lambda\mathbb{I})^{-1}\frac{1}{\sqrt{n}}g^\star(H^\star) - g^\star(h^\star)\right\|_2^2 - \inf_{f \in \mathcal{H}_K}\left[\|f - g^\star(h^\star)\|_2^2 + \lambda\|f\|_{\mathcal{H}_K}\right]$$
$$\leq C\frac{N_K(\lambda)\log(\frac{1}{\delta})^c}{n}, \tag{222}$$

*where $H^\star \in \mathbb{R}^N$ contains independent samples $h^\star(\mathbf{x})$, and $N_K(\lambda)$ denotes the "effective-dimension":*

$$N_K(\lambda) = \mathrm{Tr}[(K + \lambda)^{-1}K], \tag{223}$$

*which admits the following trivial bound:*

$$N_K(\lambda) \leq \frac{\mathrm{Tr}[K]}{\lambda} \tag{224}$$

We next translate the above bound into generalization error through a control of the approximation error term.

Note that the uniform bounds on the moments of $h^\star(\mathbf{x})$ and Markov's inequality, for any $\varepsilon > 0$, $\exists R_\varepsilon > 0$ such that for large enough $d$:

$$\Pr[h^\star(\mathbf{x}) \notin B_{R_\varepsilon}] \leq \varepsilon \tag{225}$$

Next, define the following class of functions:

$$\mathcal{F}_\varepsilon = \{f \in \mathcal{H}_K : \mathrm{supp}(f) \in B_{R_\varepsilon}\}. \tag{226}$$

Then:

$$\inf_{f \in \mathcal{H}_K}\left[\|f - g^\star(h^\star)\|_2^2 + \lambda\|f\|_{\mathcal{H}_K}\right] \leq \inf_{f \in \mathcal{F}_\varepsilon}\left[\|f - g^\star(h^\star)\|_2^2 + \lambda\|f\|_{\mathcal{H}_K}\right] \tag{227}$$

Restricted to the compact set $B_{R_\varepsilon}$, universality of random feature Kernels with non-polynomial, polynomially-bounded activations [78] implies that for any $\varepsilon > 0$, $\exists f_\varepsilon \in \mathcal{H}_K$ such that:

$$\inf_{f \in \mathcal{F}_\varepsilon}\left[\|f - g^\star(h^\star)\mathbf{1}_{h^\star \in B_{R_\varepsilon}}\|_2^2 + \lambda\|f\|_{\mathcal{H}_K}\right] \leq \varepsilon + \lambda\|f_\varepsilon\|_{\mathcal{H}_K}^2. \tag{228}$$

Therefore, by setting $\lambda$ small enough such that:

$$\lambda_\varepsilon\|f_\varepsilon\|_{\mathcal{H}_K}^2 \leq \varepsilon, \tag{229}$$

we otbain:

$$\inf_{f \in \mathcal{F}_\varepsilon}\left[\|f - g^\star(h^\star)\|_2^2 + \lambda\|f\|_{\mathcal{H}_K}\right] \leq 2\varepsilon + \|g^\star(h^\star)\mathbf{1}_{h^\star \notin B_{R_\varepsilon}}\|_2^2. \tag{230}$$

By Cauchy-Shwartz and the uniform bound on $\mathbb{E}\left[g^\star(h^\star)^2\right]$, the last term in the RHS is bounded by $C\varepsilon$ for some cosntant $C > 0$.

Subsequently, we may set $n$ in Proposition 21 large enough such that:

$$C\frac{\text{Tr}[K]\log(\frac{1}{\delta})^c}{\lambda n} \leq \varepsilon, \tag{231}$$

Implying that for small enough $\lambda(\varepsilon)$ and large enough $n(\varepsilon, \delta)$, with probability $1 - \delta$:

$$\left\|k(h^\star, H^\star)(K(H^\star, H^\star) + \lambda\mathbb{I})^{-1}g^\star(H^\star) - g^\star(h^\star)\right\|_2^2 \leq C\varepsilon, \tag{232}$$

for some constant $C > 0$.

Now, returning to the true features $h^2(\mathbf{x})$, it remains to combine the above estimate with the concentration of the gram-matrix to the associated Kernel. This is established similar to the proof of Proposition 16 through Lemma 2.

Note that the above argument does not yield the dependence of $\lambda, n$ on $\varepsilon$. Such an explicit dependence requires finer control on the approximation, source terms. For such an analysis, we refer to the explicit rademacher complexity based bounds for ReLU activation in [29].

We remark that more quantitave estimates can be obtained through rademacher-complexity based analysis for specification activations such as Relu [29].

### C.25 Relaxing Assumption 3

In this section, we adress the requirement of Assumption 3 and steps towards relaxing it. Assumption 3 simplifies our analysis by ensuring that $\mathcal{P}_{S_1}f^\star(\mathbf{x})$ is exactly $h^\star(\mathbf{x})$ arising from the first-order Hermite coefficient of $g^\star(\mathbf{x})$. In general, however, the degree-$k$ approximation of $f^\star(\mathbf{x})$ may contain additional components involving higher-degree dependence on $h^\star(\mathbf{x})$. For instance, if $g^\star(\mathbf{x})$ has a non-zero second-order Hermite coeffient, then Lemma 9 implies that $\text{He}_2(h^\star(\mathbf{x}))$ can be decomposed into components of Hermite degree $4, \cdots, 2k$. Therefore, if $k \geq 4$, gradient updates to $W_2$ result in $h_2(\mathbf{x}) \approx c(h^\star + \text{higher order components})$. While ideally one would hope that the learning of such additional components would only help towards fitting $f^\star(\mathbf{x})$ by $w_3$, this would require the second-layer pre-activations to disentangle $h^\star$ and the remaining components i.e. to specialize across non-linear features. Analysis of such a specialization remains challenging due to the reasons described in Appendix 2. Therefore, relaxing Assumption 3 requires going beyond the single-spike ($r = 1$) non-linear feature learning.

Additionally, as we saw through the decomposition of the activation into radial and spherical arguments (Equation 168), the radial components exhibit slower-decay w.r.t the degree. Therefore, $d^{k\varepsilon}$ samples, neurons suffice towards learning degree-$k$ components on $\frac{1}{\sqrt{d^{\varepsilon_1}}}\|\mathbf{x}^\star\|^2$ which correspond to degree $2k$ components on $\mathbf{x}$. We believe this to be an artifact of our choice $a^\star = 1$, which leads to a special dependence along the radial component. Going beyond the isotropic $a^\star = 1$ setting is however, challenging due to our reliance on diagonalization of the associated Kernel along a fixed basis.

## D   Deeper networks: Proof of Theorem 2

### D.1   Independence of features

The independence of $h^\star(\mathbf{x})$ follows by noting that by induction, for all $\ell \in [L]$, distinct components of $\mathbf{h}_\ell^\star(\mathbf{x})$ depend on projections of $\mathbf{x}$ along distinct subspaces.

**Lemma 9** (Block–wise independence of hidden features). *For every layer index $\ell \in [L]$, the random variables $\left\{ h_{\ell,m}^\star(\mathbf{x})\right\}_{m=1}^{d^{\varepsilon_\ell}}$ are mutually independent and each has zero mean and unit variance.*

*Proof.* We prove this result by induction.

- Base case ($\ell = 1$). The first–layer features are $\mathbf{h}_1^\star(\mathbf{x}) = W^\star\mathbf{x}$ with orthonormal rows. Hence the components $\langle\mathbf{w}_{1,m}^\star, \mathbf{x}\rangle$ are i.i.d. $\mathcal{N}(0,1)$ and independent.

- Induction step. Assume the claim holds for layer $\ell - 1$. Fix $m \in [d^{\varepsilon_\ell}]$ and recall

$$h_{\ell,m}^\star(\mathbf{x}) = \frac{1}{\sqrt{d^{\varepsilon_{\ell-1}-\varepsilon_\ell}}}\, \mathbf{a}_{\ell,m}^{\star\top}\, P_{k,m,\ell}\Big(\mathbf{h}_{\ell-1,\mathcal{B}_m}^\star(\mathbf{x})\Big),$$

where $\mathcal{B}_m$ is the $m$-th disjoint block of indices of size $d^{\varepsilon_{\ell-1}-\varepsilon_\ell}$. By the induction hypothesis the entries of $\mathbf{h}^\star_{\ell-1,\mathcal{B}_m}(\mathbf{x})$ are independent of those in any other block $\mathcal{B}_{m'}$ $(m' \neq m)$. Because $P_{k,m,\ell}$ and the inner product with $\mathbf{a}^\star_{\ell,m}$ are deterministic maps, $h^\star_{\ell,m}(\mathbf{x})$ depends only on block $\mathcal{B}_m$ and is independent of $h^\star_{\ell,m'}(\mathbf{x})$ for $m' \neq m$. The variance–normalization follow from the definition of $P_{k,m,\ell}$ and the scaling $1/\sqrt{d^{\varepsilon_{\ell-1}-\varepsilon_\ell}}$.

This concludes the proof. $\qquad\qquad\square$

## D.2    Proof of Theorem 2

By the independence and asymptotic Gaussianity of the features $\mathbf{h}^\star_\ell(\mathbf{x})$ we expect the above result to extend to a general number of layers. However, proving such a result in its full-generality requires accounting for the non-asymptotic rates for the tails of $\mathbf{h}^\star_\ell(\mathbf{x})$ and the associated kernels.

Instead, we prove a weaker result corresponding to the hierarchical weak-recovery of a single non-linear feature at a general level of depth, given by Theorem 2.

The central tool underlying our proof is a propagation of hyper-contractivity through the layers:
**Proposition 22** (Propagation of Hyper-contractivity). *Let $f : \mathbb{R} \to \mathbb{R}$ be a polynomial of finite-degree $k$. Then, for any $\ell \in \mathbb{N}$:*

*(i)* $\mathbf{h}^\star_\ell(\mathbf{x}) = \mathcal{O}_\prec(1)$.

*(ii)* $\mathbb{E}\left[|\mathbf{h}^\star_\ell(\mathbf{x})|\right] = \tilde{\mathcal{O}}(\frac{1}{\sqrt{d^{\varepsilon_\ell}}})$

*(iii)* $\mathbb{E}\left[\left\|\mathbf{h}^\star_\ell(\mathbf{x})\mathbf{h}^\star_\ell(\mathbf{x})^\top - \mathbb{I}_{P_\ell}\right\|\right] = \tilde{\mathcal{O}}(\frac{1}{\sqrt{d}}).$

*Proof.* The proof proceeds by induction. For $\ell = 1$, the statements hold by Gaussian-hypercontractivity (Lemma 5) since $\mathrm{He}_k$ for distinct $w^\star_i$ are uncorrelated, zero-mean random variables and thus $h^\star_2(\mathbf{x})$ has all moments of bounded order.

Suppose the statements hold for some $\ell \in \mathbb{N}$. Applying Lemma 1, we obtain:

$$\mathbb{E}\left[|\mathrm{He}_k(\mathbf{h}^\star_\ell)|\right] = \mathcal{O}(\frac{d^\delta}{\sqrt{d^{\varepsilon_\ell}}}), \qquad\qquad (233)$$

for any $\delta > 0$. Subsequently, applying 8 leads to the following propagation of tails:

$$h^\star_{\ell+1,m}(\mathbf{x}) = \frac{1}{\sqrt{d^{\varepsilon_{\ell-1}-\varepsilon_\ell}}}\mathbf{a}^{\star\top}_{\ell,m}P_{k,m,\ell}\left(\mathbf{h}^\star_{\ell-1,\{1+(m-1)d^{\varepsilon_{\ell-1}-\varepsilon_\ell},\dots,md^{\varepsilon_{\ell-1}-\varepsilon_\ell}\}}(\mathbf{x})\right)$$

$$= \frac{1}{\sqrt{d^{\varepsilon_{\ell-1}-\varepsilon_\ell}}}\sum_{i=1}^{\sqrt{d^{\varepsilon_{\ell-1}-\varepsilon_\ell}}}\mathcal{O}_\prec(1) = \mathcal{O}_\prec(1),$$

where we used the bound $h^\star_\ell(\mathbf{x}) = \mathcal{O}_\prec(1)$ by the induction hypothesis. By Lemma 8 and Equation 233, for any $\delta > 0$, we obtain that:

$$\mathbb{E}\left[|h^\star_{\ell+1,m}(\mathbf{x})|\right] = \tilde{\mathcal{O}}(\sqrt{d^{\varepsilon_\ell-\varepsilon_{\ell+1}}}\frac{1}{\sqrt{d^{\bar{\varepsilon}}_\ell}}) = \tilde{\mathcal{O}}(\frac{1}{\sqrt{\varepsilon_{\ell+1}}}). \qquad (234)$$

$$\square$$

The above proposition establishes that the hidden features $h^\star_\ell(\mathbf{x})$ maintain errors in means $\mathcal{O}_\prec(\frac{1}{\sqrt{d^{\varepsilon_\ell}}})$ and preserve tails of the form $\mathcal{O}_\prec(1)$. Theorem 2 then follows by noting that the above error bounds suffice for Proposition 16 to hold for the feature-matrix $\sigma(Wh^\star_{L-1}(\mathbf{x}))$. Concretely, $h^\star_\ell(\mathbf{x}) = \mathcal{O}_\prec(1)$ ensures that Lemma 2 applies while the errors in means, covariances suffice for the expected covariance of spherical harmonics to converge to $\mathbf{I}$.

Analogous to Section C.20, we introduce the following partitioning of the indices:

$$\mathcal{S}_1 = \{j_1, k_1 : k_1 = 0, j_1 \leq 2k\} \cup \{j_1, j_2, k_1, k_2 : j_1 = 0, k_1 \leq k\}$$
$$\mathcal{S}_2 = \{j_1, k_1 \in \mathbb{N}^2\}\backslash(\mathcal{S}_1 \cup \mathcal{S}_1).$$

Above, we only have two partitions as opposed to the three partitions in Section C.20 since the features $h^\star_{L-1}(\mathbf{x})$ are no longer partitioned into disjoint spaces, unlike the partitioning of $\mathbf{x}$ into $\mathbf{x}^\star, \mathbf{x}^\perp$ in Section C.14.

We again write:

$$\sigma(W h^\star_{L-1}(\mathbf{x})) = \Psi_{\mathcal{S}_1}(h^\star_{L-1}(\mathbf{x}))\Phi(W)^\top_{\mathcal{S}_1} + \Psi_{\mathcal{S}_2}(h^\star_{L-1}(\mathbf{x}))\Phi(W)^\top_{\mathcal{S}_2}, \tag{235}$$

Unlike Proposition 16 that involved approximations in $W$, the above decomposition involves approximating $h^\star_{L-1}(\mathbf{x})$ through equivalent Gaussian-inputs $\mathbf{x}$. The proof follows that of Proposition 16, with Proposition 22 implying that:

$$\left\| \frac{1}{n} \sum_{\mu=1}^n \psi_{\mathcal{S}_1}(h^\star_{L-1}(\mathbf{x}_\mu))\psi_{\mathcal{S}_1}(h^\star_{L-1}(\mathbf{x}_\mu)))^\top - \mathbb{I}_M \right\| = \mathcal{O}_\prec(\frac{1}{\sqrt{d^\delta}}). \tag{236}$$

For the corresponding non-linear features along $W$, since the rows of $W$ are independently sampled along $U(\mathcal{S}_d(1))$, we directly have:

$$\left\| \frac{1}{p} \sum_{\mu=1}^n \phi_{\mathcal{S}_1}(h^\star_{L-1}(\mathbf{w}_i))\Psi_{\mathcal{S}_1}(h^\star_{L-1}(\mathbf{w}_i)))^\top - \mathbb{I}_M \right\| = \mathcal{O}_\prec(\frac{1}{\sqrt{d^\delta}}). \tag{237}$$

The remainder of the proof follows that of Propositions 17 and 18.

# E   Extension to MIGHTs

While our analysis is restricted to $r = 1$, we discuss here the primary challenges and directions towards the extension of our results to $r > 1$:

(i) **Spherical recovery**: Under the assumption, $a^\star_i = 1$, and $\mu_1(g^\star) = c(1, 1, \cdots)$ for some constant $c$, our analysis for the recovery of $W^\star_1, \cdots, W^\star_r$ by $W_1$ remains identical. Specifically, each neuron $\mathbf{w}^1_i$ recovers $\mathbf{u}^\star = \frac{P_{W^\star_1, \cdots, W^\star_r} \mathbf{w}^1_i}{\left\| P_{W^\star_1, \cdots, W^\star_r} \mathbf{w}^1_i \right\|}$.

(ii) **Specialization**: Even under the above symmetric setup, a single pre-conditioned gradient step only leads to recovery by $h_2(\mathbf{x})$ of the symmetric direction $\frac{1}{\sqrt{r}} \sum_{i=1}^r h^\star_i(\mathbf{x})$. Hence, extension of our analysis to $r > 1$ requires specialization through multiple pre-conditioned gradient steps. One promising approach to achieve such specialization is through the use of the staircase mechanism [2, 1] in the target $g^\star(\cdot)$.

# F   Details on the Numerical Investigation

In this section, we provide additional insights into the numerical illustrations presented in the main text. We refer to https://github.com/IdePHICS/ComputationalDepth for the code.

## F.1   Shallow methods

We illustrate in Fig. 2 the performance of two shallow methods: kernels (orange) and two-layer networks (green). At stake with three-layer architectures (red and blue), shallow methods are not able to perform non-linear feature learning, hence resulting in suboptimal performance. Below, we provide additional clarifications on these methods.

**Kernel methods –**   We consider a quadratic kernel $k(\mathbf{x}_1, \mathbf{x}_2) = (\mathbf{x}^\top_1, \mathbf{x}_2)^2 + (\mathbf{x}^\top_1, \mathbf{x}_2) + c = \varphi^\top_{\text{quad}}(\mathbf{x}_1)\varphi_{\text{quad}}(\mathbf{x}_2)$ that is an optimal choice among kernel mappings in the data regime explored ($n = o_d(d^{2+\delta})$), as follows by the asymptotics results in [60]. The feature map $\varphi_{\text{quad}}$ is not learned, therefore we refer to kernel methods as "fixed feature" methods. The lack of feature learning, and therefore adaptation to the relevant low-dimensional subspaces present in the SIGHT target $f^\star$, results in a large error value achieved by the best possible kernel methods (signaled with an orange solid line in Fig. 2) that serve as a lower bound for the simulations (shown as orange points). This bound coincides with the best quadratic approximation of the target as shown by [60]. The figure shows also neatly the presence of the double descent peak when the number of data equals the dimension of the feature space, sometimes called the interpolation peak: $n_{\text{peak}} = d(d-1)/2 + d + 1$; this is illustrated by a vertical orange dashed line in the left section of Fig. 2.

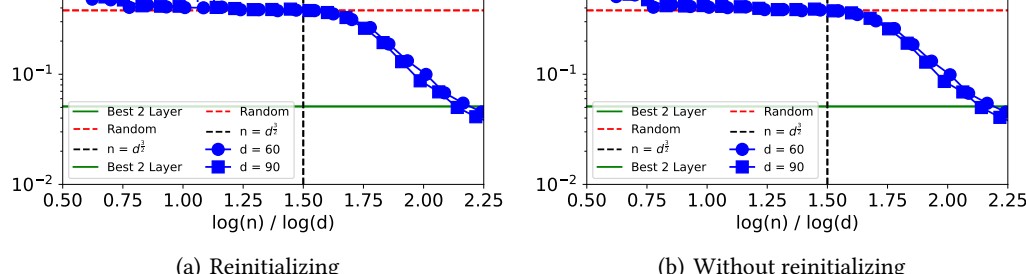

(a) Reinitializing
(b) Without reinitializing

Figure 5: **Reinitialization of subsequent layers:** The plots compare the generalization error achieved by two variants of the layerwise procedure in Theorem 1. The left panel illustrates a routine with reinitialization of the subsequent layers against a procedure where this assumption is relaxed in the right panel. There is no substantial difference between the two algorithms when looking at the generalization performance. The target is $f^\star(\mathbf{x}) = \tanh\left(\mathbf{a}^{\star\top} P_3\left(W^\star \mathbf{x}\right)/\sqrt{d^{\varepsilon_1=1/2}}\right)$ and the hyperparameters are listed in Sec. F.4.

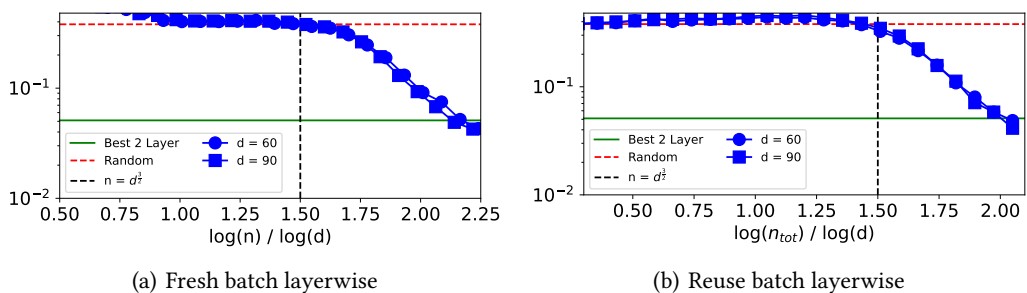

(a) Fresh batch layerwise
(b) Reuse batch layerwise

Figure 6: **Reuse of the same data batch over layers:** The plots compare the generalization error achieved by two variants of the layerwise procedure in Theorem 1. The left panel illustrates a routine without using the same batch of data for different layers of training, while on the right this assumption is relaxed by always holding constant the total number of samples seen for every layer. There is no substantial difference between the two algorithms when looking at the generalization performance. The target is $f^\star(\mathbf{x}) = \tanh\left(\mathbf{a}^{\star\top} P_3\left(W^\star \mathbf{x}\right)/\sqrt{d^{\varepsilon_1=1/2}}\right)$ and the hyperparameters are listed in Sec. F.4.

**Two-layer networks –** Two-layer networks are able, on the other hand, to capture linear features in the SIGHT target $f^\star$ (denoted $W^\star$ in eq. (1)). This is exemplified in Fig. 2 by the green points, with a net decrease in the test error with respect to kernel methods (orange ones). The generalization error shows a transition around the expected $\kappa = 1.5$, where Theorem 1 predicts that the linear features $W^\star$ are recovered (shown in the illustration by a vertical black line). However, we observe that two-layer networks in this setting cannot surpass the green solid line, corresponding to the best quartic approximation of the target. This is explained by the fact that, although partial dimensionality reduction has been achieved $d \to d^{\varepsilon_1} = \sqrt{d}$, two-layer networks are still performing random features in a $\sqrt{d}$−dimensional space. Therefore, with $n \simeq p = O(d^2) = O(\sqrt{d}^4)$ samples and neurons, we can fit the best quartic approximation of the target [60].

### F.2 Three layer networks

The results portrayed in Fig. 2 show a stark contrast between two and three-layer networks, with the latter surpassing the best possible performance for a shallow network (green solid line) thanks to the presence of non-linear feature learning.

We consider two training routines: a) the layerwise procedure, resembling Theorem 1 and algorithmically described in Alg. 1; b) training using backpropagation and vanilla regularized gradient descent for all the layers jointly.

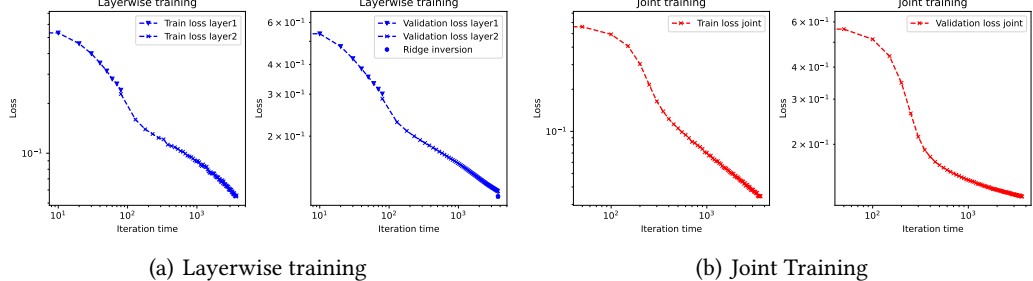

| (a) Layerwise training | (b) Joint Training |

Figure 8: **Training/Validation loss:** The plots illustrate the behavior of the training and validation losses as a function of the iteration time. It shows respectively on the left the layerwise training procedure inspired by Theorem 1 (Alg. 1), while on the right standard joint training using backpropagation. The target is $f^\star(\mathbf{x}) = \tanh\left(3\mathbf{a}^{\star\top} P_3\left(W^\star\mathbf{x}\right)/\sqrt{d^{\varepsilon_1=1/2}}\right)$ and the hyperparameters are listed in Sec. F.4

**Remark on Algorithm 1 –** Throughout this section we will consider a slight generalization of the routine in Alg. 1: we will update the second layer weights reusing a single batch of size $\mathcal{O}(d^{k\varepsilon})$ for up to $\mathcal{O}(d^{k\varepsilon})$ steps instead of using a single gradient step with preconditioning. We refer to Sec. C.16 for discussion on the difficulties of analyzing rigorously such routine.

Moreover, we do not follow all the theoretical prescriptions needed to prove rigorously the results and included in Alg. 1. The goal of Figures 5 and 6 is to exemplify the capability of lifting some of the theoretically needed assumptions. Respectively, in Fig. 5 we analyze the presence of reinitialization of subsequent layers, and in Fig. 6 we consider the presence of shared batches across layers. In both cases, we do not observe a stark difference between the two settings. Finally, while the proof scheme is limited to targets with $a_i^\star = 1$ (see eq. (1)), we consider in the numerical simulations $\mathbf{a}^\star$ drawn from a Rademacher distribution rather than being constant.

We plot the training and validation loss curves that guided our analysis in Fig. 8.

### F.3  Visualizing Feature Learning

We now show that this enhanced generalization performance is due to feature learning. Indeed, the key result in Thm 1 refers to the ability of three-layer networks to perform hierarchically dimensionality reduction through feature learning. To probe the quality of the learned representations, we shall introduce the "overlaps" (or order parameters).

**Definition 6.** *The order parameters for* $3-$*layer networks are the matrices* $M_W \in \mathbb{R}^{p_1 \times rd^{\varepsilon_1}}$ *and* $M_h \in \mathbb{R}^{p_2 \times r}$ *(with* $\mathbf{z} \sim \mathcal{N}(0, I_d)$*)*

$$M_W = \frac{W_1 W^\star}{\|W_1\|_F}, \ M_h = \frac{\mathbb{E}[\mathbf{h}(\mathbf{z})\mathbf{h}^\star(\mathbf{z})]}{\sqrt{\mathbb{E}[\mathbf{h}(\mathbf{z})^2]}}. \tag{238}$$

The behavior of these quantities as a function of the sample complexity $\kappa$ is portrayed in Fig. 7. Since we do not

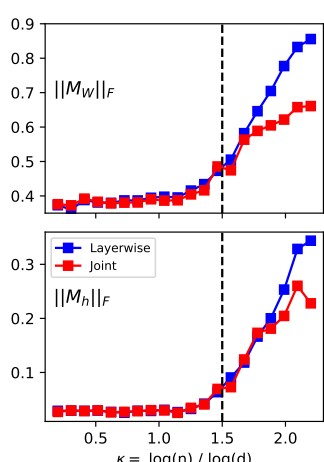

Figure 7: **Visualizing Feature Learning:** The Frobenius norm of the overlaps $M_h$, $M_W$ (Def. 6), respectively on the top and bottom panel, as a function of the sample complexity $\kappa = \frac{\log n}{\log d}$ for three-layer networks trained with the protocol described in Theorem 1 (blue circles) and standard backpropagation (red squares). Following Theorem 1, the behavior sharply changes around $\kappa = 1.5$ (vertical dashed line) where feature learning in both layers arises (same setting as in Fig. 2).

follow the strong prescription of Thm. 1, and are working with a low dimensional example, we do not expect a sharp $0/1$ transition as in the idealized scenario, but instead, the components along

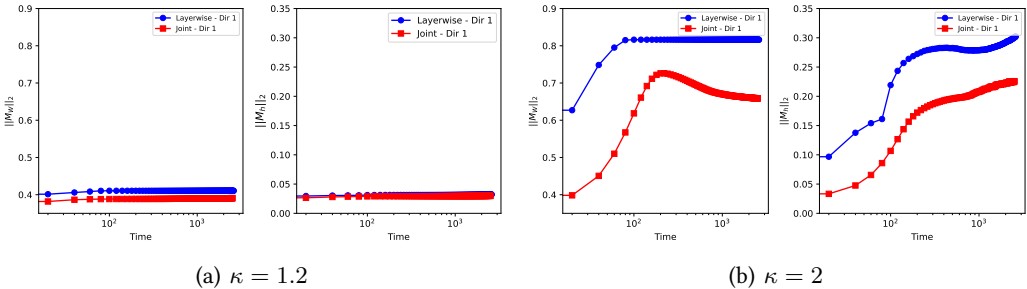

(a) $\kappa = 1.2$                                    (b) $\kappa = 2$

Figure 9: **Visualizing Feature Learning:** The plot shows the evolution of the Frobenius norm of the overlaps (Definition 6) as a function of the training time $t$ for two different values of $\kappa = \frac{\log n}{\log d}$, respectively $\kappa = 1.2$ on the left and $\kappa = 2$ on the right. Different training methods are illustrated with different colors: in blue the layerwise training (Alg. 1), in red standard joint training using backpropagation. The target is $f^\star(\mathbf{x}) = \tanh\left(\mathbf{a}^{\star\top} P_3\left(W^\star \mathbf{x}\right)/\sqrt{d^{\varepsilon_1 = 1/2}}\right)$ and the hyperparameters are listed in Sec. F.4

$W^\star$ to occupy a $\Theta(1)$ fraction (but not full) of the norm of $W_1$. This is well obeyed (Fig. 7) and the predicted crossover at $\kappa = 1.5$ is clearly observed in both layerwise and joint training.

We exemplify in Fig. 9 the "dual" plot of Fig. 7 by showing the evolution in time of the sufficient statistics $M_W, M_h$ for two different values of $\kappa = \frac{\log n}{\log d}$. The plot shows that when $\kappa < 1.5$ (the critical threshold) feature learning is impossible, as it is reflected by the overlaps attaining the random guess value. On the other hand for $\kappa > 1.5$ the overlaps grow far from the random initialization performance.

Additionally, we illustrate the evolution in time of the overlaps under the learning of MIGHT functions (eq. (4)) in Fig. 10. The figure exemplifies the necessity of Assumption 1 that refers to the generalization of the information exponents [19, 28] of the multi-index target literature to the present hierarchical setting.

### F.4    Hyperparameters

In every figure showing sufficient statistics or generalization errors, we average over 20 different seeds and plot the median. The regularization strengths for the different layers are optimized with standard hyperparameter sweeping for every value of $\kappa$ plotted, while the other hyperparameters are considered fixed. More precisely, we fix:

  (i) First hidden layer size: $p_1 = \text{int}(n_{max}^{1-\delta})$, with $n_{max}$ the maximal $n$ probed in the respective plot and $\delta = 0.1$

 (ii) Second hidden layer size: $p_2 = 600$.

(iii) Hidden layer size for two-layer network: $p = \text{int}(p_1/25)$

(iv) Learning rates: while the orders of magnitude for the different learning rates as a function of $d$ are provided in Alg. 1 for layerwise training we use fixed prefactor $\text{lr}_1 = 1, \text{lr}_2 = 2$. Concerning joint training we use instead for all the three layers all the prefactors equal to 0.2.

 (v) Minibatch size: $n_b = \text{int}(\frac{7n}{10})$, with $n = d^\kappa$.

(vi) Iteration time: we follow the prescriptions of Theorem 1 iterating for $T_1 = O(\text{polylog}(d))$ steps and $T_2 = O(d^{1.5})$ steps. In the numerical implementation we consider for layerwise training $T_1 = \text{int}(15 \log d), T_2 = \text{int}(5d^{1.5})$. On the other hand, for standard training using backpropagation, we iterate jointly all the layers for $T_2$ steps.

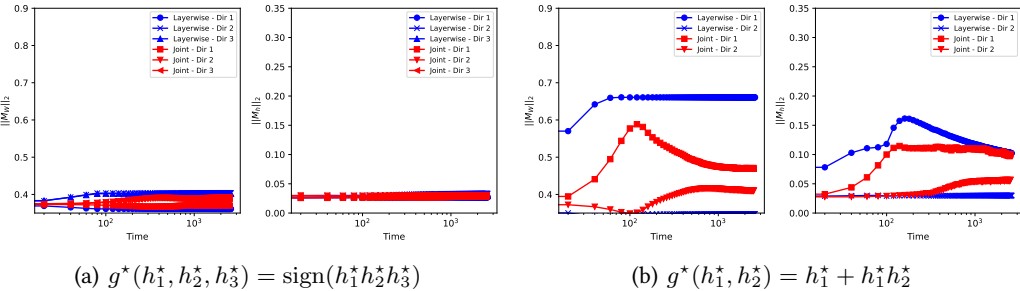

(a) $g^\star(h_1^\star, h_2^\star, h_3^\star) = \operatorname{sign}(h_1^\star h_2^\star h_3^\star)$  (b) $g^\star(h_1^\star, h_2^\star) = h_1^\star + h_1^\star h_2^\star$

Figure 10: **Easy and Hard Features:** The plot shows the evolution of the Frobenius norm of the overlaps (Definition 6) as a function of the training time $t$ for two different values MIGHT functions $f^\star(\mathbf{x}) = g^\star(\{h_l^\star\}_{l=1}^r)$ (See eq. (4)), with the non-linear features built as in Fig. 7, i.e., $h^\star(\mathbf{x}) \propto \mathbf{a}^{\star\top} P_3(W^\star \mathbf{x})$ and $P_3 = \mathrm{He}_2 + \mathrm{He}_3$. The hyperparameters are listed in Sec. F.4. Different training methods are illustrated with different colors: in blue the layerwise training (Alg. 1), in red standard backpropagation. The overlap component along different directions ($h_l^\star, l = 1 \cdots r$) are signaled with different markers.

