# OpenReview forum: "The Computational Advantage of Depth in Learning High-Dimensional Hierarchical Targets"
_NeurIPS.cc/2025/Conference — NeurIPS 2025 spotlight_

### Official Review · Reviewer_mnX3 · 2025-06-28

**Clarity:** 3
**Significance:** 3
**Originality:** 2
**Rating:** 4
**Confidence:** 3

**Summary:**

This paper quantifies the advantage of depth in models learned with gradient descent by demonstrating that deeper models learn hierarchical data models with better sample complexity compared to shallow ones. The authors propose a class of deep multi-index targets (SIGHT/MIGHT) that mimic the hierarchical structure of real data. Based on this, the authors prove (for the shallow single-index model) that three-layer networks can learn (via layerwise training) such targets with lower sample complexity compared to two-layer networks or kernel methods. Numerical simulations validate findings beyond the setting of the theory.

**Questions:**

+ My understanding is that three layers aligns closely with the target model (where the middle layer matches up with the polynomial features h). As such, is there any benefit to introducing additional layers?
+ For simple real datasets, can we observe a separation in sample complexity between two- and three-layer networks?

**Ethical Concerns:**

["NO or VERY MINOR ethics concerns only"]

**Final Justification:**

I believe this paper is a good contribution to tackling the feature learning problem in the multi-layer setting, and the authors have somewhat addressed my concern regarding novelty of the problem setup. I still feel that the experiments are a weak point, particularly for validating findings beyond the synthetic setting - nonetheless, I lean towards acceptance.

**Limitations:**

Yes.

**Paper Formatting Concerns:**

There might be some divergence in font from the usual template, authors should check this is ok.

**Quality:**

2

**Strengths And Weaknesses:**

The paper is technically sound and rigorous, providing theoretical results to back their claims. Assumptions are clear, although somewhat overly specific at times. However, numerical simulations are quite limited, and although they explore relaxing certain assumptions, are still largely rooted in the synthetic setting. The paper is mostly well written - in particular, the sketch of analysis with exploration in a simplified setting is very effective for elucidating the claims. Figure 1 is very helpful for visualizing the phase transitions. Some additional formatting could be beneficial for ease of reading. The problem studied is challenging and an active area of research, and this paper yields new developments, particularly in relaxing the assumption of fixed first layer. However, the target introduced for imposing hierarchical structure does not appear to be significantly novel, weakening this aspect of the paper.

---

> ### Author Rebuttal · Authors · 2025-07-30
>
> We sincerely thank the reviewer for the positive comments about our work and constructive feedback. Below, we answer your remarks individually.
>
> > My understanding is that three layers aligns closely with the target model (where the middle layer matches up with the polynomial features h). As such, is there any benefit to introducing additional layers?
>
>  The reviewer is correct in their interpretation. For our class of targets, there is no benefit, as far as feature learning and the scaling of sample complexity is concerned, in introducing an additional layer in the scenario described where the target model is also three-layer like.
>
>
>
> > For simple real datasets, can we observe a separation in sample complexity between two- and three-layer networks?
>
>
> We did not explore sample-complexity separation on real datasets in the present manuscript. Although the focus here is theoretical, exploring the consequences of our results on real data, via identification of effective dimension and intermediate hidden representations, is a promising avenue for future work, and we will mention this in L73–74.
>
>
> > Assumptions are clear, although somewhat overly specific at times.
>
>
> We acknowledge concerns about the specificity of our assumptions. In the revised version, we will integrate Appendix C.7 into the main text, where we will explicitly construct an activation that satisfies all assumptions to provide clarity.
>
> > However, numerical simulations are quite limited, and although they explore relaxing certain assumptions, are still largely rooted in the synthetic setting.
>
>  Extending the verification of our theory to realistic data distributions is challenging, as we currently lack methods to identify the "effective dimensions" of hidden representations in such settings. Nevertheless, our analysis provides concrete directions for such explorations, for example, by interpreting the eigenfunctions of the conjugate kernel across depth and considering appropriate notions of "effective rank" for the same. We will highlight these experiments as future work in the revised version.
>
>
> > The target introduced for imposing hierarchical structure does not appear to be significantly novel, weakening this aspect of the paper.
>
> While several hierarchical data models have inspired our targets (see L163–172), we respectfully disagree that our MIGHT targets lack novelty. They offer an exactly solvable playground enabling rigorous description of feature learning capabilities in deep networks trained with Gradient Descent through progressive dimensional reduction.
>
> Thank you for pointing out the typos that will be fixed in the revised version and the valuable observations. We remain available for any additional clarification.

---

> > ### Comment · Reviewer_mnX3 · 2025-08-03
> >
> > Thank you for addressing the points raise and suggesting additions based on these, I think discussing extensions to real datasets would be especially invaluable. Could the authors expand a little bit more on the novelty of MIGHT - in particular, how does it differ from the deep hierarchical targets that were previously explored?

---

> > > ### Author Response · Authors · 2025-08-05
> > >
> > > We thank the reviewer for the response and will certainly include a discussion of extensions to real datasets in the revised version.
> > >
> > > The key novelty of MIGHT targets lies in enabling an analysis of neural networks trained with gradient descent. This is made possible by two features: progressive reduction in dimensionality and the Gaussianity of inputs and features $h^*(x)$, which together allow gradient steps on hidden layers to be expanded in suitable bases.
> > >
> > > By contrast, prior hierarchical data models [23, 64, 66] have primarily enabled only analyses of approximation efficiency or sample complexity under non-gradient-based methods.
> > >
> > > MIGHT targets unify hierarchical structure with the analysability of feature learning seen in single/multi-index models and their extensions to non-linear features introduced in [36, 65, 82]. Unlike [36, 65, 82], however, MIGHT targets involve learning across all layers and naturally extend to arbitrary depth (see Section 1.3).
> > >
> > > We will include the above discussion of novelty in the revised version.

---

### Official Review · Reviewer_r5Rj · 2025-07-01

**Clarity:** 3
**Significance:** 3
**Originality:** 4
**Rating:** 5
**Confidence:** 4

**Summary:**

This work studies the problem of learning hierarchical Gaussian single/multi-index models with 3-layer neural networks. Specifically, for a 3-layer single-index model, the authors prove that a layerwise variation of gradient descent can achieve dimension reduction and obtain a smaller sample complexity compared to fixed kernel regression. The authors also provide intuitions on how 3-layer networks would achieve a separation with 2-layer ones through feature learning in the middle layer which enables additional dimension reduction.

**Questions:**

1. To my understanding, Theorem 2 is not really considering training in arbitrary middle layers. It only studies what happens when training the $L-1$ layer where $L$ is the network depth. Would it be possible that under a specific initialization of later layers, the same result can hold for middle layers without much additional effort?

2. A missing related work is [1] where the authors study learning multi-index models with neural nets under structured input. Specifically, a direction for future work can be to extend the definition of effective dimension in [1] to hierarchical models to capture the effect of anisotropy on sample complexity.

3. Some typos:
    * Figure 5 caption: $d^{\varepsilon_2 - \varepsilon_1}$ -> $d^{\varepsilon_1 - \varepsilon_2}$ and $d^{\varepsilon_3 - \varepsilon_2}$ -> $d^{\varepsilon_2 - \varepsilon_3}$.

    * The notation sometimes uses $\varepsilon_1$ and sometimes uses $\epsilon_1$.

    * The statement starting from point (ii) of Theorem 3 has missing words.

    * Theorem 3 uses both $w_3$ and $W_3$.

References:

[1] A. Mousavi-Hosseini, D. Wu, M. A. Erdogdu. "Learning Multi-Index Models with Neural Networks via Mean-Field Langevin Dynamics" ICLR 2025.

**Ethical Concerns:**

["NO or VERY MINOR ethics concerns only"]

**Final Justification:**

I didn't have any major concerns, and my minor comments were addressed by the authors. Other reviews are positive as well, so I'm happy to recommend acceptance.

**Limitations:**

Yes

**Paper Formatting Concerns:**

None.

**Quality:**

3

**Strengths And Weaknesses:**

I think the results are interesting as they provide examples of analyzing hierarchical models when the size of the middle layers diverge with dimension, which to my understanding is the key technical contribution of this work compared to prior works on learning features with 3-layer networks. The authors also provide some nice intuition on how to generalize their results to deeper networks and characterize what type of hierarchical models with depth larger than 3 can eventually be covered using these techniques, which can be interesting for future work.

My main concern is the lack of a concrete negative result for 2-layer networks. The authors provide a heuristic argument in the main text for why they would achieve a higher sample complexity for learning higher degree components in the target, but given the importance of such a negative result to theoretically capture the computational limitations of 2-layer nets, I think it deserves a theorem of its own. Specifically, I think the argument presented in Appendix B.1 is nice, I'm not sure why it isn't highlighted as a formal theorem in the main text.

I think the numerical simulations, in their current presentation, may not be entirely conclusive about the separation between 2-layer and 3-layer nets. The range of $\kappa$ in Figure 2 mostly includes the [0,1.5] interval where no algorithm achieves small test loss. It would be nice if the figure focused on larger $\kappa$ where a more clear separation should be observed between all algorithms, at least up to $\kappa=3$ similar to Figure 1. However, I agree that performing experiments for larger $\kappa$ can be challenging due to the amount of needed compute.

---

> ### Author Rebuttal · Authors · 2025-07-30
>
> We sincerely thank the reviewer for the positive comments about our work and constructive feedback. Below, we answer your remarks individually.
>
> > To my understanding, Theorem 2 is not really considering training in arbitrary middle layers. It only studies what happens when training the $L-1$ layer where $L$ is the network depth. Would it be possible that under a specific initialization of later layers, the same result can hold for middle layers without much additional effort?
>
>
>  Theorem 2 applies to a simplified setting where the $(L-2)$th layer has already perfectly recovered the corresponding features. Hence, only the $L-1$ layer is required to additionally learn the last level of hidden features. Ideally, we hope to apply the argument inductively to prove feature learning for all layers. The primary bottleneck toward such a generalization is that our analysis does not apply to the recovery of a diverging number of non-linear features.
>
>
> > A missing related work is [1] where the authors study learning multi-index models with neural nets under structured input. Specifically, a direction for future work can be to extend the definition of effective dimension in [1] to hierarchical models to capture the effect of anisotropy on sample complexity.
>
>
> We thank the reviewer for the suggestion regarding [1]; extending the definition of effective dimension in [1] to hierarchical models is an exciting direction for future research and will be discussed in the revised version.
>
>
> > My main concern is the lack of a concrete negative result for 2-layer networks. The authors provide a heuristic argument in the main text for why they would achieve a higher sample complexity for learning higher degree components in the target, but given the importance of such a negative result to theoretically capture the computational limitations of 2-layer nets, I think it deserves a theorem of its own. Specifically, I think the argument presented in Appendix B.1 is nice, I’m not sure why it isn’t highlighted as a formal theorem in the main text.
>
>  We agree on the importance of obtaining and stating precise negative results. The discussion in Appendix B.1 is limited to a particular choice of the MIGHT target, different from the SIGHT targets considered in Theorem 1. In the revised version, we will integrate this discussion into the main text as a formal corollary of existing results. Obtaining such negative results for the general choices of SIGHT and MIGHT targets is however, beyond the scope of our work.
>
>
> > I think the numerical simulations, in their current presentation, may not be entirely conclusive about the separation between 2-layer and 3-layer nets. The range of $\kappa$ in Figure 2 mostly includes the [0,1.5] interval where no algorithm achieves small test loss. It would be nice if the figure focused on larger $\kappa$ where a more clear separation should be observed between all algorithms, at least up to $\kappa=3$ similar to Figure 1. However, I agree that performing experiments for larger $\kappa$ can be challenging due to the amount of needed compute.
>
> To address concerns about the simulation section, we will include references to Appendix F.3, where the evolution of the order parameters (Definition 236) exemplifies sharper transitions, and add experiments for additional input dimensions and values of $\kappa$. We appreciate the reviewer’s understanding regarding computational constraints and will clarify learning-rate choices in the numerical appendix.
>
>
> Thank you for pointing out the typos that will be fixed in the revised version and the valuable observations. We remain available for any additional clarification.

---

> > ### Comment · Reviewer_r5Rj · 2025-08-01
> > **Thanks**
> >
> > Thank you for your responses. I have no other questions and I'm happy to recommend acceptance.

---

### Official Review · Reviewer_xWdV · 2025-07-01

**Clarity:** 3
**Significance:** 4
**Originality:** 3
**Rating:** 5
**Confidence:** 3

**Summary:**

The study introduces single and multi-index Gaussian hierarchical targets as learning models designed to illustrate the benefits of deep versus shallow architectures. These models are based on the assumption that features higher in the hierarchy have lower dimensionality than features at lower levels. The authors provide a theoretical analysis of the performance of different architectures on these tasks. Focusing on the three-layer case and under a specific training regime, they derive the sample complexity thresholds that separate the performance of three-layer neural networks from two-layer networks and kernel methods.

**Questions:**

### Questions:
- Can the authors comment on the technical benefits of using online batch learning for the first layer and a giant step for the second? I would have expected online batch learning in both layers to perform better, but I assume it would also be significantly harder to analyze?
- Why do the networks in Figure 2 perform worse than random chance during the initial stages of learning?
- As a curiosity: I would expect that once the condition $1 > \epsilon_1 > \dots > \epsilon_L$ is violated, depth would no longer be beneficial. Can the authors comment on this scenario?

### Minor Remarks:
- Line 217: One word seems to be missing, likely "additional" after "to fit an".
- Line 219: "completely" is spelled incorrectly.
- Lines 260–261 seem contradictory: a problem is stated to be open, yet the next line references [7] as providing a solution. Does [7] consider a different setting than the open problem just mentioned?

**Ethical Concerns:**

["NO or VERY MINOR ethics concerns only"]

**Final Justification:**

The rebuttal addressed all my question satisfactorily, I confirm my rating on the motivations of the initial review.

**Limitations:**

yes

**Paper Formatting Concerns:**

I did not notice any major formatting concern

**Quality:**

4

**Strengths And Weaknesses:**

### Strengths:
- The role of depth in learning with neural networks is considered a crucial open question in the theory of machine learning. Hence, it is significant to provide a theoretical model that showcases aspects of learning for which depth is beneficial.
- The hierarchical structure of the data model is principled, based on reasonable assumptions about real-world distributions. The narrowing of the dimensions $1 > \epsilon_1 > \dots > \epsilon_L$, a key assumption of the work, appears justified from my point of view.
- The intuitive illustration provided with example (11) simplifies and highlights the core dynamics, avoiding the heavy notation present in other sections.
- The sketch of the proof is very informative and emphasizes the key points of the analysis. In particular, point (ii) seems highly non-trivial.

### Weaknesses:
- All the analysis relies on the assumption of Gaussian inputs. While this is a clear gap with respect to real-world applications, I believe it should not carry much weight as a weakness, since most works in the field share this limitation.
- The theoretical analysis assumes a simplified layer-wise training protocol which, as the authors themselves state, could be suboptimal. However, simulations support that for this class of targets, standard SGD also achieves the same thresholds.
- The simulation section is not entirely satisfactory. Figure 2 shows very limited separation between two- and three-layer neural networks. I suggest extending the plot to larger values of $\kappa$, to verify whether the predicted plateaus are attained. It would also be advisable to run simulations for different values of $d$, to check whether the expected scalings hold. Additionally, the descent phase for both two- and three-layer networks seems to require many samples to approach their plateaus — why is that? Could this be due to suboptimal learning rates during the descent phase?

---

> ### Author Rebuttal · Authors · 2025-07-30
>
> We sincerely thank the reviewer for the positive comments about our work and constructive feedback. Below, we answer your remarks individually.
>
> > Can the authors comment on the technical benefits of using online batch learning for the first layer and a giant step for the second?
>
>  We will expand the paragraph dealing with the use of online batch learning for the first layer and a giant step for the second (L266–270) in the revised version. The use of online learning for the second layer is both harder to analyze as well as suboptimal in terms of sample complexity due to the ill-conditioning of the landscape. Due to the polynomial decay in spectrum of the conjugate Kernel, without pre-conditioning, both online or batch gradient descent would require polynomial in $d$ steps, which would in turn need a “drift plus martingale” type of analysis for the second layer features, leading to various technical challenges. However, our experiments support that our theoretical results hold for non-conditioned Gradient Descent with multiple iterations on the same batch (see L269–270 and L418–419); we will comment on this further in the revised version.
>
> > Why do the networks in Figure 2 perform worse than random chance during the initial stages of learning?
>
>  In the regimes of low sample-complexity, the gradient steps effectively add noise to the parameters. Since our simulations do not involve projections, this noise results in a scaling of the norms of the parameters. We believe this initial worsening of performance to simply be an artifact of this scaling.
>
>
> > As a curiosity: I would expect that once the condition $1 > \epsilon_1 > \dots > \epsilon_L$ is violated, depth would no longer be beneficial. Can the authors comment on this scenario?
>
> We thank the reviewer for this insightful question. We believe that an advantage of depth would persist even in the absence of a strict decrease in the chain of dimension exponents, since a high-degree polynomial expressible as a composition of lower-degree polynomials can be learned with fewer samples upon the recovery of the intermediate low-degree polynomials. For instance, consider the following heuristic example: a degree-4 target would require $O(d^4)$ samples to be directly learned in the original input-space. However, if the degree-$4$ target is expressible as a degree-2 function of $O(d)$ quadratic features, then  upon learning these $O(d)$ hidden quadratic features, the sample complexity of the target would be reduced to $O(d^2)$. Thus, the "dimension-reduction" can proceed at the level of the space of high-degree polynomial features even without an explicit reduction in the dimension of the features. Turning this heuristic into a rigorous analysis would, however, require a control over the "approximate independence" of such non-linear features, which we believe to be technically challenging. The strict decrease in dimensionality and the "tree-like" construction of targets in our setup avoids such issues by ensuring the exact independence of non-linear features. We will include the above clarifications in the revised version.
>
> > Lines 260–261 seem contradictory: a problem is stated to be open, yet the next line references [7] as providing a solution. Does [7] consider a different setting than the open problem just mentioned?
>
> The reviewer is correct and the present phrasing is confusing; it will be corrected in the revised version. What we meant is that [7] considers the problem of joint training and introduces the concept of backward feature correction, i.e., how errors in lower-level features can be corrected through training of higher-level features. Although [7] is an inspiring reference, the key element in our analysis is to rigorously control the overlap between the target and network preactivations and to precisely describe the separation of deep networks versus shallow ones thanks to non-linear feature learning and progressive dimensionality reduction. This task is currently hard to tackle in a joint training setting and it constitutes an interesting direction of research.
>
>
> > [Idealized setting] a) All the analysis relies on the assumption of Gaussian inputs. While this is a clear gap with respect to real-world applications, I believe it should not carry much weight as a weakness ... b) The theoretical analysis assumes a simplified layer-wise training protocol which ... could be suboptimal. However, simulations support that for this class of targets, standard SGD also achieves the same thresholds.
>
>
>  We strongly appreciate the constructive comments on the Gaussian-input assumption and the simplified training protocol. We agree there is a gap between theory and real-world applications, and our work provides initial steps towards understanding the mechanisms that one could expect to hold in other realistic settings.
>
>
> > The simulation section is not entirely satisfactory. Figure 2 shows very limited separation between two- and three-layer neural networks. I suggest extending the plot to larger values of $\kappa$, to verify whether the predicted plateaus are attained. It would also be advisable to run simulations for different values of $d$, to check whether the expected scalings hold. Additionally, the descent phase for both two- and three-layer networks seems to require many samples to approach their plateaus—could this be due to suboptimal learning rates during the descent phase?
>
>  To enhance the presentation of the numerical illustrations, we will include in the main text references to Appendix F.3, where we consider the evolution of the order parameters appearing in the main theorems (Definition 236)—i.e., the overlap between target and network preactivations—that exemplify a sharper transition around the predicted threshold. Moreover, we will include additional input dimensions and values of $\kappa$ as suggested. Details on learning-rate scaling and hyperparameter sweeps will be added to the numerical appendix.
>
> Thank you for pointing out the typos that will be fixed in the revised version and the valuable observations. We remain available for any additional clarification.

---

> ### Comment · Reviewer_xWdV · 2025-08-02
>
> Thanks for the detailed and satisfying answers, especially for the intuition in case the condition $1>\epsilon_1>\dots>\epsilon_L$ is violated: that polynomial example is really interesting and conveys another important aspect of depth. I suggest including it at least in the appendix. I confirm my rating.

---

> > ### Author Response · Authors · 2025-08-06
> >
> > We will make sure to include the example in the revised version, along with the other items based on your comments. Thank you again for your feedback.

---

### Official Review · Reviewer_x83L · 2025-07-03

**Clarity:** 3
**Significance:** 3
**Originality:** 3
**Rating:** 5
**Confidence:** 2

**Summary:**

The paper introduces two classes of hierarchical single-index and multi-index targets (called “SIGHT” and “MIGHT”) and employs them to establish separations between the capabilities of random feature models, shallow neural nets, and deep neural nets. They show that training deep models layer-by-layer enables progressive dimensionality reduction and better sample complexity rates than training on the same task with a shallower model. They prove sample complexity bounds on the ability of 3-layer neural networks to learn SIGHT instances. They empirically demonstrate these separations for deeper models and more complex tasks.

**Questions:**

Is there any overlap between these results and those of [1]? This also involved layer-wise training of hierarchical tasks.

Could the authors offer more intuition about why pre-conditioning is useful for training the second layers, and whether they suspect it to be necessary?

[1] https://arxiv.org/abs/2108.10573

**Ethical Concerns:**

["NO or VERY MINOR ethics concerns only"]

**Final Justification:**

My score remains as is after discussion.

**Limitations:**

yes

**Quality:**

3

**Strengths And Weaknesses:**

# Strengths

As far as I am aware, the results are novel and mathematically sound.

While the work is primarily theoretical, the ability of deep networks to iteratively reduce the intrinsic dimensionality of a task is practically relevant and believable. The involved mathematical assumptions are common amongst peer papers, and the decision to focus empirical study beyond those assumptions (e.g. for simultaneous, rather than layer-wise training) is commendable.

The paper is well-written, and Figure 1 is particularly helpful for visualizing the results of the paper.

# Weakenesses
When introducing assumptions in Section 3, it would be helpful to highlight examples that satisfy those assumptions (e.g. targets satisfying Assumption 3.)

Minor points:
- L277: I would recommend introducing the information exponent rather than referring readers to other papers, especially given its relevance to Assumption 3.
- L314: Instead of “exists a sequence of random matrices,” should this refer to a distribution of random matrices?
- L319: “preacitvation"

---

> ### Author Rebuttal · Authors · 2025-07-30
>
> We sincerely thank the reviewer for the positive comments about our work and constructive feedback. Below, we answer your remarks individually.
>
> > Is there any overlap between these results and those of [1]? This also involved layer-wise training of hierarchical tasks.
>
>  There has been an extensive line of research in the theoretical machine learning community focusing on the learnability of multi-index models by shallow two-layer networks with layer-wise gradient descent (see L27 of the submitted version). The reference [1] that the reviewer points out defines rigorously the concept of “staircase”, or sequential learning of relevant directions (in our notation, rows of $W^\star_1$). While staircase functions exhibit sequential learning, they do not require adaptation across depth. In this manuscript, we offer perspectives on the learnability of deep hierarchical targets (as opposed to multi-index models) with deeper networks (in contrast with two-layer ones).
>
>
> > Could the authors offer more intuition about why pre-conditioning is useful for training the second layers, and whether they suspect it to be necessary?
>
> The choice of pre-conditioning is needed for the rigorous scheme due to the ill-conditioning of the landscape and to the polynomial decay in the Random Feature kernel’s spectrum. Without such pre-conditioning, Gradient Descent will require polynomial in $d$ steps on the same batch of data to learn our class of target functions, which remains outside the scope of present proof techniques in the literature. However, our numerical experiments support that the theoretical claims hold for non-conditioned Gradient Descent with multiple iterations on the same batch (see L269–270). Therefore, we do not believe this element to be necessary and we will state it more clearly in the revised version.
>
> > The role of the assumptions: a) I would recommend introducing the information exponent rather than referring readers to other papers, especially given its relevance to Assumption 3. b) When introducing assumptions in Section 3, it would be helpful to highlight examples that satisfy those assumptions (e.g. targets satisfying Assumption 3).
>
> We thank the referee for the pertinent suggestions regarding Assumption 3. The existence of an activation satisfying the assumptions is present in Appendix C.7 but is not referenced correctly in the main text. We will introduce the information exponent explicitly and highlight concrete examples that satisfy each assumption.
>
> > L314: Instead of ‘exists a sequence of random matrices,’ should this refer to a distribution of random matrices?
>
>  The exact distribution of the random matrices is specified under the formal statement of the theorem in Appendix C.1.
>
> Thank you for pointing out the typos that will be fixed in the revised version and the valuable observations. We remain available for any additional clarification.

---

> > ### Comment · Reviewer_x83L · 2025-08-05
> >
> > Thank you for the detailed response. I will maintain my score.

---

### Official Review · Reviewer_G7h1 · 2025-07-07

**Clarity:** 3
**Significance:** 4
**Originality:** 3
**Rating:** 5
**Confidence:** 4

**Summary:**

This manuscript studies the problem of learning deep teacher networks in which each layer reduces the effective dimensionality through localized polynomial projections. Building on prior work on learning three-layer neural networks \[36, 65, 82], the authors extend the setting to incorporate low-dimensional structure in the input layer and to learn the first-layer weights, in contrast to \[36, 65, 82], which keep the input weights fixed.

By employing layer-wise training, the paper demonstrates that the network can successfully recover the low-dimensional representations at each layer and learn the full teacher model. The analysis relies on a universality argument for each layer, making the learning process analogous to multi-index learning with a two-stage procedure (feature learning combined with random feature approximation), a topic extensively studied in recent literature. Specifically, the results show a clear separation between shallow two-layer networks and deeper architectures when the teacher network admits a hierarchical representation. The authors also present simulations that support the theoretical insights developed in the paper.

**Questions:**

**Questions for Additional Discussion:**

* The results extend prior work on learning compositional functions [36,65,82] to deeper models and incorporate lower-dimensionality in the input layer. Although the argument for universality is explained in Section 2, the theoretical challenges involved in establishing these results are not detailed in the text. Discussing the main ideas and difficulties that led to this extension would help readers better follow the contributions.

* As far as I can tell, the paper assumes that the depth of the teacher and student networks is the same. What happens if there is a mismatch? In particular, is there any benefit to using a three-layer network to learn a two-layer teacher model, similar to what has been suggested in [65]?

* There is one part I could not follow regarding $a_i^\*$. It is first introduced as being sampled from a distribution but later fixed to a vector of all ones. I don’t think it is possible to rotate $a^\*$, given that you assume the nonlinearity in the second layer is the same polynomial $P_k$ applied elementwise—a rotation would seem to break this assumption. Is there a mistake here, or am I misunderstanding something?

**Typos:**

* In line 291, $\mathbb{E}[P_k(z) \, H_{e_2(z)}]$ appears redundant.
* In Theorem 1, the dimensions $p_2$ and $p_3$ should be equal, so the notation should be unified.
* In lines 150–153, references \[82] and \[65] should be reversed.

**Ethical Concerns:**

["NO or VERY MINOR ethics concerns only"]

**Final Justification:**

Good paper, see the *Strengths* part for the justification.

**Limitations:**

Yes.

**Quality:**

4

**Strengths And Weaknesses:**

**Strengths:**
The paper successfully combines existing two-stage algorithms for multi-index learning with results on learning compositional functions using consecutive random feature approximations. Overall, the results demonstrate that deep teacher networks can be learned under assumptions that are reasonable within the theoretical community. The paper is well written, the arguments are clearly illustrated with figures, and the results are discussed in relation to supporting simulations.

**Weaknesses:**
I do not think the work has any major weaknesses. However, the presentation could be improved in certain parts, and additional discussion of the theoretical challenges involved in applying the universality argument to gradient-based learning would help readers better understand the technical aspects of the work. I have outlined my suggestions as questions in the following section.

---

> ### Author Rebuttal · Authors · 2025-07-30
>
> We sincerely thank the reviewer for the positive comments about our work and constructive feedback. Below, we answer your remarks individually.
>
> >The theoretical challenges involved in establishing these results are not detailed in the text. Discussing the main ideas and difficulties that led to this extension would help readers better follow the contributions.
>
> We will expand upon the challenges in our analysis under the proof sketch in Section 3. The revised paragraphs will provide additional details along the lines of point (ii) in the proof sketch. More precisely, the analysis of the dynamics of the first-layer weights $W_1$ in our setting requires incorporating the diverging dimensionality of the subspace $d_{\epsilon_1}$, while for updates to the second layer $W_2$, the central novelty is combining the adaptivity of $W_1$ with the random features scaling polynomially in the dimension.
>
> > What happens if there is a mismatch between teacher and student depth? Is there any benefit to using a three‑layer network to learn a two‑layer teacher model?
>
> The benefit in [65] towards using a three-layer network to learn a two-layer target arises primarily from the ability of the second layer to learn non-linear features, a capacity which the first layer explicitly lacks. For general depth, however, all intermediate layers can learn non-linear features. We conjecture that for our class of targets, an $L$-layer model suffices to reach the optimal sample-complexity for an $L$-layer deep MIGHT.
>
> > I could not follow the part regarding $a^\star_i$. It is introduced as sampled from a distribution but later fixed to all ones. Is there a mistake here?
>
>
> Our class of targets is defined with a general distribution over $a^\star$. However, for technical reasons, such as the requirement for spherical symmetry, our proofs are limited to the case $a^\star_i = 1$. In the revised version, we will clarify this point along with reporting that the simulations have been carried out for $a^\star_i \neq 1$, but rather drawn i.i.d. from a Rademacher distribution.
>
> Thank you for pointing out the typos that will be fixed in the revised version and the valuable observations. We remain available for any additional clarification.

---

> > ### Comment · Reviewer_G7h1 · 2025-08-04
> >
> > Thank you for addressing my comments. I will keep my score.

---

### Decision · Program_Chairs · 2025-09-17

**Decision:**

Accept (spotlight)

**Comment:**

This submission studies gradient-based learning of a hierarchical target function class motivated by deep neural networks. The main contribution lies in providing optimization and statistical guarantees demonstrating that a deep network can progressively reduce dimensionality through layer-wise training, thereby achieving a sample complexity advantage over shallow models. The reviewers found the results nontrivial and of clear interest to the community. The meta-reviewer therefore recommends acceptance.